# Convergence of Actor-Critic Methods with Multi-Layer Neural Networks

**Haoxing Tian, Ioannis Ch. Paschalidis, Alex Olshevsky**
Department of Electrical and Computer Engineering
Boston University
Boston, MA 02215, USA
{tianhx, yannisp, alexols}@bu.edu

## Abstract

The early theory of actor-critic methods considered convergence using linear function approximators for the policy and value functions. Recent work has established convergence using neural network approximators with a single hidden layer. In this work we are taking the natural next step and establish convergence using deep neural networks with an arbitrary number of hidden layers, thus closing a gap between theory and practice. We show that actor-critic updates projected on a ball around the initial condition will converge to a neighborhood where the average of the squared gradients is $\tilde{O}\left(1/\sqrt{m}\right) + O\left(\epsilon\right)$, with $m$ being the width of the neural network and $\epsilon$ the approximation quality of the best critic neural network over the projected set.

## 1 Introduction

*Reinforcement Learning (RL)* has emerged as a powerful tool for solving decision-making problems in a model-free way. Among the various RL algorithms, the *Actor-Critic (AC)* method (Konda & Tsitsiklis (1999); Barto et al. (1983)) has shown great success in various domains, including robotics, game playing, and control systems (LeCun et al. (2015); Mnih et al. (2016); Silver et al. (2017)). AC involves simultaneous updates of two networks: an actor network that employs policy gradient (Sutton et al. (1999)) to update a parameterized policy, and a critic network which is driven by the *Temporal Differences (TD)* in the estimated value function. While AC methods with neural networks used for both actor and critic have achieved widespread use in practice, a fully satisfactory analysis of their convergence guarantees is currently lacking.

In recent years, a number of theoretical studies of AC have obtained provable convergence rates and performance analyses. Almost all works in this area assumed linear, rather than neural network-based, approximators for both actor and critic. A "two-timescale" linear AC was analysed in Wu et al. (2020), with a convergence rate of $\tilde{O}(T^{-1/4})$, where $T$ is the total number of iterations and $\tilde{O}\left(\cdot\right)$ refers to potential logarithmic terms omitted from the notation; the term "two-timescale" refers to the fact that the stepsizes for the actor update and critic update are not proportional to each other, but rather the actor steps are asymptotically negligible compared to the critic steps. A "single-timescale" linear AC method was considered in Olshevsky & Gharesifard (2022); Chen et al. (2021) and both works obtained a convergence rate of $O\left(T^{-0.5}\right)$ under an i.i.d. sampling assumption on the underlying MDP. The more realistic Markov sampling case was analyzed in the recent paper Chen & Zhao (2022), which also established a convergence rate of $\tilde{O}\left(T^{-0.5}\right)$. All these results relied on linear approximations.

To our knowledge, convergence rates for AC with neural approximators were analyzed only in two recent works Wang et al. (2019); Cayci et al. (2022). Both of these papers considered neural net-

works with a single hidden layer. The paper Wang et al. (2019) obtained a convergence rate of $O\left(T^{-0.5}\right)$ with a final error of $O\left(m^{-0.25}\right)$ under i.i.d. sampling, where $m$ is the width of hidden layer. The case of Markov sampling was considered in Cayci et al. (2022) which improved this to $\tilde{O}\left(T^{-0.5}\right)$ and $\tilde{O}\left(m^{-0.5}\right)$, respectively. Further, both Wang et al. (2019); Cayci et al. (2022) considered "double-loop" methods where, in the inner loop, the critic takes sufficiently many steps to accurately estimate $Q$-values. Such double-loop methods do not match prevailing practice and are considerably easier to analyze since they can be shown to approximate gradient descent.

Further, Cayci et al. (2022) required a projection onto a ball of radius $O(m^{-1/2})$ around the initial condition. Although a full representation theory for such neural networks is unknown, this is clearly limiting as compared to Wang et al. (2019) which only required projection onto a ball of constant radius. For nonlinear approximations, such projections are usually needed to stabilize the algorithm; without them, AC can diverge both in theory and practice.

Table 1: Comparisons with previous work.

| Reference | Algorithm | Sampling | Approximation | Projection Radius | Convergence rate | |
| --- | --- | --- | --- | --- | --- | --- |
| | | | | | w.r.t. $T$ | w.r.t. $m$ |
| Wu et al. (2020) | Two-timescale Single-loop | Markov | Linear | N/A | $\tilde{O}\left(T^{-0.4}\right)$ | N/A |
| Olshevsky & Gharesifard (2022) | Single-timescale Single-loop | I.i.d. | Linear | N/A | $O\left(T^{-0.5}\right)$ | N/A |
| Chen et al. (2021) | Single-timescale Single-loop | I.i.d. | Linear | N/A | $O\left(T^{-0.5}\right)$ | N/A |
| Chen & Zhao (2022) | Single-timescale Single-loop | Markov | Linear | N/A | $\tilde{O}\left(T^{-0.5}\right)$ | N/A |
| Wang et al. (2019) | Double-loop | I.i.d. | Single hidden layer | Constant | $O\left(T^{-0.5}\right)$ | $O\left(m^{-0.25}\right)$ |
| Cayci et al. (2022) | Double-loop | Markov | Single hidden layer | Decaying | $\tilde{O}\left(T^{-0.5}\right)$ $m$ sufficiently large | $\tilde{O}\left(m^{-0.5}\right)$ |
| Ours | Single-timescale Single-loop | Markov | Any depth | Constant | $\tilde{O}\left(T^{-0.5}\right)$ | $\tilde{O}\left(m^{-0.5}\right)$ |

The main contribution of this paper is to provide the first analysis of AC with neural networks of arbitrary depth. While replicating the earlier results of a $\tilde{O}\left(T^{-0.5}\right)$ convergence rate and $\tilde{O}\left(m^{-0.5}\right)$ error, our work considers a *single-loop* method with proportional step-sizes (sometimes called "single-timescale"). We prove this result under Markov sampling and project onto a ball of constant radius around the initial condition. An explicit comparison of our result to previous work is given in Table 1. A more technical comparison is also given later after the statement of our main result.

Our main technical tool is the so-called "gradient splitting" view of TD learning. This idea began with the paper Ollivier (2018) which observed that TD learning is exactly gradient descent when the underlying policy is such that the state transition matrix is reversible. In Liu & Olshevsky (2021), this was generalized to non-reversible policies by introducing the notion of a "gradient splitting" (discussed formally later in this work) and observing that, for linear approximation, TD updates are an example of gradient splitting. Gradient splitting is closely related to gradient descent, and the two processes can be analyzed similarly. A generalization to neural TD learning was given in Tian et al. (2023), which argued for an interpretation of nonlinear TD as approximate gradient splitting.

The analysis of AC that we perform in this work is trickier because both actor and critic updates rely on each other, and one must prove that the resulting errors in each process do not compound in interaction with each other. This difficulty arises because we do not consider the "double loop" case where the actor can effectively wait for the critic to converge, so that actor steps resemble gradient steps with error; rather both actor and critic update simultaneously their (imperfect) estimates. Similarly to what was done in Olshevsky & Gharesifard (2022), we show that we can draw on some ideas from control theory to prove that the resulting process converges with a so-called "small-gain" analysis.

## 2 Preliminaries

We begin by standardizing notation and stating the key concepts that will enable us to formulate our results alongside all the assumptions they require.

## 2.1 Markov Decision Processes (MDP)

A finite discounted-reward MDP can be described by a tuple $(S, A, P_{\text{env}}, r, \gamma)$ where $S$ is a finite state-space whose elements are vectors, and we use $s_0 \in S$ to denote the starting state; $A$ is a finite action space with cardinality $n_a$; $P_{\text{env}} = (P_{\text{env}}(s'|s,a))_{s,s' \in S, a \in A}$ is the transition probability matrix, where $P_{\text{env}}(s'|s,a)$ is the probability of transitioning from $s$ to $s'$ after taking action $a$; $r : S \times A \to \mathbb{R}$ is the reward function, where $r(s,a)$ stands for the expected reward at state $s$ and taking action $a$; and $\gamma \in (0, 1)$ is the discount factor.

A policy $\pi$ is a mapping $\pi : S \times A \to [0, 1]$ where $\pi(a|s)$ is the probability that the agent takes action $a$ in state $s$. Given a policy $\pi$, we can define the state transition matrix $P'_\pi = (P'_\pi(s'|s))_{s,s' \in S}$ and the state-action transition matrix $P_\pi = (P_\pi(s', a'|s, a))_{(s,a),(s',a') \in S \times A}$ as

$$P'_\pi(s'|s) = \sum_{a \in A} P_{\text{env}}(s'|s,a)\pi(a|s), \quad P_\pi(s', a'|s, a) = P_{\text{env}}(s'|s,a)\pi(a'|s').$$

The stationary distribution over state-action pairs $\mu_\pi$ is defined to be a nonnegative vector with coordinates summing to one and satisfying $\mu_\pi^T = \mu_\pi^T P_\pi$, while the stationary distribution over states $\mu'_\pi$ is defined similarly with $\mu'_\pi{}^T = \mu'_\pi{}^T P'_\pi$. The Perron-Frobenius theorem guarantees that such a $\mu_\pi$ and $\mu'_\pi$ exist and are unique subject to some conditions on $P'_\pi, P_\pi$, e.g., aperiodicity and irreducibility (Gantmacher (1964)). We use $\mu_\pi(s, a)$ to denote each entry of $\mu_\pi$ and $\mu'_\pi(s)$ each entry of $\mu'_\pi$. Clearly,

$$\mu_\pi(s, a) = \mu'_\pi(s)\pi(a|s). \tag{1}$$

The value function and the $Q$-function of a policy $\pi$ is defined as:

$$V_\pi^*(s) = \sum_{a \in A} \pi(a|s)Q_\pi^*(s, a), \quad Q_\pi^*(s, a) = \mathbb{E}_{s,a,\pi}\left[\sum_{t=0}^{+\infty} \gamma^t r(s_t, a_t)\right]. \tag{2}$$

Here, $\mathbb{E}_{s,a,\pi}$ stands for the expectation when action $a$ is chosen in state $s$ and all subsequent actions are chosen according to policy $\pi$. Throughout the paper, if $\pi$ can be parameterized by $\theta$, then we will use $\theta$ as a subscript instead of $\pi$, e.g., by writing $V_\theta^*(s)$ instead of $V_{\pi_\theta}^*(s)$.

If $\pi$ is parameterized by $\theta$, the $Q$-values satisfy the Bellman equation

$$Q_\theta^*(s, a) = r(s, a) + \gamma \sum_{s', a'} P_\theta(s', a'|s, a)Q_\theta^*(s', a'), \tag{3}$$

which can be stated in matrix notation as

$$Q_\theta^* = R + \gamma P_\theta Q_\theta^*, \tag{4}$$

where $Q_\theta^* = (Q_\theta^*(s, a))_{(s,a) \in S \times A}$ and $R = (R(s, a))_{(s,a) \in S \times A}$ are vectors that stack up the $Q$-values and rewards, respectively. We will assume rewards are bounded:

**Assumption 2.1** (Bounded Reward). *For any $s, a \in S \times A$, $|r(s, a)| \leq r_{\max}$.*

This assumption is commonly adopted throughout the literature, e.g., among the previous literature in Cayci et al. (2022); Wu et al. (2020). An obvious implication of this is an upper bound on the $Q$-values for any policy:

$$|Q_\theta^*(s, a)| \leq \frac{r_{\max}}{1 - \gamma}. \tag{5}$$

## 2.2 The Policy Gradient Theorem

We introduce the quantity $\phi_\theta(s)$, commonly called the discounted occupation measure which is defined as

$$\phi_\theta(s) = \sum_{t=0}^{+\infty} \gamma^t P_\theta(S_t = s),$$

where $P_\theta(S_t = s)$ is the probability of being in state $s$ after $t$ steps —- and recall that we always begin in state $s_0$. Next, we define $\phi_\theta(s, a)$ as

$$\phi_\theta(s, a) = \phi_\theta(s)\pi(a|s, \theta).$$

Note that the sum of both $\phi(s)$ and $\phi(s, a)$ equal to $(1 - \gamma)^{-1}$ rather than 1:

$$\sum_{s \in S} \phi_\theta(s) = \sum_{(s,a) \in S \times A} \phi_\theta(s, a) = \frac{1}{1 - \gamma}. \tag{6}$$

Now we are prepared to state the policy gradient theorem Sutton & Barto (2018).

**Theorem 2.1.** *(Policy Gradient Theorem)*

$$\nabla V_\theta^* = \sum_{s \in S} \phi_\theta(s) \sum_{a \in A} Q_\theta^*(s, a) \nabla \pi(a|s, \theta)$$

It is standard to write this as

$$\nabla V_\theta^* = \sum_{(s,a) \in S \times A} \phi_\theta(s, a) Q_\theta^*(s, a) \nabla \ln \pi(a|s, \theta),$$

which can be further rewritten in matrix form as

$$\nabla V_\theta^* = \nabla \ln \pi(\theta)^T \Phi_\theta Q_\theta^*, \tag{7}$$

where $\Phi_\theta$ is a diagonal matrix stacking up the $\phi_\theta(s, a)$ as its diagonal entries.

## 2.3 Parameterized Value Function and Policy

We will now state the various assumptions we have on the policies and their parametrizations. We will say that a function $f : \mathbb{R} \to \mathbb{R}$ is $L$-Lipschitz if

$$|f(x) - f(y)| \le L|x - y|, \ \forall x, y,$$

and a differentiable function $f : \mathbb{R} \to \mathbb{R}$ is $H$-smooth if

$$|\nabla f(x) - \nabla f(y)| \le H|x - y|, \ \forall x, y.$$

We will be using a multi-layer neural network to approximate the $Q$ values under a policy. We basically follow the same setting as in Liu et al. (2020), with some changes as far as notation goes. Specifically, we define the following recursion

$$x^{(k)} = \frac{1}{\sqrt{m}} \sigma \left( w^{(k)} x^{(k-1)} \right), \text{ for } k \in \{1, \dots, K\},$$

where $\sigma$ is an activation function and $x^{(k)}$ stands for the value of $k$'th layer ($x^{(0)} \in S \times A$ is the input to this neural network). The neural network outputs $Q(s, a, w)$, which is defined as

$$Q(s, a, w) = \frac{1}{\sqrt{m}} b^T x^{(K)}.$$

Notice that the output is linear to $x^{(K)}$ as no activation function is applied here. While this formulation does not have a bias, it is equivalent to a formulation with a bias if we pad all inputs with a single 1, and add an additional node to every hidden layer that propagates this 1 to subsequent layers. We will assume that all the hidden layers have the same width which we denote by $m$, i.e., all the matrices $w^{(k)}$ have $m$ rows and all the vector $x^{(k)}, k \ge 1$ are $m$-dimensional. The total number of layers in the neural network is denoted by $K$.

For simplicity, we will make the following assumption on the neural network. Throughout the paper, we will use $|| \cdot ||$ for the standard $l_2$-norm.

**Assumption 2.2.** *(Neural architecture and initialization) Suppose the neural network satisfies the following properties:*

- *(Input assumption) Any input to the neural network satisfies $||x^{(0)}|| \le 1$.*

- *(Activation function assumption) $\sigma$ is $L_\sigma$-Lipschitz and $H_\sigma$-smooth.*

- *(Initialization assumption) Each entry of the vector $b$ satisfies $|b_r| \le 1, \forall r$, and each entry of $w^{(k)}$ is randomly chosen from $N(0, 1)$, independently across entries.*

Liu et al. (2020) showed that with these assumptions, the following result holds with high probability – which we state as an assumption for our work.

**Assumption 2.3.** *The absolute value of each entry of $x^{(k)}$ (the output of layer $k$ of the neural network) is $\tilde{O}_m(1)$ at initialization.*

Next, we will stack up the weights of different layers into a column vector $w$ consisting of the entries of the matrices $w^{(1)}, \ldots, w^{(K)}$, with its norm defined by

$$||w||^2 = \sum_{k=1}^{K} ||w^{(k)}||_F^2,$$

where $||\cdot||_F$ is the Frobenius norm. During the training process, only the weights $w$ will be updated while the final weights $b$ will be left to their initial value. For convenience, we define the vector $Q(w) = (Q(s,a,w))_{(s,a)\in S\times A}$ which stacks up $Q(s,a,w)$ over all state-action pairs $(s,a)$. While this vector will never be actually used in the execution of any algorithm we consider due to its high dimensionality, it will be useful in some of the arguments we will make. Finally, we assume the parametrization of the policy $\pi$ is smooth.

**Assumption 2.4** (Smooth parametrization). *For all $s, a$, the quantities $\pi(a|s, \theta)$, $\ln \pi(a|s, \theta)$ are $L_\pi$-Lipschitz and $L'_\pi$-Lipschitz with respect to $\theta$, respectively.*

Note that this forces us to use a smooth activation function and rules out non-differentiable activation functions such as ReLU. If a RELU-like activation is needed, one could use a GeLU or ELU activation (which are smooth versions of ReLU) and still satisfy the above assumption. Note, also, that this assumption implicitly assumes that all policies are exploratory in the sense of assigning a positive probability to each action, since the derivative of $\ln x$ blows up as $x \to 0$.

## 2.4 Neural Actor-Critic

We will use $\mathbf{Proj}_W\{\cdot\}$ refer to projection onto a ball with constant radius around the initial condition of the critic, where
$$W = \{w \mid ||w - w_0|| \le \sigma_w\}, \ \sigma_w \text{ is a constant.}$$

We now introduce the neural AC, which updates the actor and critic parameters as

$$w_{t+1} = \mathbf{Proj}_W\left\{w_t + \alpha^w \delta_t \nabla_w Q(s_t, a_t, w_t)\right\}, \quad \theta_{t+1} = \theta_t - \frac{\alpha^\theta}{1-\gamma} Q(\hat{s}_t, \hat{a}_t, w_t) \nabla_\theta \ln \pi(\hat{a}_t|\hat{s}_t, \theta_t).$$

where $\delta_t$ is the TD error defined by

$$\delta_t = r(s_t, a_t) + \gamma Q(s'_t, a'_t, w_t) - Q(s_t, a_t, w_t), \tag{8}$$

and the samples are obtained as follows:

1. the state $s_t$ is generated by taking a step in the Markov chain $P_{\text{env}}$ from $s_{t-1}$;

2. the action $a_t$ is chosen according to the policy $\pi(a|s_t, \theta_t)$;

3. the next state $s'_t$, i.e, $s'_t = s_{t+1}$, is determined according to the transition probability $P_{\text{env}}$ of the MDP;

4. the action $a'_t$ is an action chosen at the next state according to the policy $\pi(a|s'_t, \theta_t)$;

5. the state-action pair $(\hat{s}_t, \hat{a}_t)$ is obtained by first sampling a geometric random variable $T$ with distribution $\{P(T = t) = (1 - \gamma)\gamma^t, t \ge 0\}$, and second obtaining $T$ transitions by starting at $s_0$ and taking actions according to $\pi(a|s, \theta_t)$. Note that this update has to be re-done at every step, i.e., every $t$ requires $\text{Geom}(\gamma)$ steps.

The above algorithm will be referred to as *actor-critic with Markov sampling*. It is also possible to consider a simplified variant, where step 1 is slightly altered as follows: the state $s_t$ is instead chosen i.i.d. at every step from the stationary distribution of $\mu_{\theta_t}$ of the policy $\pi_{\theta_t}$. This is referred to as *actor-critic with i.i.d. sampling*.

### 2.4.1 Approximation Assumptions

It is evident that any performance bound on AC will depend on how well the neural network used for the critic can approximate the true value function. If we choose a neural network architecture for which universal approximation theorems do not apply and it happens to poorly approximate the true $Q$-functions, we will likely obtain poor results. Here, we will largely sidestep this issue by defining $\epsilon$ to be the approximation quality of the critic; our final performance results will be in terms of $\epsilon$.

Formally, we say that the vector $Q$ is an $\epsilon$-approximation to the true value function $Q^*_{\theta_t}$ of the policy $\pi_{\theta_t}$ if

$$\max_{(s,a)\in S\times A} |Q(s,a) - Q^*_{\theta_t}(s,a)| \le \epsilon. \tag{9}$$

We then make the following assumption.

**Assumption 2.5.** *(Approximation capabilities of critic) For all $\theta$, there exists some set of weights $\hat{w}_\theta$ which give rise to an $\epsilon$-approximation of $Q^*_\theta$.*

Note that, since we do not say what $\epsilon$ is, this assumption could well be a definition of $\epsilon$. Throughout the paper we will use $\hat{Q}^*_{\theta_t}$ to denote an $\epsilon$-approximation to $Q^*_{\theta_t}$ guaranteed by the above assumption. Thus,

$$Q(\hat{w}_{\theta_t}) = \hat{Q}^*_{\theta_t}.$$

Further, we will assume that $\hat{w}_\theta$ is a smooth function of $\theta$ in the sense of its first and second derivatives.

**Assumption 2.6.** *(Smoothness of critic approximation) Suppose there exists scalars $L_w(i)$ and $H_w(i)$ such that for all $\theta$,*

$$||\nabla \hat{w}^*_\theta(i)|| \le L_w(i), \quad \lambda_{\max}\{\nabla^2 \hat{w}^*_\theta(i)\} \le H_w(i).$$

*where $\lambda_{\max}\{\cdot\}$ stands for the largest eigenvalue.*

For convenience, we define

$$L_w = \sqrt{\sum_i L_w(i)^2}, \quad H_w = \sqrt{\sum_i H_w(i)^2}.$$

Finally, we need an additional assumption on the critic neural network. It should be obvious that any analysis of actor-critic has to assume that the critic *is capable* of approximating the correct $Q$-values. One part of this was already assumed earlier in Assumption 2.5, where we assumed that an approximation exists. However, it should be clear that in the nonlinear case this is insufficient: just because there exists an approximation which is good doesn't follow that it will be found during training, which is not known converge to the global minimizer in the nonlinear case, but rather only to a critical point.

We thus need something to rule out the possibility that the critic training gets stuck at a bad crtical point. It turns out that it suffices to assume (a quantitative version of the fact that) the critic is one-to-one map from weights to value functions.

**Assumption 2.7.** *(State regularity) There exists some constant $\lambda' > 0$ such that*

$$||Q(w) - \hat{Q}^*_\theta|| \ge \lambda'||w - \hat{w}^*_\theta||.$$

Let us parse the meaning of this assumption. Because $Q(\hat{w}^*_\theta) = \hat{Q}^*_\theta$, it is appropriately viewed as a quantitative version of the statement that if $w_1 \ne w_2$, then $Q(w_1) \ne Q(w_2)$. To see why this makes sense, note that the number of states is typically many magnitudes larger than the number of parameters in the critic. For example, in many applications the number of states often corresponds to the number of images (when states are captured through images) which is astronomical. Thus $Q(w)$ will map $w$ to a much higher dimensional space.

If the states $s$ are generated from a probability distribution which has a continuous density, and the activation functions are continuous and increasing, the chance that $Q_{w_1}(s) = Q_{w_2}(s)$ even for one state $s$ is zero. That is why we label it "state regularity" as above (and recall that $Q(w)$ stacks up $Q_w(s)$ for every state $s$).

On a technical level, this property ensures that critic actually finds a good critic approximation in spite of the nonlinearity of the update. If the features are linear, this reduces to the assumption that the features are linearly independent, an assumption which is made in all previous and related work on AC method (Wu et al. (2020); Olshevsky & Gharesifard (2022); Chen & Zhao (2022); Kumar et al. (2023)) and TD Learning (Liu & Olshevsky (2021); Xu & Gu (2020); Cai et al. (2019); Zou et al. (2019)).

## 2.5  The Mixing of Markov Chains

It is standard to make an assumption to the effect that all the Markov chains that can arise satisfy a mixing condition. Otherwise, it is possible under Markov sampling for the state to fail to explore the entire state-space. This assumption, first introduced by Bhandari et al. (2018) in TD learning, now is commonly used in AC analysis (Olshevsky & Gharesifard (2022); Wu et al. (2020); Chen & Zhao (2022)).

**Assumption 2.8** (Markov chain mixing). *There exists constants $C > 0$ and $\beta \in [0, 1)$ with the following property: for all $\theta$, if we consider a Markov chain generated by $a_t \sim \pi(\cdot|s_t, \theta)$, $s_{t+1} \sim P_{\mathrm{env}}(\cdot|s_t, a_t)$ starting from state $s$, then*

$$||p_\tau - \mu'_\theta||_1 \leq C\beta^\tau, \forall \tau \geq 0, \forall s \in S,$$

*where $p_\tau$ is the probability distribution of the state of this Markov chain after $\tau$ steps.*

To assure AC explores every possible state, we make the following assumption:

**Assumption 2.9.** *(Exploration) Suppose there exists some constant $\mu_{\min} > 0$ such that, for all $\theta$, $\mu'_\theta$ is uniformly bounded away from 0. In other words,*

$$\mu'_\theta \geq \mu_{\min} > 0, \forall \theta.$$

Recall that $\mu_\theta$ was defined earlier to be the stationary distribution of the transition matrix associated with the policy $\pi_\theta$. A key point is that the constants $C$, $\beta$ and $\mu_{\min}$ in the above assumptions do not depend on $\theta$.

We note that there is some redundancy in our assumptions. As discussed above, we require $\ln \pi_\theta(a|s)$ to have a smooth gradient for all $s, a$, which ensures that $\pi_\theta$ assigns a strictly positive probability to every action. This implies Assumptions 2.8 and 2.9 which can therefore be made into propositions. Nevertheless, we explicitly make Assumptions 2.8 and 2.9 (even though both of them are actually implied by our earlier assumption) since the quantities appearing in them (specifically, the mixing time $\beta$ and the constant $\mu_{\min}$) appear in various bounds we will derive.

More precisely, we follow the earlier literature by setting $C\beta^\tau$ to be proportional to $T^{-0.5}$, the typical of stepsize in Stochastic Gradient Descent. We call the smallest $\tau$ such that $C\beta^\tau \leq O\left(T^{-0.5}\right)$ the mixing time and denote it by $\tau_{\mathrm{mix}}$. It is easy to see that $\tau_{\mathrm{mix}} = O\left((1-\beta)^{-1}\log T\right)$. The quantity $\tau_{\mathrm{mix}}$ will appear throughout our paper.

## 2.6  $D$-norm and Dirichlet Norm in MDPs

A key ingredient is our analysis is the choice of norm: we have found that a certain norm originally introduced in Ollivier (2018) significantly simplifies analysis of the problem. We next introduce this norm and state our assumptions about it.

Let $D_\theta = \mathrm{diag}(\mu_\theta(s, a))$ be the diagonal matrix whose elements are given by the entries of the stationary distribution $\mu_\theta$ associated with the policy $\pi_\theta$. Given a function $f : S \times A \to \mathbb{R}$, its $D$-norm is defined as

$$||f||_D^2 = f^T D_\theta f = \sum_{(s,a)\in S\times A} \mu_\theta(s, a) f(s, a)^2. \tag{10}$$

The $D$-norm is similar to the Euclidean norm except each entry is weighted proportionally to the stationary distribution. We also define the Dirichlet semi-norm of $f$:

$$||f||_{\mathrm{Dir}}^2 = \frac{1}{2} \sum_{(s,a),(s',a')\in S\times A} \mu_\theta(s, a) P_\theta(s', a'|s, a)(f(s', a') - f(s, a))^2. \tag{11}$$

A semi-norm satisfies the axioms of a norm except that it may be equal to zero at a non-zero vector. Note that $||f||_{\text{Dir}}$ depends on the policy both through the stationary distribution $\mu_\theta(s, a)$ as well as through the transition matrix $P_\theta$.

Finally, following Ollivier (2018), the weighted combination of the $D$-norm and the Dirichlet semi-norm is denoted as $\mathcal{N}_\theta(f)$ will be defined

$$\mathcal{N}_\theta(f) = (1 - \gamma)||f||_D^2 + \gamma||f||_{\text{Dir}}^2. \tag{12}$$

Note that as long as $\mu_\theta(s, a) > 0$, which is stated in Assumption 2.9, for all $s, a$, we have that $\sqrt{\mathcal{N}_\theta(f)}$ is a valid norm.

## 3  Our Main Results

To simplify the expression that follow, we will adopt the notations $\Delta_V$ and $\Delta_Q$ for the two losses that we want to bound in our paper:

$$\Delta_V = \frac{1}{T}\sum_{t=1}^T \mathbb{E}\left[||\nabla V_{\theta_t}^*||^2\right], \quad \Delta_Q = \frac{1}{T}\sum_{t=1}^T \mathbb{E}\left[\mathcal{N}_{\theta_t}(Q(w_t) - \hat{Q}_{\theta_t}^*)\right]. \tag{13}$$

Intuitively, $\Delta_V$ corresponds to the actor error: ideally, we want to reach a point where the gradient of the actor value function is zero. Note that, since the value function is not convex in general, the actor error is measured in terms of distance to a stationary point as above.

Similarly, $\Delta_Q$ is a measure of the critic error: it equals zero precisely if $Q(w_t)$, the approximator of $Q$-function, equals $\hat{Q}_{\theta_t}$. Of course, as discussed above, the critic neural network may not be able to perfectly represent the true $Q$-function. Now we are ready to state our main results.

**Theorem 3.1.** *Consider the neural AC algorithm mentioned in Section 2.4. Suppose Assumptions 2.1-2.9 hold and the step-sizes $\alpha^\theta$ and $\alpha^w$ are both chosen to decay proportionally to $O\left(T^{-0.5}\right)$.*

1. *In the i.i.d. sampling case,*

$$\Delta_V \leq O\left(\frac{1}{\sqrt{T}}\right) + O(\epsilon) + \tilde{O}\left(\frac{1}{\sqrt{m}}\right), \quad \Delta_Q \leq O\left(\frac{1}{\sqrt{T}}\right) + O(\epsilon) + \tilde{O}\left(\frac{1}{\sqrt{m}}\right).$$

2. *In the Markov sampling case,*

$$\Delta_V \leq O\left(\frac{(\log T)^2}{\sqrt{T}}\right) + O(\epsilon) + \tilde{O}\left(\frac{1}{\sqrt{m}}\right), \quad \Delta_Q \leq O\left(\frac{(\log T)^2}{\sqrt{T}}\right) + O(\epsilon) + \tilde{O}\left(\frac{1}{\sqrt{m}}\right).$$

In all $O(\cdot)$ notations above, we treat factors that do not depend on $T, \epsilon, m$ as constants.

We next provide a more detailed comparison to the previous works of (Wang et al. (2019); Cayci et al. (2022)). Our discussion partially reprises the discussion in the Introduction, but can now be discussed at a greater level of detail:

- **Arbitrary depth/single-timescale.** The main contribution of this paper to provide an analysis that applies to neural networks of arbitrary depth. Moreover, we do so in a single-loop/single-timescale method where the critic and actor iterate simultaneously, which is matching what is typically done in practice. Such an analysis is inherently more technically challenging, since when the actor can wait for the critic to go through sufficiently many iterations, one could argue that the resulting $Q$-values are approximately accurate and the process resembles gradient descent.

- **Representability.** Both previous works for the single-layer case assume the $Q$-function lies in some function class, which, as discussed after Assumption 6 in Farahmand et al. (2016), is one kind of "no function approximation error" assumption. By contrast, we make no such assumption: rather we allow any approximation error for the critic $\epsilon$, and our final result is given in terms of $\epsilon$.

- **Lower bound on $m$.** Previous works require $m$, the width of neural network, to be sufficiently large. In Wang et al. (2019), given that $m$ is sufficiently large, Section 3.1 and Corollary A.3 argue that the gradient, denoted by $\bar{\phi}_\theta$ and $\bar{\phi}_w$, can be well approximated by the "centered feature mapping corresponding to the initialization", denoted by $\bar{\phi}_0$. In Cayci et al. (2022), this dependency is even more emphasised since the upper bound shown in Theorem 2 could diverge with small $m$.

- **Relation to NTK theory.** NTK theory (Jacot et al. (2018)) tells us that neural networks get more linear as $m \to \infty$. The classic analyses of this proceed by arguing that as $m \to \infty$, the neural network stays close to its initialization during training Chizat et al. (2019). In that sense, we should expect to get a convergence result for AC as $m \to \infty$, but if the critic neural network stays close to its initial condition, the algorithm will effectively be using random linear features at initialization. For this reason, it is desirable not to argue that the critic neural network always stays close to its initial condition. We do not use such an argument in this work, whereas both Wang et al. (2019) and Cayci et al. (2022) obtain their results by arguing that the critic neural network stays close to its initial condition. This theoretical distinction is shown in Tian et al. (2023) to match what happens in simulations, which shows empirically that even for projected neural TD, the critic neural network will move to the *boundary* of the projection ball.

- **Linearization.** Previous works assume some kind of linearization around the initial point. The objective is explicitly linearized in Wang et al. (2019).In Cayci et al. (2022), while the objective is not linearized, the neural networks weights are projected onto a radius of size $O(1/\sqrt{m})$ around the initial point.

## 4 Tools in Our Analysis

### 4.1 Choice of Norm and Gradient Splitting

A linear function $h(\theta)$ is said to be a gradient splitting of a convex quadratic $f(\theta)$ minimized at $\theta = a$ if

$$\frac{1}{2}\nabla f(\theta)^T(a - \theta) = h(\theta)^T(\theta - a). \tag{14}$$

In other words, a splitting $h(\theta)$ has exactly the same inner product with the "direction to the optimal solution" as the true gradient of $f(\theta)$ (up to the factor of $1/2$). The connection between this idea and RL was made in the following papers:

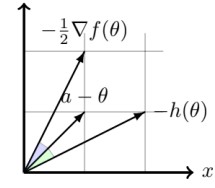

- In Ollivier (2018) it was shown that in TD Learning, if the matrix $P$ corresponds to a reversible Markov chain, then $E[\bar{g}(\theta_t)] = \nabla_\theta \mathcal{N}(f)$ for some $f$. This makes Neural TD easy to analyze in the reversible case as it is exactly gradient descent.

- In Liu & Olshevsky (2021), it was shown how to further use the function $\mathcal{N}(\cdot)$ to analyze TD learning with linear approximation when the policy is not necessarily reversible. *In particular, it was shown that the mean update of TD with linear approximation is a gradient splitting of the function $\mathcal{N}(\cdot)$. This is one of the crucial ideas we build on in this paper.*

Figure 1: Key property of gradient splitting: $h(\theta)$ has the same inner product with $a - \theta$ as $\nabla f(\theta)$ up to a factor of $1/2$.

### 4.2 Nonlinear Small-Gain Theorem

Inspired by Olshevsky & Gharesifard (2022), our second main tool is a nonlinear version of the small-gain theorem . Because the actor and critic update simultaneously, we need to rule out the possibility that errors in the actor compound with errors in the critic to create divergence. For example, it is conceivable that, when the policy is fixed, the critic converges to a reasonable approximation; when the critic is fixed, the actor converges to an approximate of the stationary point; but both updating simultaneously results in divergence.

The core idea of small-gain is to write these updates in such a way so that one can argue that if certain coefficients are small enough, this "interconnection" of the actor and critic systems converges. The

small-gain theorem we use is a nonlinear version of the textbook version Drazin (1992). This is a widely-used trick in control theory that avoids the necessity of explicitly finding a Lyapunov function.

## 5 Conclusion

We have provided an analysis of Neural AC using a convex combination of the $D$-norm and the Dirichlet semi-norm to describe the error. Our main result is an error rate of $O\left(T^{-0.5} + \epsilon\right) + \tilde{O}\left(m^{-0.5}\right)$ under the i.i.d. sampling and $O\left((\log T)^2 \cdot T^{-0.5} + \epsilon\right) + \tilde{O}\left(m^{-0.5}\right)$ under the Markov sampling for neural networks of arbitrary depth. Crucially, our proof does not make assumptions that force the neural networks to stay close to their initial conditions, relying instead on arguments that show that neural networks which are not "too nonlinear" will still converge to an approximate minimum.

## Acknowledgments and Disclosure of Funding

This research was partially supported by the NSF under grants CCF-2200052, DMS-1664644, and IIS-1914792, by the ONR under grant N00014-19-1-2571, by the DOE under grant DE-AC02-05CH11231, by the NIH under grant UL54 TR004130, and by Boston University.

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

# A   Sketch of Proof

In this section we give a basic idea of how we prove Theorem 3.1. Briefly speaking, our idea contains two directions: First, the Critic error (captured by $\Delta_Q$) can be upper-bounded by the Actor error (captured by $\Delta_V$); Next, the Actor error can also be upper-bounded by the Critic error. Therefore, both errors are bounded and converge to $0$. Based on this idea, our proof can be divided into three steps:

*Step 1: Analysis of Actor update.*

In Appendix B, we first bound $\Delta_V$ by $\Delta_Q$ through Actor update. On one hand, by considering Actor update and comparing it with mean-path update (where we replace $g$ by $\bar{g}$), one would have

$$\mathbb{E}\left[\theta_{t+1} - \theta_t | \mathcal{F}_t\right] = -\alpha^\theta \nabla V_{\theta_t}^* - \alpha^\theta D_Q.$$

$$\mathbb{E}\left[||\theta_{t+1} - \theta_t||^2 | \mathcal{F}_t\right] \leq ||\alpha^\theta \nabla V_{\theta_t}^* + \alpha^\theta D_Q||^2 + \frac{4}{(1-\gamma)^2}\alpha^{\theta 2} U_g^2.$$

where $D_Q = \nabla \ln \pi(\theta_t)^T \Phi_{\theta_t}\left[Q(w_t) - Q_{\theta_t}^*\right]$, $\mathcal{F}_t = (w_t, \theta_t)$ and $U_g$ is defined in Lemma C.13.

On the other hand, Lemma C.12 suggests $V_\theta^*$ is smooth w.r.t. $\theta$. Hence,

$$V_{\theta_{t+1}}^* \leq V_{\theta_t}^* + \nabla V_{\theta_t}^*(\theta_{t+1} - \theta_t) + \frac{H_V}{2}||\theta_{t+1} - \theta_t||^2.$$

Our claim is a combination of the above facts and some simple calculations:

$$\left(\frac{\alpha_\theta}{2} - \alpha_\theta^2 H_V\right)\Delta_V \leq \frac{\mathbb{E}\left[V_{\theta_1}^* - V_{\theta_{T+1}}^*\right]}{T} + c_1\left(\alpha_\theta^2 H_V + \frac{\alpha_\theta}{2}\right)\Delta_Q$$
$$+ c_2\left(\alpha_\theta^2 H_V + \frac{\alpha_\theta}{2}\right)\epsilon^2 + \frac{2H_V}{(1-\gamma)^2}\alpha^{\theta 2}U_g^2.$$

We successfully bound $\Delta_V$ by $\Delta_Q$.

*Step 2: Analysis of Critic update.*

In Appendix C, we next bound $\Delta_Q$ by $\Delta_V$ through Critic update. Here we perform classical way of analysis, which begins with

$$\mathbb{E}\left[||w_{t+1} - \hat{w}_{\theta_{t+1}}^*||^2\right]$$
$$\leq \underbrace{\mathbb{E}\left[||w_t - \hat{w}_{\theta_t}^* + \alpha^w f(O_t, w_t)||^2\right]}_{I_1}$$
$$+ \underbrace{\mathbb{E}\left[2(\hat{w}_{\theta_t}^* - \hat{w}_{\theta_{t+1}}^*)^T(w_t - \hat{w}_{\theta_t}^* + \alpha^w f(O_t, w_t))\right]}_{I_2} + \underbrace{\mathbb{E}\left[||\hat{w}_{\theta_t}^* - \hat{w}_{\theta_{t+1}}^*||^2\right]}_{I_3}.$$

We treat the above three terms respectively. To address $I_1$, by comparing with mean-path update:

$$\mathbb{E}\left[||w_t - \hat{w}_{\theta_t}^* + \alpha^w f(O_t, w_t)||^2\right]$$
$$= \mathbb{E}\left[||w_t - \hat{w}_{\theta_t}^*||^2\right] + \underbrace{\mathbb{E}\left[2\alpha^w(w_t - \hat{w}_{\theta_t}^*)^T \bar{f}(w_t, \theta_t)\right]}_{I_{1,1}}$$
$$+ \underbrace{\mathbb{E}\left[\alpha^{w2}||f(O_t, w_t)||^2\right]}_{I_{1,2}} + \underbrace{\mathbb{E}\left[2\alpha^w(w_t - \hat{w}_{\theta_t}^*)^T\left[f(O_t, w_t) - \bar{f}(w_t, \theta_t)\right]\right]}_{I_{1,3}}.$$

Now let us examine this equation carefully. $I_{1,1}$ is the inner product between $w_t - \hat{w}_{\theta_t}^*$ and the mean-path update $\bar{f}(w_t, \theta_t)$, which can be captured by gradient splitting; $I_{1,2}$ decays as $\alpha_t^2$, so a loose bound on $||f(O_t, w_t)||^2$ is enough (See Lemma C.15); $I_{1,3}$ is Markov sampling noise, which is handled using the same procedure as in Bhandari et al. (2018).

To discuss more about how to address Markov sampling noise, the idea is to using Assumption 2.8 to show that, after $\tau_{\text{mix}}$ steps, the distance between distribution of agent and the stationary distribution

decaying geometrically, and thus $I_{1,3}$ also decays geometrically. However, there is still a lot of difficulties to apply the same analysis in our work since TD(0) is considered in Bhandari et al. (2018) while Actor-Critic methods is considered here. The difficulties is induced by the constant changing of policy in every time steps during training. We introduce an auxiliary chain (See the definitions before Lemma C.10) to further address the changing of policy problem inspired by Zou et al. (2019); Wu et al. (2020); Chen & Zhao (2022).

Now we move on to $I_2$. we notice that the dominate term is $\mathbb{E}\left[2(\hat{w}_{\theta_t}^* - \hat{w}_{\theta_{t+1}}^*)^T(w_t - \hat{w}_{\theta_t}^*)\right]$ since the remaining term $\mathbb{E}\left[2(\hat{w}_{\theta_t}^* - \hat{w}_{\theta_{t+1}}^*)^T \alpha^w f(O_t, w_t)\right]$ decays as $\alpha^\theta \alpha^w$ ($\alpha^\theta$ comes from $\|\hat{w}_{\theta_t}^* - \hat{w}_{\theta_{t+1}}^*\|$ which can be seen using Assumption 2.6). To handle the dominate term, we first view $\hat{w}_\theta^*$ as a function of $\theta$ and use a second order expansion as follows. Then the problem get solved after noticing that we already derive relationships on $\theta_{t+1} - \theta_t$ in Step 1.

$$\hat{w}_{\theta_{t+1}}^*(i) = \hat{w}_{\theta_t}^*(i) + \nabla \hat{w}_{\theta_t}^*(i)^T(\theta_{t+1} - \theta_t) + \frac{1}{2}(\theta_{t+1} - \theta_t)^T \nabla^2 \hat{w}_{\theta_t'}^*(i)(\theta_{t+1} - \theta_t).$$

To address $I_3$, we notice that it decays as $\alpha^{w2}$ as a direct result of Assumption 2.6.

Combine all of the above result we can finally arrive at the relationship between $\Delta_Q$ and $\Delta_V$.

*Step 3: Combine result from Step 1 and 2 by small-gain theorem.*

Now we are ready to use the Small Gain theorem. We fit the results from Step 1 and Step 2 by the following form:

$$x \leq a_1 y + a_2, \quad y \leq b_1 x + b_2 + b_3 \sqrt{y}.$$

Then, Small Gain theorem implies that $y$ can be upper bounded by the following inequality:

$$y \leq \frac{2b_2 + b_3^2 + 2a_2 b_1}{1 - 2a_1 b_1}.$$

Once we have a bound for $y$, we can easily compute a bound for $x$.

# B  Actor-Critic

In this section, we will review and clarify the AC algorithm being considered in this paper.

Recall that in Eq.(8), we defined the set $W$ as $W = \{w \mid ||w - w_0|| \leq \sigma_w\}$ and the TD error $\delta_t$ as

$$\delta_t = r(s_t, a_t) + \gamma Q(s'_t, a'_t, w_t) - Q(s_t, a_t, w_t).$$

With $\delta_t$, we now define function $f$ and $g$ such that

$$f(O_t, w_t) = \delta_t \nabla_w Q(s_t, a_t, w_t), \quad g(\hat{O}_t, w_t, \theta_t) = Q(\hat{s}_t, \hat{a}_t, w_t) \nabla_\theta \ln \pi(\hat{a}_t | \hat{s}_t, \theta_t), \tag{15}$$

where we denote by $O_t = (s_t, a_t, s'_t, a'_t) \in S \times A \times S \times A$ the tuple of $s_t, a_t, s'_t, a'_t$ and by $\hat{O}_t = (\hat{s}_t, \hat{a}_t) \in S \times A$ the $\hat{s}_t, \hat{a}_t$ pair. The way of sampling $O_t$ and $\hat{O}_t$ is mentioned in Section 2.4.

With these notations, the AC update mentioned in Section 2.4 can be written as

$$w_{t+1} = \mathbf{Proj}_W \left\{ w_t + \alpha^w f(O_t, w_t) \right\}, \quad \theta_{t+1} = \theta_t - \frac{\alpha^\theta}{1 - \gamma} g(\hat{O}_t, w_t, \theta_t).$$

We find it useful to talk about the "mean path update". This just means that the functions $f(\cdot, \cdot)$ and $g(\cdot, \cdot, \cdot)$ in Eq.(15) are replaced by their means, assuming that $(s_t, a_t)$ is sampled from $\mu_{\theta_t}$ while $(\hat{s}_t, \hat{a}_t)$ is sampled from $(1 - \gamma)\phi_{\theta_t}$. More formally, the mean-path update functions $\bar{f}(\cdot, \cdot)$ and $\bar{g}(\cdot, \cdot)$ are defined as

$$\bar{f}(w_t, \theta_t) = \sum_{s_t, a_t} \mu_{\theta_t}(s_t, a_t) \nabla Q(s_t, a_t, w_t) \mathbb{E}_{s'_t, a'_t | s_t, a_t}[r(s_t, a_t) + \gamma Q(s'_t, a'_t, w_t) - Q(s_t, a_t, w_t)]$$

$$= \nabla Q(w_t)^T D_{\theta_t}(\gamma P_{\theta_t} - I)(Q(w_t) - Q^*_{\theta_t}),$$

$$\bar{g}(w_t, \theta_t) = \frac{1}{1 - \gamma} \mathbb{E}_{\hat{O}_t}[g(\hat{O}_t, w_t, \theta_t)] = \frac{1}{1 - \gamma} \mathbb{E}[g(\hat{O}_t, w_t, \theta_t) | \mathcal{F}_t] = \nabla \ln \pi(\theta_t)^T \Phi_{\theta_t} Q(w_t),$$
$$\tag{16}$$

where $\mathcal{F}_t = (w_t, \theta_t)$ and $\mathbb{E}_{O_t}$, $\mathbb{E}_{\hat{O}_t}$ assume $O_t$ follows $\mu_{\theta_t}$ and $\hat{O}_t$ follows $(1 - \gamma)\phi_{\theta_t}$. To show the latter one, as we discussed in Section 2.4, we first sample $T$ such that $P(T = t) = (1 - \gamma)\gamma^t$. We then perform $T$ transition starting from $s_0$. This mean that by total probability,

$$P(S = s) = \sum_{t=0}^{+\infty} P(S = s | T = t) \cdot P(T = t) = (1 - \gamma) \sum_{t=0}^{+\infty} \gamma^t P(S_t = s) = \phi(s).$$

Thus, if the policy here is given by $\theta_t$, it follows immediately that

$$P(\hat{O}_t = (s, a)) = (1 - \gamma)\phi_{\theta_t}(s)\pi(a | s, \theta_t) = (1 - \gamma)\phi_{\theta_t}(s, a).$$

Notice that under these notations, we have $\mathbb{E}_{\hat{O}_t}[g(\hat{O}_t, w_t, \theta_t)] = \mathbb{E}[g(\hat{O}_t, w_t, \theta_t) | \mathcal{F}_t]$.

Algorithm 1 details the algorithm considered in this paper.

---

**Algorithm 1** Actor-Critic

---

**Require:** Numbers of iterations $T$, learning rate $\alpha^w$ and $\alpha^\theta$, projection set $W$.
    Initialize $\theta_0$, $b_r$ and $w^{(k)}$ such that $|b_r| \leq 1, \forall r$ and every entry of $w^{(k)}$ is chosen from $N(0, 1)$.
    Initialize the starting state-action pair $s_0, a_0$.
    **for** $t \in \{1, 2, \ldots, T\}$ **do**
        Sample $s_t \sim P_{\text{env}}(s | s_{t-1}, a_{t-1})$, $a_t \sim \pi(a | s_t, \theta_t)$, $s'_t \sim P_{\text{env}}(s | s_t, a_t)$, $a'_t \sim \pi(a | s'_t, \theta_t)$.
        Sample $\hat{O}_t$ by first sampling a random variable $T$ with $P(T = t) = (1 - \gamma)\gamma^t$, and second obtaining $T$ transitions by starting at $s_0$ and taking actions according to $\pi(a | s, \theta_t)$.
        Compute $\delta_t$, $f(O_t, w_t)$, $g(\hat{O}_t, w_t, \theta_t)$, and update $w_{t+1}$ and $\theta_{t+1}$ as

$$w_{t+1} = \mathbf{Proj}_W \left\{ w_t + \alpha^w f(O_t, w_t) \right\}, \quad \theta_{t+1} = \theta_t - \frac{\alpha^\theta}{1 - \gamma} g(\hat{O}_t, w_t, \theta_t).$$

    **end for**

---

# C  Auxiliary Lemmas

In this section, we will present all the auxiliary lemmas needed to prove Theorem 3.1.

## C.1  Properties of the Neural Network

In this section, we will show that the neural network has Lipschitzness and smoothness properties. The following result is based on Liu et al. (2020) and has been talked about in Tian et al. (2023).

**Lemma C.1.** *For any* $(s,a) \in S \times A$, *there exists scalars* $L_Q(s,a), H_Q(s,a)$ *such that for* $w_1, w_2 \in W$,

$$||Q(s,a,w_1) - Q(s,a,w_2)|| \leq L_Q(s,a)||w_1 - w_2||.$$
$$||\nabla Q(s,a,w_1) - \nabla Q(s,a,w_2)|| \leq H_Q(s,a)||w_1 - w_2||.$$

*If we further define*

$$L_Q = \sqrt{\sum_{s,a} L_Q(s,a)^2}, \quad H_Q = \sqrt{\sum_{s,a} H_Q(s,a)^2},$$

*then* $L_Q = O(1)$ *and* $H_Q = \tilde{O}\left(\frac{1}{\sqrt{m}}\right)$ *with respect to* $m$.

*Proof.* The Lipschitzness property is proved in Tian et al. (2023) while the smoothness property is a direct result of Liu et al. (2020). $\qquad\square$

## C.2  Properties of the Operator $\mathcal{N}$

In this section, we will show several results about the operator $\mathcal{N}_\theta$ defined in Eq.(12).

**Lemma C.2.** *For any function* $f$ *defined on* $S \times A$,

$$-\mathcal{N}_\theta(f) = f^T D_\theta(\gamma P_\theta - I)f.$$

*Proof.* The proof is given by Lemma A.1 in Tian et al. (2023). $\qquad\square$

**Lemma C.3.** *There exists* $\lambda_{\min} > 0$ *and* $\lambda'_{\min} > 0$ *such that*

$$\mathcal{N}_\theta\left(Q(w) - \hat{Q}_\theta^*\right) \geq \lambda_{\min} \|w - \hat{w}_\theta^*\|^2,$$

*and*

$$\mathcal{N}_\theta\left(Q(w) - \hat{Q}_\theta^*\right) \geq \lambda'_{\min}\left\|Q(w) - \hat{Q}_\theta^*\right\|^2,$$

*where* $\lambda_{\min} = (1-\gamma)\mu_{\min}\lambda'^2$ *and* $\lambda'_{\min}$ *is given by* $\lambda'_{\min} = (1-\gamma)\mu_{\min}\lambda'^2 L_Q^2$.

*Proof.* To show the first part,

$$\begin{aligned}
\mathcal{N}_\theta\left(Q(w) - \hat{Q}_\theta^*\right) &= (1-\gamma)||Q(w) - \hat{Q}_\theta^*||_D^2 + \gamma||Q(w) - \hat{Q}_\theta^*||_{\text{Dir}}^2 \\
&\geq (1-\gamma)||Q(w) - \hat{Q}_\theta^*||_D^2 \\
&\geq (1-\gamma)\mu_{\min}\lambda'^2 \|w - \hat{w}_\theta^*\|^2,
\end{aligned}$$

where the first line is the definition of $\mathcal{N}(\cdot)$ while the last line uses Assumption 2.9. We can set $\lambda_{\min} = (1-\gamma)\mu_{\min}\lambda'^2$ and we finish the proof for the first part.

The second part is an obvious result that simply combines the first part and Lemma C.1. $\qquad\square$

**Lemma C.4.** *Suppose* $D_Q = \nabla \ln \pi(\theta_t)^T \Phi_{\theta_t}\left[Q(w_t) - Q_{\theta_t}^*\right]$. *The relationship between* $D_Q$ *and* $\mathcal{N}_{\theta_t}(Q(w_t) - \hat{Q}_{\theta_t}^*)$ *can be described as follows:*

$$\|D_Q\|^2 \leq c_1 \cdot \mathcal{N}_{\theta_t}(Q(w_t) - \hat{Q}_{\theta_t}^*) + c_2\epsilon^2,$$

*where* $c_1 = \frac{2L_\pi'^2}{(1-\gamma)^2\lambda'_{\min}}$ *and* $c_2 = \frac{2L_\pi'^2}{(1-\gamma)^2}$.

*Proof.* One can easily show that

$$(1 - \gamma)D_Q = \mathbb{E}_{(s,a)\sim(1-\gamma)\phi_{\theta_t}} \left[ \ln \pi(a|s, \theta_t)(Q(s, a, w_t) - Q^*_{\theta_t}(s, a)) \right].$$

As assumed in Assumption 2.4, $||\nabla \ln \pi(a|s, \theta_t)|| \le L'_\pi$. Hence,

$$(1 - \gamma)||D_Q|| \le L'_\pi \cdot \mathbb{E}_{(s,a)\sim(1-\gamma)\phi_{\theta_t}} \left[ |Q(s, a, w_t) - Q^*_{\theta_t}(s, a)| \right].$$

Using the facts that $(\mathbb{E}[X])^2 \le \mathbb{E}[X^2]$,

$$(1 - \gamma)^2||D_Q||^2 \le {L'_\pi}^2 \cdot \mathbb{E}_{(s,a)\sim(1-\gamma)\phi_{\theta_t}} \left[ |Q(s, a, w_t) - Q^*_{\theta_t}(s, a)|^2 \right].$$

On the other hand,

$$|Q(s, a, w_t) - Q^*_{\theta_t}(s, a)|^2 \le 2|Q(s, a, w_t) - \hat{Q}^*_{\theta_t}(s, a)|^2 + 2\epsilon^2.$$

where we use Assumption 2.5 which tells us $|\hat{Q}^*_{\theta_t}(s, a) - Q^*_{\theta_t}(s, a)| \le \epsilon$. Hence,

$$(1 - \gamma)^2||D_Q||^2 \le 2{L'_\pi}^2 \cdot \mathbb{E}_{(s,a)\sim(1-\gamma)\phi_{\theta_t}} \left[ |Q(s, a, w_t) - \hat{Q}^*_{\theta_t}(s, a)|^2 \right] + 2{L'_\pi}^2\epsilon^2.$$

Combine with Lemma C.3 and the fact that $(1 - \gamma)\phi_{\theta_t}(s, a) \le 1$,

$$(1 - \gamma)^2||D_Q||^2 \le \frac{2{L'_\pi}^2}{\lambda'_{\min}}\mathcal{N}_\theta \left( Q(w) - \hat{Q}^*_\theta \right) + 2{L'_\pi}^2\epsilon^2.$$

This finishes the proof.

$\square$

## C.3   Mean-value Theorem and Extensions

**Lemma C.5.** *These following lemmas generalize the mean-value theorem to higher dimensional input and output cases.*

*(a) Let $h : \mathbb{R} \to \mathbb{R}$ be any differentiable function. For any $x, y \in \mathbb{R}$, there exists $\lambda \in (0, 1)$ and $z = \lambda x + (1 - \lambda)y$ such that*

$$h(y) - h(x) = h'(z)(y - x).$$

*(b) Let $\xi : \mathbb{R}^a \to \mathbb{R}$ be any differentiable function. For any $x, y \in \mathbb{R}^a$, there exists $\lambda \in (0, 1)$ and $z = \lambda x + (1 - \lambda)y$ such that*

$$\xi(y) - \xi(x) = \xi'(z)(y - x).$$

*(c) Let $f : \mathbb{R}^a \to \mathbb{R}^b$ be any differentiable function and $e \in \mathbb{R}^b$ be any vector. For any $x, y \in \mathbb{R}^a$, there exists $\lambda \in (0, 1)$ and $z = \lambda x + (1 - \lambda)y$ such that*

$$e^T(f(y) - f(x)) = e^T f'(z)(y - x),$$

*where $f'(z)$ is the Jacobian at $z$.*

*Proof.* This proof is given by Lemma A.2 in Tian et al. (2023). $\square$

## C.4   The Mixing of Two Markov Chains

In this section, we argue that if two Markov chains satisfy Assumption 2.8, then the difference between their distributions could be very small. This is inspired by and follows the same logic as Chen & Zhao (2022); Zou et al. (2019). Before that, we will first introduce the total variation norm for vectors and matrices, which can be used to measure a difference between distributions.

Denote $f : X \to R$ to be any real value function. We can define the total variation norm of $f$, denoted by $||f||_{\mathrm{TV}}$, as

$$||f||_{\mathrm{TV}} = \sum_{x \in X} |f(x)|.$$

For matrix $A$, we can also define $||A||_{\text{TV}}$ to be

$$||A||_{\text{TV}} = \sup_{||f||_{\text{TV}}=1} ||f^T A||_{\text{TV}}.$$

If $f$ is some probability measure, since $f(x) \in [0,1]$, it is easy to conclude that $||f||_{\text{TV}} = 1$. Likewise, if $A$ is a Markov transition matrix, we can show that $||A||_{\text{TV}} = 1$.

The following lemma establishes the relationship between the total variation norm with the more familiar 1-norm and $\infty$-norm.

**Lemma C.6.** *The following statements are true:*

    *a. For any vector $f$, $||f||_{TV} = ||f||_1$.*

    *b. For any matrix $A$, $||A||_{TV} = ||A||_\infty$.*

*Proof.* The lemma is obvious so we omit the proof here. □

Based on Assumption 2.8, we have the following result:

**Lemma C.7.** *If the Markov chain has transition probability matrix $A$, then we have*

$$||A^t||_{\text{TV}} \le 1, \forall t.$$

*Further, if the Markov chain satisfies Assumption 2.8, then we have*

$$||A^t||_{\text{TV}} \le C\beta^t, \forall t \ge \tau_{\text{mix}}.$$

*Proof.* The first part of this lemma is obvious because $A$ is a stochastic (Markov) matrix and by Lemma C.6, $|| \cdot ||_{\text{TV}}$ is just the same as $|| \cdot ||_\infty$.

For the second part, by the definition of total variation norm,

$$||A||_{\text{TV}} = \sup_{||f||_{TV}=1} ||f^T A||_{\text{TV}} = \frac{1}{2} \sup_{i,j} ||(e_i - e_j)^T A||_{\text{TV}},$$

where $e_i$ means the all-zero vector except a 1 at the $i$'th entry. By Assumption 2.8,

$$||A^t||_{\text{TV}} \le \frac{1}{2} \sup_{i,j} \left( ||(e_i - \mu')^T A^t||_{\text{TV}} + ||(e_j - \mu')^T A^t||_{\text{TV}} \right) \le C\beta^t, \forall t \ge \tau_{\text{mix}}.$$

□

The following theorem, which is inspired by Theorem 3.1 in Mitrophanov (2005), is very important in many analyses of AC that take Markov sampling into consideration (i.e., Wu et al. (2020); Chen & Zhao (2022)). However, since our settings are slightly different, we provide our own version.

**Lemma C.8.** *Suppose we have the following two Markov Chains which of both satisfy Assumption 2.8,*

$$p_0^A \xrightarrow{A} p_1^A \xrightarrow{A} \dots \xrightarrow{A} p_t^A,$$
$$p_0^B \xrightarrow{B} p_1^B \xrightarrow{B} \dots \xrightarrow{B} p_t^B,$$

*where $p_i^A, p_i^B$ stand for the probability at step $i$ under transition matrix $A, B$, respectively. The following inequality holds:*

$$||p_t^A - p_t^B||_1 \le C\beta^t ||p_0^A - p_0^B||_1 + (\tau_{\text{mix}} + C\frac{1}{1-\beta})||A - B||_\infty, \forall t \ge \tau_{\text{mix}} + 1.$$

*Proof.* First, we prove that

$$p_t^{A^T} - p_t^{B^T} = \left(p_0^A - p_0^B\right)^T B^t + \sum_{i=0}^{t-1} p_0^{A^T} (A - B) B^{t-i-1}.$$

by induction. If $t = 1$, by definition we know

$$p_1^{A^T} - p_1^{B^T} = p_0^{A^T} A - p_0^{B^T} B = \left(p_0^A - p_0^B\right)^T B + p_0^{A^T}(A - B).$$

If the result holds for $t = k$, then when $t = k + 1$,

$$\begin{aligned}
p_{k+1}^{A}{}^T - p_{k+1}^{B}{}^T &= p_k^{A^T} A - p_k^{B^T} B \\
&= \left(p_k^A - p_k^B\right)^T B + p_k^{A^T}(A - B) \\
&= \left(p_0^A - p_0^B\right)^T B^{k+1} + \sum_{i=0}^{k-1} p_0^{A^T}(A - B) B^{k-i} + p_k^{A^T}(A - B) \\
&= \left(p_0^A - p_0^B\right)^T B^{k+1} + \sum_{i=0}^{k} p_0^{A^T}(A - B) B^{k-i},
\end{aligned}$$

where the third line is because we assume the result holds for the $t = k$ case. Now, we can take the total variation norm on both sides:

$$\begin{aligned}
\left\|p_t^A - p_t^B\right\|_{TV} &\leq \left\|\left(p_0^A - p_0^B\right)^T B^t\right\|_{TV} + \left\|\sum_{i=0}^{t-1} p_0^{A^T}(A - B) B^{t-i-1}\right\|_{TV} \\
&\leq \left\|p_0^A - p_0^B\right\|_{TV} \cdot \left\|B^t\right\|_{TV} + \left\|p_0^A\right\|_{TV} \cdot \|A - B\|_{TV} \cdot \sum_{i=0}^{t-1} \left\|B^{t-i-1}\right\|_{TV} \\
&\leq C\beta^t \left\|p_0^A - p_0^B\right\|_1 + \|A - B\|_\infty \left(\tau_{\text{mix}} + \sum_{i=\tau_{\text{mix}}}^{t-1} C\beta^i\right) \\
&= C\beta^t \left\|p_0^A - p_0^B\right\|_1 + \|A - B\|_\infty \left(\tau_{\text{mix}} + C\frac{\beta^{\tau_{\text{mix}}} - \beta^t}{1 - \beta}\right) \\
&\leq C\beta^t \|p_0^A - p_0^B\|_1 + (\tau_{\text{mix}} + C\frac{1}{1 - \beta})\|A - B\|_\infty,
\end{aligned}$$

where the third line utilizes Lemma C.6 and Lemma C.7. $\qquad \square$

With Lemma C.8, we can derive many useful results. The following result is similar to Lemma 3 in Zou et al. (2019), Lemma B.1 in Wu et al. (2020), and Lemma B.4 in Chen & Zhao (2022), which shows that both of $\mu'_\theta$, the stationary distribution over states, and $\mu_\theta$, the stationary distribution over state-action pairs, are Lipschitz with respect to $\theta$.

**Lemma C.9.** *The following statements hold:*

*a.*

$$||\mu'_{\theta_1} - \mu'_{\theta_2}||_1 \leq (\tau_{\text{mix}} + C\frac{1}{1 - \beta})n_a L_\pi ||\theta_1 - \theta_2||.$$

*b.*

$$||\mu_{\theta_1} - \mu_{\theta_2}||_1 \leq (1 + \tau_{\text{mix}} + C\frac{1}{1 - \beta})n_a L_\pi ||\theta_1 - \theta_2||,$$

*where $n_a$ is the number of actions. In other words, $n_a = |A|$.*

*Proof.* Recall the state transition matrix $P'_\theta$ is defined by $P_\theta(s'|s) = \sum_a P_{\text{env}}(s'|s, a)\pi(a|s, \theta)$ and corresponding stationary distribution $\mu'_\theta{}^T = \mu'_\theta{}^T P'_\theta$. Use Lemma C.9, we know that

$$||\mu'_{\theta_1} - \mu'_{\theta_2}||_1 \leq C\beta^t ||\mu'_{\theta_1} - \mu'_{\theta_2}||_1 + (\tau_{\text{mix}} + C\frac{1}{1 - \beta})||P'_{\theta_1} - P'_{\theta_2}||_\infty.$$

Notice that

$$\begin{aligned}
||P'_{\theta_1} - P'_{\theta_2}||_\infty &= \sup_s \sum_{s'} \sum_a P_{\text{env}}(s'|s, a) |\pi(a|s, \theta_1) - \pi(a|s, \theta_2)| \\
&\leq \sup_s \sum_{s'} \sum_a P_{\text{env}}(s'|s, a) L_\pi ||\theta_1 - \theta_2|| \\
&= n_a L_\pi ||\theta_1 - \theta_2||,
\end{aligned}$$

where we use Assumption 2.4 in the second line. Hence,

$$||\mu'_{\theta_1} - \mu'_{\theta_2}||_1 \leq C\beta^t ||\mu'_{\theta_1} - \mu'_{\theta_2}||_1 + (\tau_{\mathrm{mix}} + C\frac{1}{1-\beta})n_a L_\pi ||\theta_1 - \theta_2||.$$

Taking $\lim_{t\to\infty}$ on both sides, we derive

$$||\mu'_{\theta_1} - \mu'_{\theta_2}||_1 \leq (\tau_{\mathrm{mix}} + C\frac{1}{1-\beta})n_a L_\pi ||\theta_1 - \theta_2||.$$

This finishes the proof of the first part.

Now, Eq.(1) implies

$$
\begin{aligned}
||\mu_{\theta_1} - \mu_{\theta_2}||_1 &= \sum_{s,a} |\mu'_{\theta_1}(s)\pi(a|s,\theta_1) - \mu'_{\theta_2}(s)\pi(a|s,\theta_2)| \\
&\leq \sum_{s,a} \left[|\mu'_{\theta_1}(s)\pi(a|s,\theta_1) - \mu'_{\theta_1}(s)\pi(a|s,\theta_2)| + |\mu'_{\theta_1}(s)\pi(a|s,\theta_2) - \mu'_{\theta_2}(s)\pi(a|s,\theta_2)|\right] \\
&\leq \sum_{s,a} \mu'_{\theta_1}(s)L_\pi ||\theta_1 - \theta_2|| + ||\mu'_{\theta_1} - \mu'_{\theta_2}||_1 \\
&\leq (1 + \tau_{\mathrm{mix}} + C\frac{1}{1-\beta})n_a L_\pi ||\theta_1 - \theta_2||.
\end{aligned}
$$

$\square$

In order to show the following lemmas, we need to introduce the following auxiliary Markov chain:

$$s_{t-\tau_{\mathrm{mix}}-1} \xrightarrow{\pi(\theta_{t-\tau_{\mathrm{mix}}-1})} a_{t-\tau_{\mathrm{mix}}-1} \xrightarrow{P_{\mathrm{env}}} s_{t-\tau_{\mathrm{mix}}} \xrightarrow{\pi(\theta_{t-\tau_{\mathrm{mix}}})} a_{t-\tau_{\mathrm{mix}}} \xrightarrow{P_{\mathrm{env}}} s_{t-\tau_{\mathrm{mix}}+1} \xrightarrow{\pi(\theta_{t-\tau_{\mathrm{mix}}})}$$

$$\tilde{a}_{t-\tau_{\mathrm{mix}}+1} \xrightarrow{P_{\mathrm{env}}} \tilde{s}_{t-\tau_{\mathrm{mix}}+2} \xrightarrow{\pi(\theta_{t-\tau_{\mathrm{mix}}})} \tilde{a}_{t-\tau_{\mathrm{mix}}+2} \xrightarrow{P_{\mathrm{env}}} \ldots \xrightarrow{P_{\mathrm{env}}} \tilde{s}_t \xrightarrow{\pi(\theta_{t-\tau_{\mathrm{mix}}})} \tilde{a}_t \xrightarrow{P_{\mathrm{env}}} \tilde{s}_{t+1}.$$

For reference, the original Markov chain around $t$ is

$$s_{t-\tau_{\mathrm{mix}}-1} \xrightarrow{\pi(\theta_{t-\tau_{\mathrm{mix}}-1})} a_{t-\tau_{\mathrm{mix}}-1} \xrightarrow{P_{\mathrm{env}}} s_{t-\tau_{\mathrm{mix}}} \xrightarrow{\pi(\theta_{t-\tau_{\mathrm{mix}}})} a_{t-\tau_{\mathrm{mix}}} \xrightarrow{P_{\mathrm{env}}} s_{t-\tau_{\mathrm{mix}}+1} \xrightarrow{\pi(\theta_{t-\tau_{\mathrm{mix}}+1})}$$

$$a_{t-\tau_{\mathrm{mix}}+1} \xrightarrow{P_{\mathrm{env}}} s_{t-\tau_{\mathrm{mix}}+2} \xrightarrow{\pi(\theta_{t-\tau_{\mathrm{mix}}+2})} a_{t-\tau_{\mathrm{mix}}+2} \xrightarrow{P_{\mathrm{env}}} \ldots \xrightarrow{P_{\mathrm{env}}} s_t \xrightarrow{\pi(\theta_t)} a_t \xrightarrow{P_{\mathrm{env}}} s_{t+1}.$$

For the consistency of notations, we will denote $O_\tau = (s_\tau, a_\tau, s'_\tau, a'_\tau)$ and $\tilde{O}_\tau = (\tilde{s}_\tau, \tilde{a}_\tau, \tilde{s}'_\tau, \tilde{a}'_\tau)$ where in this case we have $s'_\tau = s_{\tau+1}$, $\tilde{s}'_\tau = \tilde{s}_{\tau+1}$ and $a'_\tau \sim \pi(a|s, \theta_\tau)$, $\tilde{a}'_\tau \sim \pi(a|s, \theta_{t-\tau_{\mathrm{mix}}})$. This kind of notations will immediately implies $P(\tilde{O}_{t-\tau_{\mathrm{mix}}-1} \in \cdot) = P(O_{t-\tau_{\mathrm{mix}}-1} \in \cdot)$.

The following lemma claims that the distribution difference between the two Markov chains above will be very small.

**Lemma C.10.** *The following statements are true:*

a. *For any possible $\tau \in \{t - \tau_{\mathrm{mix}}, t - \tau_{\mathrm{mix}} + 1, \ldots, t\}$,*

$$||P(S_{\tau+1} \in \cdot) - P(\tilde{S}_{\tau+1} \in \cdot)||_1 \leq ||P(O_\tau \in \cdot) - P(\tilde{O}_\tau \in \cdot)||_1.$$

b. *For any possible $\tau \in \{t - \tau_{\mathrm{mix}}, t - \tau_{\mathrm{mix}} + 1, \ldots, t\}$,*

$$||P(O_\tau \in \cdot) - P(\tilde{O}_\tau \in \cdot)||_1 \leq 2n_a L_\pi \mathbb{E}\left[||\theta_\tau - \theta_{t-\tau_{\mathrm{mix}}}||\right] + ||P(S_\tau \in \cdot) - P(\tilde{S}_\tau \in \cdot)||_1.$$

c. *Consider $P(O_t \in \cdot)$ and $P(\tilde{O}_t \in \cdot)$,*

$$||P(O_t \in \cdot) - P(\tilde{O}_t \in \cdot)||_1 \leq 2n_a L_\pi \sum_{i=t-\tau_{\mathrm{mix}}}^{t} \mathbb{E}\left[||\theta_i - \theta_{t-\tau_{\mathrm{mix}}}||\right].$$

*Proof.* For the first part,

$$||P(S_{\tau+1} \in \cdot) - P(\tilde{S}_{\tau+1} \in \cdot)||_1$$

$$= \sum_{s'} \left| P(S_{\tau+1} = s') - P(\tilde{S}_{\tau+1} = s') \right|$$

$$\leq \sum_{s,a,s',a'} \left| P(S_\tau = s, A_\tau = a, S_{\tau+1} = s' A_{\tau+1} = a') - P(\tilde{S}_\tau = s, \tilde{A}_\tau = a, \tilde{S}_{\tau+1} = s', \tilde{A}_{\tau+1} = a') \right|$$

$$= ||P(O_\tau \in \cdot) - P(\tilde{O}_\tau \in \cdot)||_1.$$

For the second part, conditioned on $\theta_{t-\tau_{\text{mix}}}$ and $s_{t-\tau_{\text{mix}}+1}$, we denote

$$M_1 = P(S_\tau = s)P(\theta_\tau = z_\tau|S_\tau = s)\pi(a|s, z_\tau)P_{\text{env}}(s'|s, a)\pi(a'|s', z_\tau)$$

$$M_2 = P(S_\tau = s)P(\theta_\tau = z_\tau|S_\tau = s)\pi(a|s, z_\tau)P_{\text{env}}(s'|s, a)\pi(a'|s', \theta_{t-\tau_{\text{mix}}})$$

$$M_3 = P(S_\tau = s)P(\theta_\tau = z_\tau|S_\tau = s)\pi(a|s, \theta_{t-\tau_{\text{mix}}})P_{\text{env}}(s'|s, a)\pi(a'|s', \theta_{t-\tau_{\text{mix}}})$$

$$M_4 = P(\tilde{S}_\tau = s)P(\theta_\tau = z_\tau|S_\tau = s)\pi(a|s, \theta_{t-\tau_{\text{mix}}})P_{\text{env}}(s'|s, a)\pi(a'|s', \theta_{t-\tau_{\text{mix}}}),$$

which will be useful later. Using notations from the original Markov Chain,

$$P(S_\tau = s, A_\tau = a, S_{\tau+1} = s', A_{\tau+1} = a')$$

$$= \int P(S_\tau = s, A_\tau = a, S_{\tau+1} = s', A_{\tau+1} = a', \theta_\tau = z_\tau)dz_\tau$$

$$= \int P(S_\tau = s)P(\theta_\tau = z_\tau|S_\tau = s)\pi(a|s, z_\tau)P_{\text{env}}(s'|s, a)\pi(a'|s', z_\tau)dz_\tau$$

$$= \int M_1 dz_\tau.$$

Similarly, the auxiliary Markov Chain gives us

$$P(\tilde{S}_\tau = s, \tilde{A}_\tau = a, \tilde{S}_{\tau+1} = s', \tilde{A}_{\tau+1} = a')$$

$$= P(\tilde{S}_\tau = s)\pi(a|s, \theta_{t-\tau_{\text{mix}}})P_{\text{env}}(s'|s, a)\pi(a'|s', \theta_{t-\tau_{\text{mix}}})$$

$$= \int P(\tilde{S}_\tau = s)P(\theta_\tau = z_\tau|S_\tau = s)\pi(a|s, \theta_{t-\tau_{\text{mix}}})P_{\text{env}}(s'|s, a)\pi(a'|s', \theta_{t-\tau_{\text{mix}}})dz_\tau$$

$$= \int M_4 dz_\tau.$$

We now rephrase the left-hand side term we want to prove,

$$||P(O_\tau \in \cdot) - P(\tilde{O}_\tau \in \cdot)||_1$$

$$\leq \sum_{s,a,s',a'} \left| P(S_\tau = s, A_\tau = a, S_{\tau+1} = s', A_{\tau+1} = a') - P(\tilde{S}_\tau = s, \tilde{A}_\tau = a, \tilde{S}_{\tau+1} = s', \tilde{A}_{\tau+1} = a') \right|$$

$$\leq \sum_{s,a,s',a'} \int |M_1 - M_4|dz_\tau$$

$$\leq \underbrace{\sum_{s,a,s',a'} \int |M_1 - M_2|dz_\tau}_{I_1} + \underbrace{\sum_{s,a,s',a'} \int |M_2 - M_3|dz_\tau}_{I_2} + \underbrace{\sum_{s,a,s',a'} \int |M_3 - M_4|dz_\tau}_{I_3}.$$

For $I_1$,

$$I_1 = \sum_{s,a,s',a'} \int P(S_\tau = s)P(\theta_\tau = z_\tau|S_\tau = s)\pi(a|s, z_\tau)P_{\text{env}}(s'|s, a)$$

$$\cdot |\pi(a'|s', z_\tau) - \pi(a'|s', \theta_{t-\tau_{\text{mix}}})| dz_\tau$$

$$\leq \sum_{s,a,s',a'} \int P(S_\tau = s)P(\theta_\tau = z_\tau|S_\tau = s)\pi(a|s, z_\tau)P_{\text{env}}(s'|s, a)L_\pi \|z_\tau - \theta_{t-\tau_{\text{mix}}}\| dz_\tau$$

$$= n_a L_\pi \sum_s \int P(S_\tau = s)P(\theta_\tau = z_\tau|S_\tau = s) \|z_\tau - \theta_{t-\tau_{\text{mix}}}\| dz_\tau$$

$$= n_a L_\pi \mathbb{E} \left[ \|\theta_\tau - \theta_{t-\tau_{\text{mix}}}\| \right].$$

For $I_2$,

$$I_2 = \sum_{s,a,s',a'} \int P(S_\tau = s) P(\theta_\tau = z_\tau | S_\tau = s) \left[ \pi(a|s, z_\tau) - \pi(a|s, \theta_{t-\tau_{\mathrm{mix}}}) \right] P_{\mathrm{env}}(s'|s, a)$$

$$\cdot \pi(a'|s', \theta_{t-\tau_{\mathrm{mix}}}) dz_\tau$$

$$\leq \sum_{s,a,s',a'} \int P(S_\tau = s) P(\theta_\tau = z_\tau | S_\tau = s) L_\pi \left\| z_\tau - \theta_{t-\tau_{\mathrm{mix}}} \right\| P_{\mathrm{env}}(s'|s, a) \pi(a'|s', \theta_{t-\tau_{\mathrm{mix}}}) dz_\tau$$

$$= n_a L_\pi \sum_s \int P(S_\tau = s) P(\theta_\tau = z_\tau | S_\tau = s) \left\| z_\tau - \theta_{\tau-\tau_{\mathrm{mix}}} \right\| dz_\tau$$

$$= n_a L_\pi \mathbb{E} \left[ \left\| \theta_\tau - \theta_{t-\tau_{\mathrm{mix}}} \right\| \right].$$

For $I_3$,

$$I_3 = \sum_{s,a,s',a'} \int \left| P(S_\tau = s) - P(\tilde{S}_\tau = s) \right| P(\theta_\tau = z_\tau | S_\tau = s) \pi(a|s, \theta_{t-\tau_{\mathrm{mix}}}) P_{\mathrm{env}}(s'|s, a)$$

$$\cdot \pi(a'|s', \theta_{t-\tau_{\mathrm{mix}}}) dz_\tau$$

$$= \sum_s \left| P(S_\tau = s) - P(\tilde{S}_\tau = s) \right|$$

$$= ||P(S_\tau \in \cdot) - P(\tilde{S}_\tau \in \cdot)||_1.$$

Hence,

$$||P(O_\tau \in \cdot) - P(\tilde{O}_\tau \in \cdot)||_1 \leq 2 n_a L_\pi \mathbb{E} \left[ \left\| \theta_\tau - \theta_{t-\tau_{\mathrm{mix}}} \right\| \right] + ||P(S_\tau \in \cdot) - P(\tilde{S}_\tau \in \cdot)||_1,$$

which finishes the proof for part *b*.

It is easy to check that part *a* implies the following:

$$||P(S_\tau \in \cdot) - P(\tilde{S}_\tau \in \cdot)||_1 \leq ||P(O_{\tau-1} \in \cdot) - P(\tilde{O}_{\tau-1} \in \cdot)||_1.$$

The above fact, along with the result from part *b*, tells us the following:

$$||P(O_\tau \in \cdot) - P(\tilde{O}_\tau \in \cdot)||_1 \leq 2 n_a L_\pi \mathbb{E} \left[ \left\| \theta_\tau - \theta_{t-\tau_{\mathrm{mix}}} \right\| \right] + ||P(O_{\tau-1} \in \cdot) - P(\tilde{O}_{\tau-1} \in \cdot)||_1.$$

Repeat the inequality above over $t$ to $t - \tau_{\mathrm{mix}}$ we have

$$||P(O_t \in \cdot) - P(\tilde{O}_t \in \cdot)||_1 \leq 2 n_a L_\pi \sum_{i=t-\tau_{\mathrm{mix}}}^{t} \mathbb{E} \left[ \left\| \theta_i - \theta_{t-\tau_{\mathrm{mix}}} \right\| \right].$$

$\square$

## C.5 Smoothness of the State-Value Function

The following two lemmas show that the state value function is actually smooth with respect to $\theta$. The idea here is inspired by Olshevsky & Gharesifard (2022). The first lemma requires the following basic identity from Zwillinger (2018): for matrix $A_\theta$,

$$\frac{\partial \left( A_\theta^{-1} \right)}{\partial \theta_i} = -A_\theta^{-1} \frac{\partial A_\theta}{\partial \theta_i} A_\theta^{-1}.$$

**Lemma C.11.** *For two vectors $u, v \in R^n$ whose entries are bounded. Suppose*

$$q_\theta = u^T (I - \gamma P_\theta)^{-1} v.$$

*Then, there exists a constant $L_q$ such that*

$$||\nabla q_\theta|| \leq L_q.$$

*Similarly, if*

$$q'_\theta = u^T (I - \gamma P'_\theta)^{-1} v.$$

*Then, there exists a constant $L'_q$ such that*

$$||\nabla q'_\theta|| \leq L'_q.$$

*Proof.* The proof can be found in Olshevsky & Gharesifard (2022). $\qquad\square$

The following lemma, which is our goal in this section, claims that $V_\theta^*$ is smooth with respect to $\theta$.

**Lemma C.12.** $V_{\theta_t}^*$ *is $H_V$-smoothness with respect to $\theta_t$.*

*Proof.* Using Theorem 2.1, we obtain the following result:

$$\left\| \nabla V_{\theta_1}^* - \nabla V_{\theta_2}^* \right\|$$

$$= \left\| \sum_{(s,a)\in S\times A} \phi_{\theta_1}(s,a)Q_{\theta_1}^*(s,a)\nabla\ln\pi(a|s,\theta_1) - \sum_{(s,a)\in S\times A} \phi_{\theta_2}(s,a)Q_{\theta_2}^*(s,a)\nabla\ln\pi(a|s,\theta_2) \right\|$$

$$\leq \left\| \sum_{(s,a)\in S\times A} \phi_{\theta_1}(s,a)Q_{\theta_1}^*(s,a)\nabla\ln\pi(a|s,\theta_1) - \sum_{(s,a)\in S\times A} \phi_{\theta_1}(s,a)Q_{\theta_1}^*(s,a)\nabla\ln\pi(a|s,\theta_2) \right\|$$

$$+ \left\| \sum_{(s,a)\in S\times A} \phi_{\theta_1}(s,a)Q_{\theta_1}^*(s,a)\nabla\ln\pi(a|s,\theta_2) - \sum_{(s,a)\in S\times A} \phi_{\theta_1}(s,a)Q_{\theta_2}^*(s,a)\nabla\ln\pi(a|s,\theta_2) \right\|$$

$$+ \left\| \sum_{(s,a)\in S\times A} \phi_{\theta_1}(s,a)Q_{\theta_2}^*(s,a)\nabla\ln\pi(a|s,\theta_2) - \sum_{(s,a)\in S\times A} \phi_{\theta_2}(s,a)Q_{\theta_2}^*(s,a)\nabla\ln\pi(a|s,\theta_2) \right\|$$

$$\leq \underbrace{\sum_{(s,a)\in S\times A} \phi_{\theta_1}(s,a)Q_{\theta_1}^*(s,a)\left\|\nabla\ln\pi(a|s,\theta_1) - \nabla\ln\pi(a|s,\theta_2)\right\|}_{I_1}$$

$$+ \underbrace{\sum_{(s,a)\in S\times A} \phi_{\theta_1}(s,a)\left|Q_{\theta_1}^*(s,a) - Q_{\theta_2}^*(s,a)\right|\nabla\ln\pi(a|s,\theta_2)}_{I_2}$$

$$+ \underbrace{\sum_{(s,a)\in S\times A} \left|\phi_{\theta_1}(s,a) - \phi_{\theta_2}(s,a)\right|Q_{\theta_2}^*(s,a)\nabla\ln\pi(a|s,\theta_2)}_{I_3}.$$

We now show that all $I_1, I_2, I_3$ can be bounded by a multiple of $||\theta_1 - \theta_2||$.

For $I_1$, by Assumption 2.4 we know that $\nabla\ln\pi(a|s,\theta)$ is Lipschitz, which, together with $\phi_\theta(s,a) \leq \frac{1}{1-\gamma}$ and $Q_\theta^*(s,a) \leq \frac{r_{\max}}{1-\gamma}$, implies that $I_1$ can be upper bounded by a multiple of $||\theta_1 - \theta_2||$.

For $I_2$, since $Q_\theta^*$ satisfies Bellman equation, we can write $Q_\theta^*$ using matrix multiplication, which is

$$Q_\theta^* = (1 - \gamma P_\theta)^{-1}R.$$

By Lemma C.11, this implies

$$Q_\theta^*(s,a) = e_{s,a}^T(1 - \gamma P_\theta)^{-1}R,$$

where $e_{s,a}$ has only one non-zero entry of one corresponding to the pair $(s,a)$. Hence, $Q_\theta^*(s,a)$ is Lipschitz with respect to $\theta$.

For $I_3$, by definition,

$$P_\theta(S_t = s|S_0 = s_0) = e_{s_0}^T P_\theta'^t e_s.$$

Thus,

$$\phi_\theta(s,a) = \phi_\theta(s)\pi(a|s,\theta) = e_{s_0}^T(I - \gamma P_\theta')^{-1}e_s\pi(a|s,\theta),$$

where $e_s$ has only one non-zero entry of one corresponding to $s$. Again, by Lemma C.11, $\phi_\theta(s,a)$ is Lipschitz with respect to $\theta$.

$\qquad\square$

## C.6  Properties of the Actor Update

The following lemma shows that the incremental in actor update is bounded.

**Lemma C.13.** *For $g(\hat{O}_t, w_t, \theta_t)$ and $\bar{g}(w_t, \theta_t)$, we have the following properties:*

a. *For $g(\hat{O}_t, w_t, \theta_t)$,*

$$||g(\hat{O}_t, w_t, \theta_t)|| \leq U_g$$

*where $U_g = 2L'_\pi L_Q \sigma_w + L'_\pi \epsilon + L'_\pi \frac{r_{\max}}{1-\gamma}$.*

b. *For $\bar{g}(w_t, \theta_t)$,*

$$||\bar{g}(w_t, \theta_t)|| \leq \frac{1}{1-\gamma} U_g$$

*where $U_g = 2L'_\pi L_Q \sigma_w + L'_\pi \epsilon + L'_\pi \frac{r_{\max}}{1-\gamma}$.*

*Proof.* Recall that in Eq.(15), $g(\hat{O}_t, w_t, \theta_t)$ is defined to be

$$g(\hat{O}_t, w_t, \theta_t) = Q(\hat{s}_t, \hat{a}_t, w_t) \nabla_\theta \ln \pi(\hat{a}_t | \hat{s}_t, \theta_t).$$

To bound $||g(\hat{O}_t, w_t, \theta_t)||$, by Assumption 2.4 and Lemma C.1, we can do the following manipulations

$$
\begin{aligned}
||g(\hat{O}_t, w_t, \theta_t)|| &= \left\| \nabla_\theta \ln \pi(\hat{a}_t | \hat{s}_t, \theta_t) \left( Q(\hat{s}_t, \hat{a}_t, w_t) - Q(\hat{s}_t, \hat{a}_t, \hat{w}^*_{\theta_t}) + Q(\hat{s}_t, \hat{a}_t, \hat{w}^*_{\theta_t}) \right) \right\| \\
&\leq \left\| \nabla_\theta \ln \pi(\hat{a}_t | \hat{s}_t, \theta_t) \left( Q(\hat{s}_t, \hat{a}_t, w_t) - Q(\hat{s}_t, \hat{a}_t, \hat{w}^*_{\theta_t}) \right) \right\| \\
&\quad + \left\| \nabla_\theta \ln \pi(\hat{a}_t | \hat{s}_t, \theta_t) Q(\hat{s}_t, \hat{a}_t, \hat{w}^*_{\theta_t}) \right\| \\
&\leq L'_\pi L_Q ||w_t - \hat{w}^*_{\theta_t}|| + \left\| \nabla_\theta \ln \pi(\hat{a}_t | \hat{s}_t, \theta_t) Q(\hat{s}_t, \hat{a}_t, \hat{w}^*_{\theta_t}) \right\| \\
&\leq 2L'_\pi L_Q \sigma_w + \left\| \nabla_\theta \ln \pi(\hat{a}_t | \hat{s}_t, \theta_t) \left( Q(\hat{s}_t, \hat{a}_t, \hat{w}^*_{\theta_t}) - Q^*(\hat{s}_t, \hat{a}_t) + Q^*(\hat{s}_t, \hat{a}_t) \right) \right\| \\
&= 2L'_\pi L_Q \sigma_w + \left\| \nabla_\theta \ln \pi(\hat{a}_t | \hat{s}_t, \theta_t) \left( Q(\hat{s}_t, \hat{a}_t, \hat{w}^*_{\theta_t}) - Q^*(\hat{s}_t, \hat{a}_t) \right) \right\| \\
&\quad + \left\| \nabla_\theta \ln \pi(\hat{a}_t | \hat{s}_t, \theta_t) Q^*(\hat{s}_t, \hat{a}_t) \right\| \\
&\leq 2L'_\pi L_Q \sigma_w + L'_\pi \epsilon + L'_\pi \frac{r_{\max}}{1-\gamma}.
\end{aligned}
$$

Because Eq.(16) implies that $\bar{g}(w_t, \theta_t)$ is some expectation of $g(\hat{O}_t, w_t, \theta_t)$ with a coefficient $\frac{1}{1-\gamma}$, the second part of this lemma is a direct result of the first part in Lemma C.13. $\qquad\square$

## C.7  Properties of the Critic Update

In this section, we will introduce some properties that we find useful in analyzing critic update.

Notice that, as defined in Eq.(8), $\delta_t$ actually depends on $O_t$ and $w_t$. Here we make this dependency explicitly and thus write $\delta_t$ as $\delta_t = \delta(O_t, w_t)$. In this following lemma, we explore some properties of $\delta_t$.

**Lemma C.14.** *For $\delta_t$, we have the following two results:*

a. *$\delta(O_t, w)$ is $L_\delta$-Lipschitz with respect to $w$,*

$$|\delta(O_t, w_1) - \delta(O_t, w_2)| \leq L_\delta ||w_1 - w_2||,$$

*where $L_\delta = (1+\gamma)L_Q$.*

b. *$|\delta(O_t, w)|$ is upper bounded by $U_\delta$,*

$$|\delta(O_t, w)| \leq U_\delta,$$

*where $U_\delta = 2(1+\gamma)L_Q \sigma_w + (1+\gamma)\epsilon + \frac{2}{1-\gamma} r_{\max}$.*

*Proof.* Using Eq.(8),

$$|\delta(O_t, w_1) - \delta(O_t, w_2)|$$
$$= |(r(s_t, a_t) + \gamma Q(s_t', a_t', w_1) - Q(s_t, a_t, w_1)) - (r(s_t, a_t) + \gamma Q(s_t', a_t', w_2) - Q(s_t, a_t, w_2))|$$
$$= |\gamma \left(Q(s_t', a_t', w_1) - Q(s_t', a_t', w_2)\right) - \left(Q(s_t, a_t, w_1) - Q(s_t, a_t, w_2)\right)|$$
$$\leq (1 + \gamma) L_Q ||w_1 - w_2||,$$

where the last line uses Lemma C.1. On the other hand,

$$|\delta(O_t, w)| = |r(s_t, a_t) + \gamma Q(s_t', a_t', w) - Q(s_t, a_t, w)|$$
$$= \left| r(s_t, a_t) + \gamma \left(Q(s_t', a_t', w) - \hat{Q}_\theta^*(s_t', a_t')\right) - \left(Q(s_t, a_t, w) - \hat{Q}_\theta^*(s_t, a_t)\right) \right.$$
$$\left. + \gamma \left(\hat{Q}_\theta^*(s_t', a_t') - Q_\theta^*(s_t', a_t')\right) - \left(\hat{Q}_\theta^*(s_t, a_t) - Q_\theta^*(s_t, a_t)\right) + \gamma Q_\theta^*(s_t', a_t') - Q_\theta^*(s_t, a_t) \right|$$
$$\leq r_{\max} + (1 + \gamma) L_Q ||w - \hat{w}_\theta^*|| + (1 + \gamma)\epsilon + \frac{1 + \gamma}{1 - \gamma} r_{\max}$$
$$\leq 2(1 + \gamma) L_Q \sigma_w + (1 + \gamma)\epsilon + \frac{2}{1 - \gamma} r_{\max},$$

where the second equation uses Assumption 2.1, Lemma C.1, Eq.(9) and Eq.(5). □

With the above properties of $\delta_t$, now we can further consider $f$, $\bar{f}$, and $F$ defined in Lemma C.17.

**Lemma C.15.** *For $f(O_t, w)$ defined in Eq.(15), we have the following two results:*

    *a.  $f(O_t, w)$ is $L_f$-Lipschitz with respect to $w$,*

$$||f(O_t, w_1) - f(O_t, w_2)|| \leq L_f ||w_1 - w_2||,$$

    *where $L_f = (1 + \gamma)L_Q^2 + H_Q U_\delta$.*

    *b.  $||f(O_t, w)||$ can be upper bounded by $U_f$,*

$$||f(O_t, w)|| \leq U_f,$$

    *where $U_f = L_Q U_\delta$.*

*Proof.* By definition,

$$f(O_t, w) = \delta(O_t, w)\nabla Q(s_t, a_t, w).$$

Hence,

$$f(O_t, w_1) - f(O_t, w_2)$$
$$= \delta(O_t, w_1)\nabla Q(s_t, a_t, w_1) - \delta(O_t, w_2)\nabla Q(s_t, a_t, w_2)$$
$$= \delta(O_t, w_1)\nabla Q(s_t, a_t, w_1) - \delta(O_t, w_2)\nabla Q(s_t, a_t, w_1)$$
$$\quad + \delta(O_t, w_2)\nabla Q(s_t, a_t, w_1) - \delta(O_t, w_2)\nabla Q(s_t, a_t, w_2)$$
$$= \underbrace{[\delta(O_t, w_1) - \delta(O_t, w_2)]\nabla Q(s_t, a_t, w_1)}_{I_1} + \underbrace{\delta(O_t, w_2)\left(\nabla Q(s_t, a_t, w_1) - \nabla Q(s_t, a_t, w_2)\right)}_{I_2}.$$

For $I_1$, by Lemma C.14 and Lemma C.1, we perform the following manipulations:

$$||[\delta(O_t, w_1) - \delta(O_t, w_2)]\nabla Q(s_t, a_t, w_1)|| \leq L_\delta L_Q ||w_1 - w_2||$$
$$= (1 + \gamma)L_Q^2 ||w_1 - w_2||.$$

For $I_2$, by Lemma C.14 and Lemma C.1, we derive

$$||\delta(O_t, w_2)\left(\nabla Q(s_t, a_t, w_1) - \nabla Q(s_t, a_t, w_2)\right)|| \leq U_\delta H_Q ||w_1 - w_2||.$$

Combining $I_1$ and $I_2$ we prove the first part of this lemma.

For the second part, by Lemma C.14 and Lemma C.1,

$$||f(O_t, w)|| = ||\delta(O_t, w)\nabla Q(s_t, a_t, w)|| \leq U_\delta L_Q.$$

□

**Lemma C.16.** *For $\bar{f}(w, \theta)$ defined in Eq.(16), we have the following results:*

    a. $||\bar{f}(w, \theta)||$ *can be upper bounded by $U_f$,*

$$||\bar{f}(w, \theta)|| \leq U_f,$$

    *where $U_f = L_Q U_\delta$.*

    b. $\bar{f}(w, \theta)$ *is $L_f$-Lipschitz with respect to $w$,*

$$||\bar{f}(w_1, \theta) - \bar{f}(w_2, \theta)|| \leq L_{\bar{f}}||w_1 - w_2||,$$

    *where $L_f = (1 + \gamma)L_Q^2 + H_Q U_\delta$.*

    c. $\bar{f}(w, \theta)$ *is $L_{\bar{f}}$-Lipschitz with respect to $\theta$,*

$$||\bar{f}(w, \theta_1) - \bar{f}(w, \theta_2)|| \leq L_{\bar{f}}||\theta_1 - \theta_2||,$$

    *where $L_{\bar{f}} = L_Q U_\delta (2 + \tau_{\mathrm{mix}} + C \frac{1}{1-\beta}) n_a L_\pi$.*

*Proof.* Because of Eq.(16), the first and second part of this lemma is a direct result of Lemma C.15. For the third part,

$$\left\| \bar{f}(w, \theta_1) - \bar{f}(w, \theta_2) \right\|$$

$$= \left\| \sum_{s,a} \mu_{\theta_1}(s, a) \sum_{s',a'} P_{\mathrm{env}}(s'|s, a) \pi(a'|s', \theta_1) f(O, w) \right.$$

$$\left. - \sum_{s,a} \mu_{\theta_2}(s, a) \sum_{s',a'} P_{\mathrm{env}}(s'|s, a) \pi(a'|s', \theta_2) f(O, w) \right\|$$

$$= \left\| \sum_{s,a} \sum_{s',a'} \left[ \mu_{\theta_1}(s, a) P_{\mathrm{env}}(s'|s, a) \pi(a'|s', \theta_1) - \mu_{\theta_2}(s, a) P_{\mathrm{env}}(s'|s, a) \pi(a'|s', \theta_2) \right] f(O, w) \right\|$$

$$\leq \sum_{s,a,s',a'} P_{\mathrm{env}}(s'|s, a) |\mu_{\theta_1}(s, a) \pi(a'|s', \theta_1) - \mu_{\theta_2}(s, a) \pi(a'|s', \theta_2)| \cdot ||f(O, w)||$$

$$\leq U_f \sum_{s,a,s',a'} P_{\mathrm{env}}(s'|s, a) |\mu_{\theta_1}(s, a) \pi(a'|s', \theta_1) - \mu_{\theta_2}(s, a) \pi(a'|s', \theta_2)|,$$

where the last line is by Lemma C.15. Further notice that

$$|\mu_{\theta_1}(s, a) \pi(a'|s', \theta_1) - \mu_{\theta_2}(s, a) \pi(a'|s', \theta_2)|$$
$$\leq |\mu_{\theta_1}(s, a) \pi(a'|s', \theta_1) - \mu_{\theta_1}(s, a) \pi(a'|s', \theta_2)|$$
$$+ |\mu_{\theta_1}(s, a) \pi(a'|s', \theta_2) - \mu_{\theta_2}(s, a) \pi(a'|s', \theta_2)|$$
$$= \mu_{\theta_1}(s, a) |\pi(a'|s', \theta_1) - \pi(a'|s', \theta_2)| + \pi(a'|s', \theta_2) |\mu_{\theta_1}(s, a) - \mu_{\theta_2}(s, a)|.$$

Hence,

$$\sum_{s,a,s',a'} P_{\mathrm{env}}(s'|s,a)\,|\mu_{\theta_1}(s,a)\pi(a'|s',\theta_1) - \mu_{\theta_2}(s,a)\pi(a'|s',\theta_2)|$$

$$\leq \sum_{s,a,s',a'} P_{\mathrm{env}}(s'|s,a)\mu_{\theta_1}(s,a)\,|\pi(a'|s',\theta_1) - \pi(a'|s',\theta_2)|$$

$$+ \sum_{s,a,s',a'} P_{\mathrm{env}}(s'|s,a)\pi(a'|s',\theta_2)\,|\mu_{\theta_1}(s,a) - \mu_{\theta_2}(s,a)|$$

$$\leq L_\pi \|\theta_1 - \theta_2\| \sum_{s,a,s',a'} P_{\mathrm{env}}(s'|s,a)\mu_{\theta_1}(s,a)$$

$$+ \sum_{s,a,s',a'} P_{\mathrm{env}}(s'|s,a)\pi(a'|s',\theta_2)\,|\mu_{\theta_1}(s,a) - \mu_{\theta_2}(s,a)|$$

$$= n_a L_\pi \|\theta_1 - \theta_2\| + \|\mu_{\theta_1} - \mu_{\theta_2}\|_1$$

$$\leq (2 + \tau_{\mathrm{mix}} + C\frac{1}{1-\beta})n_a L_\pi \|\theta_1 - \theta_2\|,$$

where the last line is by Lemma C.9. This implies

$$\left\|\bar{f}(w,\theta_1) - \bar{f}(w,\theta_2)\right\| \leq U_f(2 + \tau_{\mathrm{mix}} + C\frac{1}{1-\beta})n_a L_\pi \|\theta_1 - \theta_2\|$$

$$= L_Q U_\delta (2 + \tau_{\mathrm{mix}} + C\frac{1}{1-\beta})n_a L_\pi \|\theta_1 - \theta_2\|.$$

$\square$

**Lemma C.17.** *Denote $F(O,w,\theta) = (w - \hat{w}_\theta^*)^T \left[f(O,w) - \bar{f}(w,\theta)\right]$. The following results hold:*

a. *$F(O,w,\theta)$ is $L_{F_\theta}$-Lipschitz with respect to $\theta$,*
$$|F(O,w,\theta_1) - F(O,w,\theta_2)| \leq L_{F_\theta}\|\theta_1 - \theta_2\|,$$
*where*
$$L_{F_\theta} = 2\left(U_f L_w + \sigma_w L_{\bar{f}}\right).$$

b. *$F(O,w,\theta)$ is $L_{F_w}$-Lipschitz with respect to $w$,*
$$|F(O,w_1,\theta) - F(O,w_2,\theta)| \leq L_{F_w}\|w_1 - w_2\|,$$
*where*
$$L_{F_w} = 4\sigma_w L_f + 2U_f.$$

c. *Conditioned on $\theta_{t-\tau_{\mathrm{mix}}}$ and $s_{t-\tau_{\mathrm{mix}}+1}$,*
$$\left|\mathbb{E}\left[F(O_t, w_{t-\tau_{\mathrm{mix}}}, \theta_{t-\tau_{\mathrm{mix}}}) - F(\tilde{O}_t, w_{t-\tau_{\mathrm{mix}}}, \theta_{t-\tau_{\mathrm{mix}}})|\theta_{t-\tau_{\mathrm{mix}}}, s_{t-\tau_{\mathrm{mix}}+1}\right]\right|$$
$$\leq 4\sigma_w U_f n_a L_\pi \sum_{i=t-\tau_{\mathrm{mix}}}^{t} \mathbb{E}\left[\|\theta_i - \theta_{t-\tau_{\mathrm{mix}}}\|\,|\theta_{t-\tau_{\mathrm{mix}}}, s_{t-\tau_{\mathrm{mix}}+1}\right].$$

d. *Conditioned on $\theta_{t-\tau_{\mathrm{mix}}}$ and $s_{t-\tau_{\mathrm{mix}}+1}$,*
$$\left|\mathbb{E}\left[F(\tilde{O}_t, w_{t-\tau_{\mathrm{mix}}}, \theta_{t-\tau_{\mathrm{mix}}})|\theta_{t-\tau_{\mathrm{mix}}}, s_{t-\tau_{\mathrm{mix}}+1}\right]\right| \leq 2\sigma_w U_f C\beta^{\tau_{\mathrm{mix}}-1}.$$

*Proof.* First, we observe that

$$|F(O,w,\theta_1) - F(O,w,\theta_2)|$$
$$= \left|(w - \hat{w}_{\theta_1}^*)^T\left[f(O,w) - \bar{f}(w,\theta_1)\right] - (w - \hat{w}_{\theta_2}^*)^T\left[f(O,w) - \bar{f}(w,\theta_2)\right]\right|$$
$$\leq \left|(w - \hat{w}_{\theta_1}^*)^T\left[f(O,w) - \bar{f}(w,\theta_1)\right] - (w - \hat{w}_{\theta_2}^*)^T\left[f(O,w) - \bar{f}(w,\theta_1)\right]\right|$$
$$+ \left|(w - \hat{w}_{\theta_2}^*)^T\left[f(O,w) - \bar{f}(w,\theta_1)\right] - (w - \hat{w}_{\theta_2}^*)^T\left[f(O,w) - \bar{f}(w,\theta_2)\right]\right|$$
$$= \underbrace{\left|(\hat{w}_{\theta_2}^* - \hat{w}_{\theta_1}^*)^T\left[f(O,w) - \bar{f}(w,\theta_1)\right]\right|}_{I_1} + \underbrace{\left|(w - \hat{w}_{\theta_2}^*)^T\left[\bar{f}(w,\theta_2) - \bar{f}(w,\theta_1)\right]\right|}_{I_2}.$$

For $I_1$, we have
$$\left|(\hat{w}^*_{\theta_2} - \hat{w}^*_{\theta_1})^T \left[f(O, w) - \bar{f}(w, \theta_1)\right]\right| \leq 2U_f L_w \|\theta_1 - \theta_2\|,$$
which is by Lemma C.15 and Assumption 2.6.

For $I_2$, we have
$$\left|(w - \hat{w}^*_{\theta_2})^T \left[\bar{f}(w, \theta_2) - \bar{f}(w, \theta_1)\right]\right| \leq 2\sigma_w L_{\bar{f}} \|\theta_1 - \theta_2\|,$$
which is by Lemma C.16. Combining the above two facts we end the proof of the first part.

For the second part,
$$
\begin{aligned}
&|F(O, w_1, \theta) - F(O, w_2, \theta)| \\
&= \left|(w_1 - \hat{w}^*_\theta)^T \left[f(O, w_1) - \bar{f}(w_1, \theta)\right] - (w_2 - \hat{w}^*_\theta)^T \left[f(O, w_2) - \bar{f}(w_2, \theta)\right]\right| \\
&\leq \left|(w_1 - \hat{w}^*_\theta)^T \left[f(O, w_1) - \bar{f}(w_1, \theta)\right] - (w_1 - \hat{w}^*_\theta)^T \left[f(O, w_2) - \bar{f}(w_2, \theta)\right]\right| \\
&\quad + \left|(w_1 - \hat{w}^*_\theta)^T \left[f(O, w_2) - \bar{f}(w_2, \theta)\right] - (w_2 - \hat{w}^*_\theta)^T \left[f(O, w_2) - \bar{f}(w_2, \theta)\right]\right| \\
&= \left|(w_1 - \hat{w}^*_\theta)^T \left[(f(O, w_1) - f(O, w_2)) - \left(\bar{f}(w_1, \theta) - \bar{f}(w_2, \theta)\right)\right]\right| \\
&\quad + \left|(w_1 - w_2)^T \left[f(O, w_2) - \bar{f}(w_2, \theta)\right]\right| \\
&\leq 4\sigma_w L_f \|w_1 - w_2\| + 2U_f \|w_1 - w_2\|,
\end{aligned}
$$
where the last line is by Lemma C.15 and Lemma C.16.

For the third part, conditioned on $\theta_{t-\tau_{\mathrm{mix}}}$ and $s_{t-\tau_{\mathrm{mix}}+1}$,
$$
\begin{aligned}
&\left|\mathbb{E}\left[F(O_t, w_{t-\tau_{\mathrm{mix}}}, \theta_{t-\tau_{\mathrm{mix}}}) - F(\tilde{O}_t, w_{t-\tau_{\mathrm{mix}}}, \theta_{t-\tau_{\mathrm{mix}}})\right]\right| \\
&= \left|\mathbb{E}\left[(w_{t-\tau_{\mathrm{mix}}} - \hat{w}^*_{\theta_{t-\tau_{\mathrm{mix}}}})^T \left[f(O_t, w_{t-\tau_{\mathrm{mix}}}) - \bar{f}(w_{t-\tau_{\mathrm{mix}}}, \theta_{t-\tau_{\mathrm{mix}}})\right]\right.\right. \\
&\quad \left.\left. -(w_{t-\tau_{\mathrm{mix}}} - \hat{w}^*_{\theta_{t-\tau_{\mathrm{mix}}}})^T \left[f(\tilde{O}_t, w_{t-\tau_{\mathrm{mix}}}) - \bar{f}(w_{t-\tau_{\mathrm{mix}}}, \theta_{t-\tau_{\mathrm{mix}}})\right]\right]\right| \\
&= \left|\mathbb{E}\left[(w_{t-\tau_{\mathrm{mix}}} - \hat{w}^*_{\theta_{t-\tau_{\mathrm{mix}}}})^T \left[f(O_t, w_{t-\tau_{\mathrm{mix}}}) - f(\tilde{O}_t, w_{t-\tau_{\mathrm{mix}}})\right]\right]\right| \\
&\leq 2\sigma_w \left\|\mathbb{E}\left[f(O_t, w_{t-\tau_{\mathrm{mix}}}) - f(\tilde{O}_t, w_{t-\tau_{\mathrm{mix}}})\right]\right\| \\
&\leq 2\sigma_w U_f \|P(O_t \in \cdot) - P(\tilde{O}_t \in \cdot)\|_1 \\
&\leq 4\sigma_w U_f n_a L_\pi \sum_{i=t-\tau_{\mathrm{mix}}}^{t} \mathbb{E}\left[\|\theta_i - \theta_{t-\tau_{\mathrm{mix}}}\|\right],
\end{aligned}
$$
where we use Lemma C.15 and Lemma C.10.

For the fourth part, we first denote $O^+ = (s^+, a^+, s^{+\prime}, a^{+\prime})$ such that $(s^+, a^+) \sim \mu_{\theta_{t-\tau_{\mathrm{mix}}}}$, $s^{+\prime} \sim P_{\mathrm{env}}(s'|s, a)$ and $a^{+\prime} \sim \pi(a'|s', \theta_{t-\tau_{\mathrm{mix}}})$. Under this definition, we have
$$\mathbb{E}\left[F(O^+, w_{t-\tau_{\mathrm{mix}}}, \theta_{t-\tau_{\mathrm{mix}}})|\theta_{t-\tau_{\mathrm{mix}}}, s_{t-\tau_{\mathrm{mix}}+1}\right] = 0.$$
By Assumption 2.8, we have
$$\|P(\tilde{S}_t \in \cdot|\theta_{t-\tau_{\mathrm{mix}}}, s_{t-\tau_{\mathrm{mix}}-1}) - \mu'_{\theta_{t-\tau_{\mathrm{mix}}}}\|_1 \leq C\beta^{\tau_{\mathrm{mix}}-1}.$$
Hence, conditioned on $\theta_{t-\tau_{\mathrm{mix}}}$ and $s_{t-\tau_{\mathrm{mix}}+1}$,
$$
\begin{aligned}
&\left|\mathbb{E}\left[F(\tilde{O}_t, w_{t-\tau_{\mathrm{mix}}}, \theta_{t-\tau_{\mathrm{mix}}})\right]\right| \\
&= \left|\mathbb{E}\left[F(\tilde{O}_t, w_{t-\tau_{\mathrm{mix}}}, \theta_{t-\tau_{\mathrm{mix}}}) - F(O^+, w_{t-\tau_{\mathrm{mix}}}, \theta_{t-\tau_{\mathrm{mix}}})\right]\right| \\
&= \left|\mathbb{E}\left[(w_{t-\tau_{\mathrm{mix}}} - \hat{w}^*_{\theta_{t-\tau_{\mathrm{mix}}}})^T \left[f(\tilde{O}_t, w_{t-\tau_{\mathrm{mix}}}) - f(O^+, w_{t-\tau_{\mathrm{mix}}})\right]\right]\right| \\
&\leq 2\sigma_w U_f \|P(\tilde{O}_t \in \cdot) - P(O^+ \in \cdot)\|_1,
\end{aligned}
$$

where the last line is because of Lemma C.15. However,

$$||P(\tilde{O}_t \in \cdot) - P(O^+ \in \cdot)||_1$$

$$= \sum_{s,a,s',a'} |P(\tilde{S}_t = s, \tilde{A}_t = a, \tilde{S}_{t+1} = s', \tilde{A}_{t+1} = a') - P(S_t^+ = s, A_t^+ = a, S_{t+1}^+ = s', A_{t+1}^+ = a')|$$

$$= \sum_{s,a,s',a'} |P(\tilde{S}_t = s)\pi(a|s, \theta_{t-\tau_{\mathrm{mix}}})P_{\mathrm{env}}(s'|s,a)\pi(a'|s', \theta_{t-\tau_{\mathrm{mix}}})$$

$$- P(S_t^+ = s)\pi(a|s, \theta_{t-\tau_{\mathrm{mix}}})P_{\mathrm{env}}(s'|s,a)\pi(a'|s', \theta_{t-\tau_{\mathrm{mix}}})|$$

$$= ||P(\tilde{S}_t \in \cdot) - P(S_t^+ \in \cdot)||_1$$

$$\leq C\beta^{\tau_{\mathrm{mix}}-1}.$$

Hence, conditioned on $\theta_{t-\tau_{\mathrm{mix}}}$ and $s_{t-\tau_{\mathrm{mix}}+1}$,

$$\left| \mathbb{E}\left[ F(\tilde{O}_t, w_{t-\tau_{\mathrm{mix}}}, \theta_{t-\tau_{\mathrm{mix}}}) \right] \right| \leq 2\sigma_w U_f C\beta^{\tau_{\mathrm{mix}}-1}.$$

$$\square$$

The following lemma, which reveals a useful property for critic update, is inspired by Olshevsky & Gharesifard (2022).

**Lemma C.18.** *For critic update, we have the following two results:*

$$\mathbb{E}\left[ (\hat{w}_{\theta_t}^* - \hat{w}_{\theta_{t+1}}^*)^T (w_t - \hat{w}_{\theta_t}^* + \alpha^w \bar{f}(w_t, \theta_t)) | \mathcal{F}_t \right]$$

$$\leq \left( \sqrt{2}\alpha^\theta L_w ||\bar{g}(w_t, \theta_t)|| + \frac{\alpha^{\theta^2} H_w U_g^2}{\sqrt{2}(1-\gamma)^2} \right) \cdot \left\| w_t - \hat{w}_{\theta_t}^* + \alpha^w \bar{f}(w_t, \theta_t) \right\|,$$

*and*

$$\mathbb{E}\left[ (\hat{w}_{\theta_t}^* - \hat{w}_{\theta_{t+1}}^*)^T \alpha^w \left[ f(O_t, w_t) - \bar{f}(w_t, \theta_t) \right] \right] \leq 2\alpha^w U_f \left( \frac{\sqrt{2}\alpha^\theta L_w U_g}{1-\gamma} + \frac{\alpha^{\theta^2} H_w U_g^2}{\sqrt{2}(1-\gamma)^2} \right).$$

*Proof.* If we use $x(i)$ to denote the $i$'th entry of vector $x$,

$$(\hat{w}_{\theta_t}^* - \hat{w}_{\theta_{t+1}}^*)^T (w_t - \hat{w}_{\theta_t}^* + \alpha^w \bar{f}(w_t, \theta_t)) = \sum_i (\hat{w}_{\theta_t}^*(i) - \hat{w}_{\theta_{t+1}}^*(i)) \cdot (w_t - \hat{w}_{\theta_t}^* + \alpha^w \bar{f}(w_t, \theta_t))(i).$$

We can view $\hat{w}_\theta^*$ as a function of $\theta$. Using a second order expansion,

$$\hat{w}_{\theta_{t+1}}^*(i) = \hat{w}_{\theta_t}^*(i) + \nabla \hat{w}_{\theta_t}^*(i)^T (\theta_{t+1} - \theta_t) + \frac{1}{2}(\theta_{t+1} - \theta_t)^T \nabla^2 \hat{w}_{\theta_t'}^*(i)(\theta_{t+1} - \theta_t).$$

If we take expectation conditioned on $\mathcal{F}_t$,

$$\mathbb{E}\left[ \hat{w}_{\theta_t}^*(i) - \hat{w}_{\theta_{t+1}}^*(i) | \mathcal{F}_t \right] = \mathbb{E}\left[ \nabla \hat{w}_{\theta_t}^*(i)^T (\theta_t - \theta_{t+1}) | \mathcal{F}_t \right] - \mathbb{E}\left[ \frac{1}{2}(\theta_{t+1} - \theta_t)^T \nabla^2 \hat{w}_{\theta_t'}^*(i)(\theta_{t+1} - \theta_t) | \mathcal{F}_t \right]$$

$$= \alpha^\theta \nabla \hat{w}_{\theta_t}^*(i)^T \bar{g}(w_t, \theta_t) - \frac{\alpha^{\theta^2}}{2(1-\gamma)^2} \mathbb{E}\left[ g(\hat{O}_t, w_t, \theta_t)^T \nabla^2 \hat{w}_{\theta_t'}^*(i) g(\hat{O}_t, w_t, \theta_t) | \mathcal{F}_t \right].$$

This leads to

$$\mathbb{E}\left[ \hat{w}_{\theta_t}^*(i) - \hat{w}_{\theta_{t+1}}^*(i) | \mathcal{F}_t \right]^2$$

$$\leq 2\alpha^{\theta^2} \left( \nabla \hat{w}_{\theta_t}^*(i)^T \bar{g}(w_t, \theta_t) \right)^2 + \frac{\alpha^{\theta^4}}{2(1-\gamma)^4} \mathbb{E}\left[ g(\hat{O}_t, w_t, \theta_t)^T \nabla^2 \hat{w}_{\theta_t'}^*(i) g(\hat{O}_t, w_t, \theta_t) | \mathcal{F}_t \right]^2$$

$$\leq 2\alpha^{\theta^2} L_w(i)^2 ||\bar{g}(w_t, \theta_t)||^2 + \frac{\alpha^{\theta^4} H_w(i)^2 U_g^4}{2(1-\gamma)^4}.$$

where in the last line we use Assumption 2.6 and Lemma C.13. Now we go back to what we really care about,

$$\mathbb{E}\left[(\hat{w}^*_{\theta_t} - \hat{w}^*_{\theta_{t+1}})^T (w_t - \hat{w}^*_{\theta_t} + \alpha^w \bar{f}(w_t, \theta_t)) | \mathcal{F}_t\right]$$

$$= \sum_i \mathbb{E}\left[(\hat{w}^*_{\theta_t}(i) - \hat{w}^*_{\theta_{t+1}}(i)) \cdot (w_t - \hat{w}^*_{\theta_t} + \alpha^w \bar{f}(w_t, \theta_t))(i) | \mathcal{F}_t\right]$$

$$= \sum_i \mathbb{E}\left[\hat{w}^*_{\theta_t}(i) - \hat{w}^*_{\theta_{t+1}}(i) | \mathcal{F}_t\right] \cdot \left[(w_t - \hat{w}^*_{\theta_t} + \alpha^w \bar{f}(w_t, \theta_t))(i)\right]$$

$$\leq \sqrt{\sum_i \mathbb{E}\left[\hat{w}^*_{\theta_t}(i) - \hat{w}^*_{\theta_{t+1}}(i) | \mathcal{F}_t\right]^2} \cdot \sqrt{\sum_i \left[(w_t - \hat{w}^*_{\theta_t} + \alpha^w \bar{f}(w_t, \theta_t))(i)\right]^2}$$

$$\leq \sqrt{\sum_i \left(2\alpha^{\theta^2} L_w(i)^2 ||\bar{g}(w_t, \theta_t)||^2 + \frac{\alpha^{\theta^4} H_w(i)^2 U_g^4}{2(1-\gamma)^4}\right)} \cdot \left\|w_t - \hat{w}^*_{\theta_t} + \alpha^w \bar{f}(w_t, \theta_t)\right\|$$

$$\leq \left(\sqrt{2}\alpha^\theta L_w ||\bar{g}(w_t, \theta_t)|| + \frac{\alpha^{\theta^2} H_w U_g^2}{\sqrt{2}(1-\gamma)^2}\right) \cdot \left\|w_t - \hat{w}^*_{\theta_t} + \alpha^w \bar{f}(w_t, \theta_t)\right\|,$$

where the third line is because the second term is constant conditioned on $\mathcal{F}_t$.

For the second part, notice that we already have the following result

$$\mathbb{E}\left[\hat{w}^*_{\theta_t}(i) - \hat{w}^*_{\theta_{t+1}}(i) | \mathcal{F}_t\right]^2 \leq 2\alpha^{\theta^2} L_w(i)^2 ||\bar{g}(w_t, \theta_t)||^2 + \frac{\alpha^{\theta^4} H_w(i)^2 U_g^4}{2(1-\gamma)^4}$$

$$\leq \frac{2\alpha^{\theta^2} L_w(i)^2 U_g^2}{(1-\gamma)^2} + \frac{\alpha^{\theta^4} H_w(i)^2 U_g^4}{2(1-\gamma)^4},$$

where we simply bound $||\bar{g}(w_t, \theta_t)||$ by $U_g$ (this result is from Lemma C.13). On the other hand, by Lemma C.15 and Lemma C.16, a rough bound for $f(O_t, w_t) - \bar{f}(w_t, \theta_t)$ would be simply

$$\left\|f(O_t, w_t) - \bar{f}(w_t, \theta_t)\right\| \leq 2U_f.$$

Now we go back to what we really care about,

$$\mathbb{E}\left[(\hat{w}^*_{\theta_t} - \hat{w}^*_{\theta_{t+1}})^T \alpha^w \left[f(O_t, w_t) - \bar{f}(w_t, \theta_t)\right] | \mathcal{F}_t\right]$$

$$= \alpha^w \sum_i \mathbb{E}\left[(\hat{w}^*_{\theta_t}(i) - \hat{w}^*_{\theta_{t+1}}(i)) \cdot \left(f(O_t, w_t) - \bar{f}(w_t, \theta_t)\right)(i) | \mathcal{F}_t\right]$$

$$= \alpha^w \sum_i \mathbb{E}\left[\hat{w}^*_{\theta_t}(i) - \hat{w}^*_{\theta_{t+1}}(i) | \mathcal{F}_t\right] \cdot \mathbb{E}\left[\left(f(O_t, w_t) - \bar{f}(w_t, \theta_t)\right)(i) | \mathcal{F}_t\right]$$

$$\leq \alpha^w \sqrt{\sum_i \mathbb{E}\left[\hat{w}^*_{\theta_t}(i) - \hat{w}^*_{\theta_{t+1}}(i) | \mathcal{F}_t\right]^2} \cdot \sqrt{\sum_i \mathbb{E}\left[\left(f(O_t, w_t) - \bar{f}(w_t, \theta_t)\right)(i) | \mathcal{F}_t\right]^2}$$

$$\leq \alpha^w \sqrt{\sum_i \left(\frac{2\alpha^{\theta^2} L_w(i)^2 U_g^2}{(1-\gamma)^2} + \frac{\alpha^{\theta^4} H_w(i)^2 U_g^4}{2(1-\gamma)^4}\right)} \cdot \sqrt{\mathbb{E}\left[\sum_i \left(f(O_t, w_t) - \bar{f}(w_t, \theta_t)\right)(i)^2 | \mathcal{F}_t\right]}$$

$$\leq 2\alpha^w U_f \left(\frac{\sqrt{2}\alpha^\theta L_w U_g}{1-\gamma} + \frac{\alpha^{\theta^2} H_w U_g^2}{\sqrt{2}(1-\gamma)^2}\right),$$

where the third line is because the two terms are independent when conditioned on $\mathcal{F}_t$. The second part of this lemma is proved after we take expectation on both sides. $\square$

# D Actor Update Analysis

**Lemma D.1.**

$$\left(\frac{\alpha_\theta}{2} - \alpha_\theta{}^2 H_V\right)\Delta_V \leq \frac{\mathbb{E}\left[V^*_{\theta_1} - V^*_{\theta_{T+1}}\right]}{T} + c_1\left(\alpha_\theta{}^2 H_V + \frac{\alpha_\theta}{2}\right)\Delta_Q$$
$$+ c_2\left(\alpha_\theta{}^2 H_V + \frac{\alpha_\theta}{2}\right)\epsilon^2 + \frac{2H_V}{(1-\gamma)^2}\alpha^{\theta 2}U_g^2.$$

*Proof.* Actor update says the following:

$$\theta_{t+1} = \theta_t - \frac{\alpha^\theta}{1-\gamma}g(\hat{O}_t, w_t, \theta_t)$$

$$= \theta_t - \alpha^\theta \bar{g}(w_t, \theta_t) + \alpha^\theta \bar{g}(w_t, \theta_t) - \frac{\alpha^\theta}{1-\gamma}g(\hat{O}_t, w_t, \theta_t).$$

By the definition of $\bar{g}(w_t, \theta_t)$ in Eq.(16),

$$\bar{g}(w_t, \theta_t) = \nabla \ln \pi(\theta_t)^T \Phi_{\theta_t} Q(w_t)$$

$$= \nabla \ln \pi(\theta_t)^T \Phi_{\theta_t}\left[Q^*_{\theta_t} + Q(w_t) - Q^*_{\theta_t}\right]$$

$$= \nabla V^*_{\theta_t}{}^T + \nabla \ln \pi(\theta_t)^T \Phi_{\theta_t}\left[Q(w_t) - Q^*_{\theta_t}\right],$$

where we use the fact in Eq.(7) .

For simplicity, denote $D_Q = \nabla \ln \pi(\theta_t)^T \Phi_{\theta_t}\left[Q(w_t) - Q^*_{\theta_t}\right]$. So far we have the following result:

$$\theta_{t+1} - \theta_t = -\alpha^\theta \nabla V^*_{\theta_t} - \alpha^\theta D_Q + \alpha^\theta \bar{g}(w_t, \theta_t) - \frac{\alpha^\theta}{1-\gamma}g(\hat{O}_t, w_t, \theta_t).$$

On one hand, if we take expectation (conditioned on $\mathcal{F}_t$) on both sides, we get

$$\mathbb{E}[\theta_{t+1} - \theta_t | \mathcal{F}_t] = -\alpha^\theta \nabla V^*_{\theta_t} - \alpha^\theta D_Q.$$

where we use the fact in Eq.(16) that $\mathbb{E}\left[\frac{1}{1-\gamma}g(\hat{O}_t, w_t, \theta_t)\right] = \bar{g}(w_t, \theta_t)$. On the other hand,

$$\mathbb{E}\left[||\theta_{t+1} - \theta_t||^2 | \mathcal{F}_t\right] = ||\alpha^\theta \nabla V^*_{\theta_t} + \alpha^\theta D_Q||^2 + \mathbb{E}\left[||\alpha^\theta \bar{g}(w_t, \theta_t) - \frac{\alpha^\theta}{1-\gamma}g(\hat{O}_t, w_t, \theta_t)||^2 | \mathcal{F}_t\right]$$

$$\leq ||\alpha^\theta \nabla V^*_{\theta_t} + \alpha^\theta D_Q||^2 + \frac{4}{(1-\gamma)^2}\alpha^{\theta 2}U_g^2.$$

where we keep using $\mathbb{E}\left[\frac{1}{1-\gamma}g(\hat{O}_t, w_t, \theta_t)\right] = \bar{g}(w_t, \theta_t)$. Using Lemma C.12,

$$V^*_{\theta_{t+1}} \leq V^*_{\theta_t} + \nabla V^*_{\theta_t}(\theta_{t+1} - \theta_t) + \frac{H_V}{2}||\theta_{t+1} - \theta_t||^2.$$

Taking expectation (conditioned on $\mathcal{F}_t$) on both sides and we obtain

$$\mathbb{E}\left[V^*_{\theta_{t+1}} | \mathcal{F}_t\right] \leq V^*_{\theta_t} + \nabla V^*_{\theta_t} \cdot \mathbb{E}[\theta_{t+1} - \theta_t | \mathcal{F}_t] + \frac{H_V}{2}\mathbb{E}\left[||\theta_{t+1} - \theta_t||^2 | \mathcal{F}_t\right].$$

Plug in the facts about $\theta_{t+1} - \theta_t$, we know

$$\mathbb{E}\left[V^*_{\theta_{t+1}} | \mathcal{F}_t\right] \leq V^*_{\theta_t} - \alpha^\theta ||\nabla V^*_{\theta_t}||^2 - \alpha^\theta \nabla V^*_{\theta_t} D_Q + \frac{H_V}{2}||\alpha^\theta \nabla V^*_{\theta_t}{}^T + \alpha^\theta D_Q||^2 + \frac{2H_V}{(1-\gamma)^2}\alpha^{\theta 2}U_g^2.$$

We can use the facts that $2ab \leq a^2 + b^2$ and $(a+b)^2 \leq 2a^2 + 2b^2$ to obtain

$$\mathbb{E}\left[V^*_{\theta_{t+1}} | \mathcal{F}_t\right] \leq V^*_{\theta_t} - \alpha^\theta ||\nabla V^*_{\theta_t}||^2 + \frac{\alpha^\theta}{2}||\nabla V^*_{\theta_t}||^2 + \frac{\alpha^\theta}{2}||D_Q||^2$$

$$+ \alpha^{\theta 2}H_V ||\nabla V^*_{\theta_t}||^2 + \alpha^{\theta 2}H_V ||D_Q||^2 + \frac{2H_V}{(1-\gamma)^2}\alpha^{\theta 2}U_g^2$$

$$= V^*_{\theta_t} + \left(\alpha_\theta{}^2 H_V - \frac{\alpha_\theta}{2}\right)||\nabla V^*_{\theta_t}||^2 + \left(\alpha_\theta{}^2 H_V + \frac{\alpha_\theta}{2}\right)||D_Q||^2 + \frac{2H_V}{(1-\gamma)^2}\alpha^{\theta 2}U_g^2$$

$$\leq V^*_{\theta_t} + \left(\alpha_\theta{}^2 H_V - \frac{\alpha_\theta}{2}\right)||\nabla V^*_{\theta_t}||^2 + \frac{2H_V}{(1-\gamma)^2}\alpha^{\theta 2}U_g^2$$

$$+ \left(\alpha_\theta{}^2 H_V + \frac{\alpha_\theta}{2}\right) \cdot \left[c_1 \mathcal{N}_{\theta_t}(Q(w_t) - \hat{Q}^*_{\theta_t}) + c_2\epsilon^2\right],$$

where the last inequality we uses the fact from LemmaC.4. We can rewrite it as

$$\left(\frac{\alpha_\theta}{2} - \alpha_\theta^2 H_V\right) ||\nabla V_{\theta_t}^*||^2 \leq V_{\theta_t}^* - \mathbb{E}\left[V_{\theta_{t+1}}^* | \mathcal{F}_t\right] + \frac{2H_V}{(1-\gamma)^2}\alpha^{\theta^2} U_g^2$$
$$+ \left(\alpha_\theta^2 H_V + \frac{\alpha_\theta}{2}\right) \cdot \left[c_1 \mathcal{N}_{\theta_t}(Q(w_t) - \hat{Q}_{\theta_t}^*) + c_2 \epsilon^2\right].$$

Taking expectation on both sides and telescoping sum:

$$\left(\frac{\alpha_\theta}{2} - \alpha_\theta^2 H_V\right) \frac{1}{T} \sum_{t=1}^T \mathbb{E}\left[||\nabla V_{\theta_t}^*||^2\right] \leq \frac{\mathbb{E}[V_{\theta_1}^* - V_{\theta_{T+1}}^*]}{T} + c_2\left(\alpha_\theta^2 H_V + \frac{\alpha_\theta}{2}\right)\epsilon^2 + \frac{2H_V}{(1-\gamma)^2}\alpha^{\theta^2} U_g^2$$
$$+ c_1\left(\alpha_\theta^2 H_V + \frac{\alpha_\theta}{2}\right)\frac{1}{T}\sum_{t=1}^T \mathbb{E}\left[\mathcal{N}_{\theta_t}(Q(w_t) - \hat{Q}_{\theta_t}^*)\right].$$

If we use notations from Eq.(13), the above fact can be rewritten as

$$\left(\frac{\alpha_\theta}{2} - \alpha_\theta^2 H_V\right)\Delta_V \leq \frac{\mathbb{E}\left[V_{\theta_1}^* - V_{\theta_{T+1}}^*\right]}{T} + c_1\left(\alpha_\theta^2 H_V + \frac{\alpha_\theta}{2}\right)\Delta_Q$$
$$+ c_2\left(\alpha_\theta^2 H_V + \frac{\alpha_\theta}{2}\right)\epsilon^2 + \frac{2H_V}{(1-\gamma)^2}\alpha^{\theta^2} U_g^2.$$

$\square$

# E Critic Update Analysis under the Markov Sampling Case

**Lemma E.1.** *In the Markov sampling case,*

$$\left(2\alpha^w - \frac{1}{\sqrt{2}}\alpha^\theta L_w\left(\frac{1}{\lambda_{\min}} - 2\alpha^w\right) - \sqrt{2}\alpha^\theta L_w\sqrt{c_1\left(\frac{1}{\lambda_{\min}} - 2\alpha^w\right)} - 2\alpha^{\theta^2}L_w^2 c_1\right)\cdot\Delta_Q$$

$$\leq\frac{\mathbb{E}\left[||w_1 - \hat{w}_{\theta_1}^*||^2\right]}{T} + \left(\frac{1}{\sqrt{2}}\alpha^\theta L_w + 2\alpha^{\theta^2}L_w^2\right)\Delta_V$$

$$+\left(\sqrt{2}\alpha^\theta L_w\sqrt{c_1 C_{w,1} + c_2\epsilon^2\left(\frac{1}{\lambda_{\min}} - 2\alpha^w\right)} + \frac{\alpha^{\theta^2}H_w U_g^2}{\sqrt{2}(1-\gamma)^2}\sqrt{\frac{1}{\lambda_{\min}} - 2\alpha^w}\right)\sqrt{\Delta_Q}$$

$$+C_{w,1} + C_{w,2} + \frac{1}{\sqrt{2}}\alpha^\theta L_w C_{w,1} + \sqrt{2}\alpha^\theta L_w\sqrt{c_2 C_{w,1}\epsilon^2} + \frac{\sqrt{C_{w,1}}\alpha^{\theta^2}H_w U_g^2}{\sqrt{2}(1-\gamma)^2}$$

$$+2\alpha^w U_f\left(\frac{\sqrt{2}\alpha^\theta L_w U_g}{1-\gamma} + \frac{\alpha^{\theta^2}H_w U_g^2}{\sqrt{2}(1-\gamma)^2}\right) + 2\alpha^{\theta^2}L_w^2 c_2\epsilon^2 + \frac{4L_w^2}{(1-\gamma)^2}\alpha^{\theta^2}U_g^2,$$

*where, for simplicity, we denote*

$$C_{w,1} = 16\alpha^w(1+\gamma)L_Q H_Q\sigma_w^3 + 4\alpha^w(1+\gamma)L_Q\epsilon\sigma_w + \alpha^{w2}U_f^2,$$

*and*

$$C_{w,2} = 2\alpha^w\alpha^\theta\tau_{\mathrm{mix}}\left(L_{F_\theta} + 4\sigma_w U_f n_a L_\pi(\tau_{\mathrm{mix}} - 1)\right)U_g + 2\alpha^{w2}\tau_{\mathrm{mix}}L_{F_w}\sigma_w + 4\alpha^w\sigma_w U_f C\beta^{\tau_{\mathrm{mix}}-1}.$$

*Proof.* Recall the critic update is

$$w_{t+1} = \mathbf{Proj}_W\left\{w_t + \alpha^w f(O_t, w_t)\right\},$$

which implies

$$
\begin{aligned}
||w_{t+1} - \hat{w}_{\theta_{t+1}}^*||^2 =& ||\mathbf{Proj}_W\left\{w_t + \alpha^w f(O_t, w_t)\right\} - \hat{w}_{\theta_{t+1}}^*||^2\\
\leq& ||w_t + \alpha^w f(O_t, w_t) - \hat{w}_{\theta_{t+1}}^*||^2\\
=& ||w_t - \hat{w}_{\theta_t}^* + \hat{w}_{\theta_t}^* - \hat{w}_{\theta_{t+1}}^* + \alpha^w f(O_t, w_t)||^2\\
\leq& ||w_t - \hat{w}_{\theta_t}^* + \alpha^w f(O_t, w_t)||^2\\
&+ 2(\hat{w}_{\theta_t}^* - \hat{w}_{\theta_{t+1}}^*)^T(w_t - \hat{w}_{\theta_t}^* + \alpha^w f(O_t, w_t)) + ||\hat{w}_{\theta_t}^* - \hat{w}_{\theta_{t+1}}^*||^2.
\end{aligned}
$$

We can take expectation on both sides,

$$
\begin{aligned}
&\mathbb{E}\left[||w_{t+1} - \hat{w}_{\theta_{t+1}}^*||^2\right]\\
\leq& \underbrace{\mathbb{E}\left[||w_t - \hat{w}_{\theta_t}^* + \alpha^w f(O_t, w_t)||^2\right]}_{I_1}\\
&+ \underbrace{\mathbb{E}\left[2(\hat{w}_{\theta_t}^* - \hat{w}_{\theta_{t+1}}^*)^T(w_t - \hat{w}_{\theta_t}^* + \alpha^w f(O_t, w_t))\right]}_{I_2} + \underbrace{\mathbb{E}\left[||\hat{w}_{\theta_t}^* - \hat{w}_{\theta_{t+1}}^*||^2\right]}_{I_3}.
\end{aligned}
\tag{17}
$$

To analyze $I_1$, we derive

$$
\begin{aligned}
&\mathbb{E}\left[||w_t - \hat{w}_{\theta_t}^* + \alpha^w f(O_t, w_t)||^2\right]\\
=& \mathbb{E}\left[||w_t - \hat{w}_{\theta_t}^*||^2 + 2\alpha^w(w_t - \hat{w}_{\theta_t}^*)^T f(O_t, w_t) + \alpha^{w2}||f(O_t, w_t)||^2\right]\\
=& \mathbb{E}\left[||w_t - \hat{w}_{\theta_t}^*||^2\right] + \underbrace{\mathbb{E}\left[2\alpha^w(w_t - \hat{w}_{\theta_t}^*)^T\bar{f}(w_t, \theta_t)\right]}_{I_{1,1}}\\
&+ \underbrace{\mathbb{E}\left[\alpha^{w2}||f(O_t, w_t)||^2\right]}_{I_{1,2}} + \underbrace{\mathbb{E}\left[2\alpha^w(w_t - \hat{w}_{\theta_t}^*)^T\left[f(O_t, w_t) - \bar{f}(w_t, \theta_t)\right]\right]}_{I_{1,3}}.
\end{aligned}
\tag{18}
$$

To analyze $I_{1,1}$, we perform the following of manipulations:

$$2\alpha^w(w_t - \hat{w}^*_{\theta_t})^T \bar{f}(w_t, \theta_t)$$

$$=2\alpha^w(w_t - \hat{w}^*_{\theta_t})^T \nabla Q(w_t)^T D_{\theta_t}(\gamma P_{\theta_t} - I)(Q(w_t) - Q^*_{\theta_t})$$

$$=2\alpha^w(w_t - \hat{w}^*_{\theta_t})^T \nabla Q(w_t)^T D_{\theta_t}(\gamma P_{\theta_t} - I)(Q(w_t) - \hat{Q}^*_{\theta_t} + \hat{Q}^*_{\theta_t} - Q^*_{\theta_t})$$

$$=2\alpha^w \left[ (Q(w_t) - \hat{Q}^*_{\theta_t})^T + (w_t - \hat{w}^*_{\theta_t})^T \nabla Q(w_t)^T - (Q(w_t) - \hat{Q}^*_{\theta_t})^T \right] D_{\theta_t}(\gamma P_{\theta_t} - I)(Q(w_t) - \hat{Q}^*_{\theta_t})$$

$$\quad + 2\alpha^w(w_t - \hat{w}^*_{\theta_t})^T \nabla Q(w_t)^T D_{\theta_t}(\gamma P_{\theta_t} - I)(\hat{Q}^*_{\theta_t} - Q^*_{\theta_t})$$

$$= - 2\alpha^w \mathcal{N}_{\theta_t}(Q(w_t) - \hat{Q}^*_{\theta_t}) + 2\alpha^w(w_t - \hat{w}^*_{\theta_t})^T(\nabla Q(w_t) - \nabla Q(w_{\mathrm{mid}}))^T D_{\theta_t}(\gamma P_{\theta_t} - I)(Q(w_t) - \hat{Q}^*_{\theta_t})$$

$$\quad + 2\alpha^w(w_t - \hat{w}^*_{\theta_t})^T \nabla Q(w_t)^T D_{\theta_t}(\gamma P_{\theta_t} - I)(\hat{Q}^*_{\theta_t} - Q^*_{\theta_t})$$

$$\leq - 2\alpha^w \mathcal{N}_{\theta_t}(Q(w_t) - \hat{Q}^*_{\theta_t}) + 2\alpha^w(1 + \gamma)L_Q H_Q ||w_t - \hat{w}^*_{\theta_t}||^3 + 2\alpha^w(1 + \gamma)L_Q \epsilon ||w_t - \hat{w}^*_{\theta_t}||$$

$$\leq - 2\alpha^w \mathcal{N}_{\theta_t}(Q(w_t) - \hat{Q}^*_{\theta_t}) + 16\alpha^w(1 + \gamma)L_Q H_Q \sigma_w^3 + 4\alpha^w(1 + \gamma)L_Q \epsilon \sigma_w,$$

where the second line is by the definition in Eq.(16), the sixth line is by Lemma (C.2) and Lemma C.5, and the eighth line is by Lemma C.1. Here, $\lambda \in [0, 1]$ is some scalar and $w_{\mathrm{mid}} = \lambda w_t + (1 - \lambda)\hat{w}^*_{\theta_t}$. So we arrive at the final bound for $I_{1,1}$:

$$2\alpha^w(w_t - \hat{w}^*_{\theta_t})^T \bar{f}(w_t, \theta_t) \leq - 2\alpha^w \mathcal{N}_{\theta_t}(Q(w_t) - \hat{Q}^*_{\theta_t})$$
$$+ 16\alpha^w(1 + \gamma)L_Q H_Q \sigma_w^3 + 4\alpha^w(1 + \gamma)L_Q \epsilon \sigma_w.$$

To analyze $I_{1,2}$, we conclude

$$\mathbb{E}\left[ \alpha^{w2} ||f(O_t, w_t)||^2 \right] \leq \alpha^{w2} U_f^2.$$

To analyze $I_{1,3}$, for simplicity, we denote $F(O_t, w, \theta) = (w - \hat{w}^*_\theta)^T \left[ f(O_t, w) - \bar{f}(w, \theta) \right]$. We have

$$F(O_t, w_t, \theta_t) = \underbrace{F(O_t, w_t, \theta_t) - F(O_t, w_t, \theta_{t-\tau_{\mathrm{mix}}})}_{J_1}$$

$$+ \underbrace{F(O_t, w_t, \theta_{t-\tau_{\mathrm{mix}}}) - F(O_t, w_{t-\tau_{\mathrm{mix}}}, \theta_{t-\tau_{\mathrm{mix}}})}_{J_2}$$

$$+ \underbrace{F(O_t, w_{t-\tau_{\mathrm{mix}}}, \theta_{t-\tau_{\mathrm{mix}}}) - F(\tilde{O}_t, w_{t-\tau_{\mathrm{mix}}}, \theta_{t-\tau_{\mathrm{mix}}})}_{J_3}$$

$$+ \underbrace{F(\tilde{O}_t, w_{t-\tau_{\mathrm{mix}}}, \theta_{t-\tau_{\mathrm{mix}}})}_{J_4}.$$

For $J_1$, we have

$$|F(O_t, w_t, \theta_t) - F(O_t, w_t, \theta_{t-\tau_{\mathrm{mix}}})| \leq L_{F_\theta} ||\theta_t - \theta_{t-\tau_{\mathrm{mix}}}||$$
$$\leq \frac{\alpha^\theta \tau_{\mathrm{mix}} L_{F_\theta} U_g}{1 - \gamma}.$$

For $J_2$, we have

$$|F(O_t, w_t, \theta_{t-\tau_{\mathrm{mix}}}) - F(O_t, w_{t-\tau_{\mathrm{mix}}}, \theta_{t-\tau_{\mathrm{mix}}})| \leq L_{F_w} ||w_t - w_{t-\tau_{\mathrm{mix}}}||$$
$$\leq \alpha^w \tau_{\mathrm{mix}} L_{F_w} \sigma_w.$$

For $J_3$, by Lemma C.17, we have

$$\left| \mathbb{E}\left[ F(O_t, w_{t-\tau_{\mathrm{mix}}}, \theta_{t-\tau_{\mathrm{mix}}}) - F(\tilde{O}_t, w_{t-\tau_{\mathrm{mix}}}, \theta_{t-\tau_{\mathrm{mix}}}) | \theta_{t-\tau_{\mathrm{mix}}}, s_{t-\tau_{\mathrm{mix}}+1} \right] \right|$$

$$\leq 4\sigma_w U_f n_a L_\pi \sum_{i=t-\tau_{\mathrm{mix}}}^{t} \mathbb{E}\left[ ||\theta_i - \theta_{t-\tau_{\mathrm{mix}}}|| \, | \theta_{t-\tau_{\mathrm{mix}}}, s_{t-\tau_{\mathrm{mix}}+1} \right]$$

$$\leq \frac{4\alpha^\theta \sigma_w U_f n_a L_\pi \tau_{\mathrm{mix}}(\tau_{\mathrm{mix}} - 1)U_g}{1 - \gamma},$$

which means

$$\left| \mathbb{E}\left[ F(O_t, w_{t-\tau_{\text{mix}}}, \theta_{t-\tau_{\text{mix}}}) - F(\tilde{O}_t, w_{t-\tau_{\text{mix}}}, \theta_{t-\tau_{\text{mix}}}) \right] \right| \leq 4\alpha^\theta \sigma_w U_f n_a L_\pi \tau_{\text{mix}}(\tau_{\text{mix}} - 1) U_g.$$

For $J_4$, by Lemma C.17, we have

$$\left| \mathbb{E}\left[ F(\tilde{O}_t, w_{t-\tau_{\text{mix}}}, \theta_{t-\tau_{\text{mix}}}) | \theta_{t-\tau_{\text{mix}}}, s_{t-\tau_{\text{mix}}+1} \right] \right| \leq 2\sigma_w U_f C \beta^{\tau_{\text{mix}}-1},$$

which means

$$\left| \mathbb{E}\left[ F(\tilde{O}_t, w_{t-\tau_{\text{mix}}}, \theta_{t-\tau_{\text{mix}}}) \right] \right| \leq 2\sigma_w U_f C \beta^{\tau_{\text{mix}}-1}.$$

Hence, for $I_{1,3}$, we have

$$\mathbb{E}\left[ F(O_t, w_t, \theta_t) \right] \leq \frac{\alpha^\theta \tau_{\text{mix}} \left( L_{F_\theta} + 4\sigma_w U_f n_a L_\pi (\tau_{\text{mix}} - 1) \right) U_g}{1 - \gamma} + \alpha^w \tau_{\text{mix}} L_{F_w} \sigma_w + 2\sigma_w U_f C \beta^{\tau_{\text{mix}}-1},$$

which implies

$$\mathbb{E}\left[ 2\alpha^w (w_t - \hat{w}_{\theta_t}^*)^T \left[ f(O_t, w_t) - \bar{f}(w_t, \theta_t) \right] \right] \leq \frac{2\alpha^w \alpha^\theta \tau_{\text{mix}} \left( L_{F_\theta} + 4\sigma_w U_f n_a L_\pi (\tau_{\text{mix}} - 1) \right) U_g}{1 - \gamma}$$
$$+ 2\alpha^{w^2} \tau_{\text{mix}} L_{F_w} \sigma_w + 4\alpha^w \sigma_w U_f C \beta^{\tau_{\text{mix}}-1}.$$

Hence, for $I_1$,

$$\mathbb{E}\left[ ||w_t - \hat{w}_{\theta_t}^* + \alpha^w f(O_t, w_t)||^2 \right] \leq - 2\alpha^w \mathbb{E}\left[ \mathcal{N}_{\theta_t}(Q(w_t) - \hat{Q}_{\theta_t}^*) \right] + \mathbb{E}\left[ ||w_t - \hat{w}_{\theta_t}^*||^2 \right]$$
$$+ 16\alpha^w (1 + \gamma) L_Q H_Q \sigma_w^3 + 4\alpha^w (1 + \gamma) L_Q \epsilon \sigma_w$$
$$+ \alpha^{w^2} U_f^2 + \frac{2\alpha^w \alpha^\theta \tau_{\text{mix}} \left( L_{F_\theta} + 4\sigma_w U_f n_a L_\pi (\tau_{\text{mix}} - 1) \right) U_g}{1 - \gamma}$$
$$+ 2\alpha^{w^2} \tau_{\text{mix}} L_{F_w} \sigma_w + 4\alpha^w \sigma_w U_f C \beta^{\tau_{\text{mix}}-1}$$
$$= - 2\alpha^w \mathbb{E}\left[ \mathcal{N}_{\theta_t}(Q(w_t) - \hat{Q}_{\theta_t}^*) \right] + \mathbb{E}\left[ ||w_t - \hat{w}_{\theta_t}^*||^2 \right] + C_{w,1} + C_{w,2},$$
(19)

where, for simplicity, we denote

$$C_{w,1} = 16\alpha^w (1 + \gamma) L_Q H_Q \sigma_w^3 + 4\alpha^w (1 + \gamma) L_Q \epsilon \sigma_w + \alpha^{w^2} U_f^2$$

and

$$C_{w,2} = \frac{2\alpha^w \alpha^\theta \tau_{\text{mix}} \left( L_{F_\theta} + 4\sigma_w U_f n_a L_\pi (\tau_{\text{mix}} - 1) \right) U_g}{1 - \gamma} + 2\alpha^{w^2} \tau_{\text{mix}} L_{F_w} \sigma_w + 4\alpha^w \sigma_w U_f C \beta^{\tau_{\text{mix}}-1}.$$

To analyze $I_2$, we derive

$$\mathbb{E}\left[ (\hat{w}_{\theta_t}^* - \hat{w}_{\theta_{t+1}}^*)^T (w_t - \hat{w}_{\theta_t}^* + \alpha^w f(O_t, w_t)) \right]$$
$$= \underbrace{\mathbb{E}\left[ (\hat{w}_{\theta_t}^* - \hat{w}_{\theta_{t+1}}^*)^T (w_t - \hat{w}_{\theta_t}^* + \alpha^w \bar{f}(w_t, \theta_t)) \right]}_{I_{2,1}} + \underbrace{\mathbb{E}\left[ (\hat{w}_{\theta_t}^* - \hat{w}_{\theta_{t+1}}^*)^T \alpha^w \left[ f(O_t, w_t) - \bar{f}(w_t, \theta_t) \right] \right]}_{I_{2,2}}.$$

To analyze $I_{2,1}$ first, by lemma C.18, we already know

$$\mathbb{E}\left[ (\hat{w}_{\theta_t}^* - \hat{w}_{\theta_{t+1}}^*)^T (w_t - \hat{w}_{\theta_t}^* + \alpha^w \bar{f}(w_t, \theta_t)) | \mathcal{F}_t \right]$$
$$\leq \underbrace{\left( \sqrt{2}\alpha^\theta L_w ||\bar{g}(w_t, \theta_t)|| + \frac{\alpha^{\theta^2} H_w U_g^2}{\sqrt{2}(1 - \gamma)^2} \right)}_{I_{2,1,1}} \cdot \underbrace{\left\| w_t - \hat{w}_{\theta_t}^* + \alpha^w \bar{f}(w_t, \theta_t) \right\|}_{I_{2,1,2}}.$$

For $I_{2,1,2}$, a rough bound would be

$$\left\| w_t - \hat{w}_{\theta_t}^* + \alpha^w \bar{f}(w_t, \theta_t) \right\|^2 = ||w_t - \hat{w}_{\theta_t}^*||^2 + 2\alpha^w (w_t - \hat{w}_{\theta_t}^*)^T \bar{f}(w_t, \theta_t) + \alpha^{w^2} ||\bar{f}(w_t, \theta_t)||^2.$$

Notice that this term is very similar to what we have in Eq.(18). The two difference are (a) all the $f$ in Eq.(18) are replaced by $\bar{f}$ and (b) expectation is removed. In this case, $I_{1,3}$ will be 0 and a similar bound for $I_{1,1}$ and $I_{1,2}$ will also hold. This implies the following result:

$$
\begin{aligned}
\left\|w_t - \hat{w}^*_{\theta_t} + \alpha^w \bar{f}(w_t, \theta_t)\right\|^2 \leq &- 2\alpha^w \mathcal{N}_{\theta_t}(Q(w_t) - \hat{Q}^*_{\theta_t}) + \|w_t - \hat{w}^*_{\theta_t}\|^2 + 16\alpha^w(1+\gamma)L_Q H_Q \sigma^3_w \\
&+ 4\alpha^w(1+\gamma)L_Q \epsilon \sigma_w + \alpha^{w2} U_f^2 \\
\leq &\left(\frac{1}{\lambda_{\min}} - 2\alpha^w\right) \mathcal{N}_{\theta_t}(Q(w_t) - \hat{Q}^*_{\theta_t}) + C_{w,1},
\end{aligned}
\tag{20}
$$

where the last line uses Lemma C.3.

For $I_{2,1,1}$, recall that

$$
\begin{aligned}
\|\bar{g}(w_t, \theta_t)\| &\leq \left\|\nabla V^*_{\theta_t}\right\| + \|D_Q\| \\
&\leq \left\|\nabla V^*_{\theta_t}\right\| + \sqrt{c_1 \mathcal{N}_{\theta_t}(Q(w_t) - \hat{Q}^*_{\theta_t}) + c_2 \epsilon^2}.
\end{aligned}
$$

Hence, we have the following bound for $I_{2,1,1}$:

$$
\begin{aligned}
&\sqrt{2}\alpha^\theta L_w \|\bar{g}(w_t, \theta_t)\| + \frac{\alpha^{\theta 2} H_w U_g^2}{\sqrt{2}(1-\gamma)^2} \\
\leq &\sqrt{2}\alpha^\theta L_w \left\|\nabla V^*_{\theta_t}\right\| + \sqrt{2}\alpha^\theta L_w \sqrt{c_1 \mathcal{N}_{\theta_t}(Q(w_t) - \hat{Q}^*_{\theta_t}) + c_2 \epsilon^2} + \frac{\alpha^{\theta 2} H_w U_g^2}{\sqrt{2}(1-\gamma)^2}.
\end{aligned}
$$

Hence, we have the following result:

$$
\begin{aligned}
&\mathbb{E}\left[(\hat{w}^*_{\theta_t} - \hat{w}^*_{\theta_{t+1}})^T (w_t - \hat{w}^*_{\theta_t} + \alpha^w \bar{f}(w_t, \theta_t))|\mathcal{F}_t\right] \\
\leq &\left(\sqrt{2}\alpha^\theta L_w \left\|\nabla V^*_{\theta_t}\right\| + \sqrt{2}\alpha^\theta L_w \sqrt{c_1 \mathcal{N}_{\theta_t}(Q(w_t) - \hat{Q}^*_{\theta_t}) + c_2 \epsilon^2} + \frac{\alpha^{\theta 2} H_w U_g^2}{\sqrt{2}(1-\gamma)^2}\right) \\
&\quad \cdot \sqrt{\left(\frac{1}{\lambda_{\min}} - 2\alpha^w\right) \mathcal{N}_{\theta_t}(Q(w_t) - \hat{Q}^*_{\theta_t}) + C_{w,1}} \\
\leq &\frac{1}{\sqrt{2}}\alpha^\theta L_w \left\|\nabla V^*_{\theta_t}\right\|^2 + \frac{1}{\sqrt{2}}\alpha^\theta L_w \left(\frac{1}{\lambda_{\min}} - 2\alpha^w\right) \mathcal{N}_{\theta_t}(Q(w_t) - \hat{Q}^*_{\theta_t}) + \frac{1}{\sqrt{2}}\alpha^\theta L_w C_{w,1} \\
&+ \sqrt{2}\alpha^\theta L_w \sqrt{\left[c_1 \mathcal{N}_{\theta_t}(Q(w_t) - \hat{Q}^*_{\theta_t}) + c_2 \epsilon^2\right] \cdot \left[\left(\frac{1}{\lambda_{\min}} - 2\alpha^w\right) \mathcal{N}_{\theta_t}(Q(w_t) - \hat{Q}^*_{\theta_t}) + C_{w,1}\right]} \\
&+ \frac{\alpha^{\theta 2} H_w U_g^2}{\sqrt{2}(1-\gamma)^2} \sqrt{\left(\frac{1}{\lambda_{\min}} - 2\alpha^w\right) \mathcal{N}_{\theta_t}(Q(w_t) - \hat{Q}^*_{\theta_t}) + C_{w,1}} \\
\leq &\frac{1}{\sqrt{2}}\alpha^\theta L_w \left\|\nabla V^*_{\theta_t}\right\|^2 \\
&+ \left(\frac{1}{\sqrt{2}}\alpha^\theta L_w \left(\frac{1}{\lambda_{\min}} - 2\alpha^w\right) + \sqrt{2}\alpha^\theta L_w \sqrt{c_1 \left(\frac{1}{\lambda_{\min}} - 2\alpha^w\right)}\right) \cdot \mathcal{N}_{\theta_t}(Q(w_t) - \hat{Q}^*_{\theta_t}) \\
&+ \left(\sqrt{2}\alpha^\theta L_w \sqrt{c_1 C_{w,1} + c_2 \epsilon^2 \left(\frac{1}{\lambda_{\min}} - 2\alpha^w\right)} + \frac{\alpha^{\theta 2} H_w U_g^2}{\sqrt{2}(1-\gamma)^2} \sqrt{\frac{1}{\lambda_{\min}} - 2\alpha^w}\right) \sqrt{\mathcal{N}_{\theta_t}(Q(w_t) - \hat{Q}^*_{\theta_t})} \\
&+ \frac{1}{\sqrt{2}}\alpha^\theta L_w C_{w,1} + \sqrt{2}\alpha^\theta L_w \sqrt{c_2 C_{w,1} \epsilon^2} + \frac{\sqrt{C_{w,1}}\alpha^{\theta 2} H_w U_g^2}{\sqrt{2}(1-\gamma)^2}.
\end{aligned}
$$

After taking expectation on both sides, we know

$$\mathbb{E}\left[(\hat{w}_{\theta_t}^* - \hat{w}_{\theta_{t+1}}^*)^T(w_t - \hat{w}_{\theta_t}^* + \alpha^w \bar{f}(w_t, \theta_t))\right]$$

$$\leq \frac{1}{\sqrt{2}}\alpha^\theta L_w \mathbb{E}\left[\|\nabla V_{\theta_t}^*\|^2\right]$$

$$+ \left(\frac{1}{\sqrt{2}}\alpha^\theta L_w\left(\frac{1}{\lambda_{\min}} - 2\alpha^w\right) + \sqrt{2}\alpha^\theta L_w\sqrt{c_1\left(\frac{1}{\lambda_{\min}} - 2\alpha^w\right)}\right) \cdot \mathbb{E}\left[\mathcal{N}_{\theta_t}(Q(w_t) - \hat{Q}_{\theta_t}^*)\right]$$

$$+ \left(\sqrt{2}\alpha^\theta L_w\sqrt{c_1 C_{w,1} + c_2\epsilon^2\left(\frac{1}{\lambda_{\min}} - 2\alpha^w\right)} + \frac{\alpha^{\theta^2}H_w U_g^2}{\sqrt{2}(1-\gamma)^2}\sqrt{\frac{1}{\lambda_{\min}} - 2\alpha^w}\right)\sqrt{\mathbb{E}\left[\mathcal{N}_{\theta_t}(Q(w_t) - \hat{Q}_{\theta_t}^*)\right]}$$

$$+ \frac{1}{\sqrt{2}}\alpha^\theta L_w C_{w,1} + \sqrt{2}\alpha^\theta L_w\sqrt{c_2 C_{w,1}\epsilon^2} + \frac{\sqrt{C_{w,1}}\alpha^{\theta^2}H_w U_g^2}{\sqrt{2}(1-\gamma)^2},$$

which is the bound for $I_{2,1}$.

For $I_{2,2}$, by Lemma C.18, we have

$$\mathbb{E}\left[(\hat{w}_{\theta_t}^* - \hat{w}_{\theta_{t+1}}^*)^T\alpha^w\left[f(O_t, w_t) - \bar{f}(w_t, \theta_t)\right]\right] \leq 2\alpha^w U_f\left(\frac{\sqrt{2}\alpha^\theta L_w U_g}{1-\gamma} + \frac{\alpha^{\theta^2}H_w U_g^2}{\sqrt{2}(1-\gamma)^2}\right).$$

This ends the bound for $I_2$, which, after combining the bound for $I_{2,1}$ and $I_{2,2}$, will be

$$\mathbb{E}\left[(\hat{w}_{\theta_t}^* - \hat{w}_{\theta_{t+1}}^*)^T(w_t - \hat{w}_{\theta_t}^* + \alpha^w f(O_t, w_t))\right]$$

$$\leq \frac{1}{\sqrt{2}}\alpha^\theta L_w \mathbb{E}\left[\|\nabla V_{\theta_t}^*\|^2\right]$$

$$+ \left(\frac{1}{\sqrt{2}}\alpha^\theta L_w\left(\frac{1}{\lambda_{\min}} - 2\alpha^w\right) + \sqrt{2}\alpha^\theta L_w\sqrt{c_1\left(\frac{1}{\lambda_{\min}} - 2\alpha^w\right)}\right) \cdot \mathbb{E}\left[\mathcal{N}_{\theta_t}(Q(w_t) - \hat{Q}_{\theta_t}^*)\right]$$

$$+ \left(\sqrt{2}\alpha^\theta L_w\sqrt{c_1 C_{w,1} + c_2\epsilon^2\left(\frac{1}{\lambda_{\min}} - 2\alpha^w\right)} + \frac{\alpha^{\theta^2}H_w U_g^2}{\sqrt{2}(1-\gamma)^2}\sqrt{\frac{1}{\lambda_{\min}} - 2\alpha^w}\right)\sqrt{\mathbb{E}\left[\mathcal{N}_{\theta_t}(Q(w_t) - \hat{Q}_{\theta_t}^*)\right]}$$

$$+ \frac{1}{\sqrt{2}}\alpha^\theta L_w C_{w,1} + \sqrt{2}\alpha^\theta L_w\sqrt{c_2 C_{w,1}\epsilon^2} + \frac{\sqrt{C_{w,1}}\alpha^{\theta^2}H_w U_g^2}{\sqrt{2}(1-\gamma)^2} + 2\alpha^w U_f\left(\frac{\sqrt{2}\alpha^\theta L_w U_g}{1-\gamma} + \frac{\alpha^{\theta^2}H_w U_g^2}{\sqrt{2}(1-\gamma)^2}\right).$$

To analyze $I_3$, we have

$$\mathbb{E}\left[\|\hat{w}_{\theta_t}^* - \hat{w}_{\theta_{t+1}}^*\|^2|\mathcal{F}_t\right] \leq L_w^2 \mathbb{E}\left[\|\theta_{t+1} - \theta_t\|^2|\mathcal{F}_t\right].$$

Recall that

$$\theta_{t+1} - \theta_t = -\alpha^\theta\nabla V_{\theta_t}^* - \alpha^\theta D_Q + \alpha^\theta\bar{g}(w_t, \theta_t) - \frac{\alpha^\theta}{1-\gamma}g(\hat{O}_t, w_t, \theta_t),$$

which implies

$$\mathbb{E}\left[\|\theta_{t+1} - \theta_t\|^2|\mathcal{F}_t\right] \leq \|\alpha^\theta\nabla V_{\theta_t}^* + \alpha^\theta D_Q\|^2 + \frac{4}{(1-\gamma)^2}\alpha^{\theta^2}U_g^2$$

$$\leq 2\alpha^{\theta^2}\left(\|\nabla V_{\theta_t}^*\|^2 + c_1 \cdot \mathcal{N}_{\theta_t}(Q(w_t) - \hat{Q}_{\theta_t}^*) + c_2\epsilon^2\right) + \frac{4}{(1-\gamma)^2}\alpha^{\theta^2}U_g^2,$$

where we use the fact that $\mathbb{E}\left[\bar{g}(w_t, \theta_t) - \frac{1}{1-\gamma}g(\hat{O}_t, w_t, \theta_t)|\mathcal{F}_t\right] = 0$. Hence,

$$\mathbb{E}\left[\|\hat{w}_{\theta_t}^* - \hat{w}_{\theta_{t+1}}^*\|^2|\mathcal{F}_t\right] \leq 2\alpha^{\theta^2}L_w^2\left(\|\nabla V_{\theta_t}^*\|^2 + c_1 \cdot \mathcal{N}_{\theta_t}(Q(w_t) - \hat{Q}_{\theta_t}^*) + c_2\epsilon^2\right) + \frac{4L_w^2}{(1-\gamma)^2}\alpha^{\theta^2}U_g^2.$$

After taking expectation on both side, we will arrive at the bound for $I_3$, which is

$$\mathbb{E}\left[\|\hat{w}_{\theta_t}^* - \hat{w}_{\theta_{t+1}}^*\|^2\right] \leq 2\alpha^{\theta^2}L_w^2\left(\mathbb{E}\left[\|\nabla V_{\theta_t}^*\|^2\right] + c_1 \cdot \mathbb{E}\left[\mathcal{N}_{\theta_t}(Q(w_t) - \hat{Q}_{\theta_t}^*)\right] + c_2\epsilon^2\right) + \frac{4L_w^2}{(1-\gamma)^2}\alpha^{\theta^2}U_g^2.$$

Now, go back to Eq.(17) and we obtain

$$
\mathbb{E}\left[||w_{t+1} - \hat{w}^*_{\theta_{t+1}}||^2\right]
$$

$$
\leq \mathbb{E}\left[||w_t - \hat{w}^*_{\theta_t}||^2\right] + \left(\frac{1}{\sqrt{2}}\alpha^\theta L_w + 2\alpha^{\theta^2} L_w^2\right)\mathbb{E}\left[||\nabla V^*_{\theta_t}||^2\right]
$$

$$
+ \left(\frac{1}{\sqrt{2}}\alpha^\theta L_w \left(\frac{1}{\lambda_{\min}} - 2\alpha^w\right) + \sqrt{2}\alpha^\theta L_w \sqrt{c_1\left(\frac{1}{\lambda_{\min}} - 2\alpha^w\right)} + 2\alpha^{\theta^2} L_w^2 c_1 - 2\alpha^w\right) \cdot \mathbb{E}\left[\mathcal{N}_{\theta_t}(Q(w_t) - \hat{Q}^*_{\theta_t})\right]
$$

$$
+ \left(\sqrt{2}\alpha^\theta L_w \sqrt{c_1 C_{w,1} + c_2\epsilon^2\left(\frac{1}{\lambda_{\min}} - 2\alpha^w\right)} + \frac{\alpha^{\theta^2} H_w U_g^2}{\sqrt{2}(1-\gamma)^2}\sqrt{\frac{1}{\lambda_{\min}} - 2\alpha^w}\right)\sqrt{\mathbb{E}\left[\mathcal{N}_{\theta_t}(Q(w_t) - \hat{Q}^*_{\theta_t})\right]}
$$

$$
+ C_{w,1} + C_{w,2} + \frac{1}{\sqrt{2}}\alpha^\theta L_w C_{w,1} + \sqrt{2}\alpha^\theta L_w \sqrt{c_2 C_{w,1}\epsilon^2} + \frac{\sqrt{C_{w,1}}\alpha^{\theta^2} H_w U_g^2}{\sqrt{2}(1-\gamma)^2}
$$

$$
+ 2\alpha^w U_f\left(\frac{\sqrt{2}\alpha^\theta L_w U_g}{1-\gamma} + \frac{\alpha^{\theta^2} H_w U_g^2}{\sqrt{2}(1-\gamma)^2}\right) + 2\alpha^{\theta^2} L_w^2 c_2\epsilon^2 + \frac{4L_w^2}{(1-\gamma)^2}\alpha^{\theta^2} U_g^2.
$$

Now, we can do a telescoping sum for $i$ to $T$:

$$
\left(2\alpha^w - \frac{1}{\sqrt{2}}\alpha^\theta L_w\left(\frac{1}{\lambda_{\min}} - 2\alpha^w\right) - \sqrt{2}\alpha^\theta L_w\sqrt{c_1\left(\frac{1}{\lambda_{\min}} - 2\alpha^w\right)} - 2\alpha^{\theta^2} L_w^2 c_1\right) \cdot \left(\frac{1}{T}\sum_{t=1}^T \mathbb{E}\left[\mathcal{N}_{\theta_t}(Q(w_t) - \hat{Q}^*_{\theta_t})\right]\right)
$$

$$
\leq \frac{\mathbb{E}\left[||w_1 - \hat{w}^*_{\theta_1}||^2\right]}{T} + \left(\frac{1}{\sqrt{2}}\alpha^\theta L_w + 2\alpha^{\theta^2} L_w^2\right)\left(\frac{1}{T}\sum_{t=1}^T \mathbb{E}\left[||\nabla V^*_{\theta_t}||^2\right]\right)
$$

$$
+ \left(\sqrt{2}\alpha^\theta L_w \sqrt{c_1 C_{w,1} + c_2\epsilon^2\left(\frac{1}{\lambda_{\min}} - 2\alpha^w\right)} + \frac{\alpha^{\theta^2} H_w U_g^2}{\sqrt{2}(1-\gamma)^2}\sqrt{\frac{1}{\lambda_{\min}} - 2\alpha^w}\right)\sqrt{\frac{1}{T}\sum_{t=1}^T \mathbb{E}\left[\mathcal{N}_{\theta_t}(Q(w_t) - \hat{Q}^*_{\theta_t})\right]}
$$

$$
+ C_{w,1} + C_{w,2} + \frac{1}{\sqrt{2}}\alpha^\theta L_w C_{w,1} + \sqrt{2}\alpha^\theta L_w \sqrt{c_2 C_{w,1}\epsilon^2} + \frac{\sqrt{C_{w,1}}\alpha^{\theta^2} H_w U_g^2}{\sqrt{2}(1-\gamma)^2}
$$

$$
+ 2\alpha^w U_f\left(\frac{\sqrt{2}\alpha^\theta L_w U_g}{1-\gamma} + \frac{\alpha^{\theta^2} H_w U_g^2}{\sqrt{2}(1-\gamma)^2}\right) + 2\alpha^{\theta^2} L_w^2 c_2\epsilon^2 + \frac{4L_w^2}{(1-\gamma)^2}\alpha^{\theta^2} U_g^2,
$$

which, if we adopt notations from Eq.(13), can be rewritten as

$$
\left(2\alpha^w - \frac{1}{\sqrt{2}}\alpha^\theta L_w\left(\frac{1}{\lambda_{\min}} - 2\alpha^w\right) - \sqrt{2}\alpha^\theta L_w\sqrt{c_1\left(\frac{1}{\lambda_{\min}} - 2\alpha^w\right)} - 2\alpha^{\theta^2} L_w^2 c_1\right) \cdot \Delta_Q
$$

$$
\leq \frac{\mathbb{E}\left[||w_1 - \hat{w}^*_{\theta_1}||^2\right]}{T} + \left(\frac{1}{\sqrt{2}}\alpha^\theta L_w + 2\alpha^{\theta^2} L_w^2\right)\Delta_V
$$

$$
+ \left(\sqrt{2}\alpha^\theta L_w \sqrt{c_1 C_{w,1} + c_2\epsilon^2\left(\frac{1}{\lambda_{\min}} - 2\alpha^w\right)} + \frac{\alpha^{\theta^2} H_w U_g^2}{\sqrt{2}(1-\gamma)^2}\sqrt{\frac{1}{\lambda_{\min}} - 2\alpha^w}\right)\sqrt{\Delta_Q}
$$

$$
+ C_{w,1} + C_{w,2} + \frac{1}{\sqrt{2}}\alpha^\theta L_w C_{w,1} + \sqrt{2}\alpha^\theta L_w \sqrt{c_2 C_{w,1}\epsilon^2} + \frac{\sqrt{C_{w,1}}\alpha^{\theta^2} H_w U_g^2}{\sqrt{2}(1-\gamma)^2}
$$

$$
+ 2\alpha^w U_f\left(\frac{\sqrt{2}\alpha^\theta L_w U_g}{1-\gamma} + \frac{\alpha^{\theta^2} H_w U_g^2}{\sqrt{2}(1-\gamma)^2}\right) + 2\alpha^{\theta^2} L_w^2 c_2\epsilon^2 + \frac{4L_w^2}{(1-\gamma)^2}\alpha^{\theta^2} U_g^2.
$$

$\square$

## F Critic Update Analysis under the i.i.d. Sampling Case

**Lemma F.1.** *In the i.i.d. sampling case,*

$$\left(2\alpha^w - \frac{1}{\sqrt{2}}\alpha^\theta L_w \left(\frac{1}{\lambda_{\min}} - 2\alpha^w\right) - \sqrt{2}\alpha^\theta L_w \sqrt{c_1\left(\frac{1}{\lambda_{\min}} - 2\alpha^w\right)} - 2\alpha^{\theta^2} L_w^2 c_1\right) \cdot \Delta_Q$$

$$\leq \frac{\mathbb{E}\left[||w_1 - \hat{w}_{\theta_1}^*||^2\right]}{T} + \left(\frac{1}{\sqrt{2}}\alpha^\theta L_w + 2\alpha^{\theta^2} L_w^2\right)\Delta_V$$

$$+ \left(\sqrt{2}\alpha^\theta L_w \sqrt{c_1 C_{w,1} + c_2\epsilon^2\left(\frac{1}{\lambda_{\min}} - 2\alpha^w\right)} + \frac{\alpha^{\theta^2} H_w U_g^2}{\sqrt{2}(1-\gamma)^2}\sqrt{\frac{1}{\lambda_{\min}} - 2\alpha^w}\right)\sqrt{\Delta_Q}$$

$$+ C_{w,1} + \frac{1}{\sqrt{2}}\alpha^\theta L_w C_{w,1} + \sqrt{2}\alpha^\theta L_w \sqrt{c_2 C_{w,1}\epsilon^2} + \frac{\sqrt{C_{w,1}}\alpha^{\theta^2} H_w U_g^2}{\sqrt{2}(1-\gamma)^2}$$

$$+ 2\alpha^{\theta^2} L_w^2 c_2\epsilon^2 + \frac{4L_w^2}{(1-\gamma)^2}\alpha^{\theta^2} U_g^2.$$

*Proof.* The i.i.d. assumption implies that $f(O_t, w_t)$ is replaced by $\bar{f}(w_t, \theta_t)$. This will bring a change in both the analysis for $I_1$ and $I_2$.

First, we will figure out how the i.i.d. sampling effect $I_1$. Now we know that $I_{1,3}$ in Eq.(18) is 0. That means, for $I_1$ in Eq.(17),

$$\mathbb{E}\left[||w_t - \hat{w}_{\theta_t}^* + \alpha^w f(O_t, w_t)||^2|\mathcal{F}_t\right] \leq -2\alpha^w \mathcal{N}_{\theta_t}(Q(w_t) - \hat{Q}_{\theta_t}^*) + ||w_t - \hat{w}_{\theta_t}^*||^2$$
$$+ 16\alpha^w(1+\gamma)L_Q H_Q \sigma_w^3 + 4\alpha^w(1+\gamma)L_Q\epsilon\sigma_w + \alpha^{w^2} U_f^2$$
$$= -2\alpha^w \mathcal{N}_{\theta_t}(Q(w_t) - \hat{Q}_{\theta_t}^*) + ||w_t - \hat{w}_{\theta_t}^*||^2 + C_{w,1},$$

where $C_{w,1}$ is defined the same as before:

$$C_{w,1} = 16\alpha^w(1+\gamma)L_Q H_Q \sigma_w^3 + 4\alpha^w(1+\gamma)L_Q\epsilon\sigma_w + \alpha^{w^2} U_f^2.$$

Next, after a removal of $I_{2,2}$ term ($I_{2,2}$ will just be 0 if we replace $f$ by $\bar{f}$), we can derive the new bound for $I_2$, which is

$$\mathbb{E}\left[(\hat{w}_{\theta_t}^* - \hat{w}_{\theta_{t+1}}^*)^T(w_t - \hat{w}_{\theta_t}^* + \alpha^w f(O_t, w_t))\right]$$

$$\leq \frac{1}{\sqrt{2}}\alpha^\theta L_w \mathbb{E}\left[||\nabla V_{\theta_t}^*||^2\right]$$

$$+ \left(\frac{1}{\sqrt{2}}\alpha^\theta L_w\left(\frac{1}{\lambda_{\min}} - 2\alpha^w\right) + \sqrt{2}\alpha^\theta L_w \sqrt{c_1\left(\frac{1}{\lambda_{\min}} - 2\alpha^w\right)}\right) \cdot \mathbb{E}\left[\mathcal{N}_{\theta_t}(Q(w_t) - \hat{Q}_{\theta_t}^*)\right]$$

$$+ \left(\sqrt{2}\alpha^\theta L_w\sqrt{c_1 C_{w,1} + c_2\epsilon^2\left(\frac{1}{\lambda_{\min}} - 2\alpha^w\right)} + \frac{\alpha^{\theta^2} H_w U_g^2}{\sqrt{2}(1-\gamma)^2}\sqrt{\frac{1}{\lambda_{\min}} - 2\alpha^w}\right)\sqrt{\mathbb{E}\left[\mathcal{N}_{\theta_t}(Q(w_t) - \hat{Q}_{\theta_t}^*)\right]}$$

$$+ \frac{1}{\sqrt{2}}\alpha^\theta L_w C_{w,1} + \sqrt{2}\alpha^\theta L_w\sqrt{c_2 C_{w,1}\epsilon^2} + \frac{\sqrt{C_{w,1}}\alpha^{\theta^2} H_w U_g^2}{\sqrt{2}(1-\gamma)^2}.$$

Based on the new bounds for $I_1$ and $I_2$, the critic update now gives the following:

$$\left(2\alpha^w - \frac{1}{\sqrt{2}}\alpha^\theta L_w \left(\frac{1}{\lambda_{\min}} - 2\alpha^w\right) - \sqrt{2}\alpha^\theta L_w \sqrt{c_1\left(\frac{1}{\lambda_{\min}} - 2\alpha^w\right)} - 2\alpha^{\theta^2} L_w^2 c_1\right) \cdot \Delta_Q$$

$$\leq \frac{\mathbb{E}\left[||w_1 - \hat{w}_{\theta_1}^*||^2\right]}{T} + \left(\frac{1}{\sqrt{2}}\alpha^\theta L_w + 2\alpha^{\theta^2} L_w^2\right)\Delta_V$$

$$+ \left(\sqrt{2}\alpha^\theta L_w \sqrt{c_1 C_{w,1} + c_2\epsilon^2\left(\frac{1}{\lambda_{\min}} - 2\alpha^w\right)} + \frac{\alpha^{\theta^2} H_w U_g^2}{\sqrt{2}(1-\gamma)^2}\sqrt{\frac{1}{\lambda_{\min}} - 2\alpha^w}\right)\sqrt{\Delta_Q}$$

$$+ C_{w,1} + \frac{1}{\sqrt{2}}\alpha^\theta L_w C_{w,1} + \sqrt{2}\alpha^\theta L_w\sqrt{c_2 C_{w,1}\epsilon^2} + \frac{\sqrt{C_{w,1}}\alpha^{\theta^2} H_w U_g^2}{\sqrt{2}(1-\gamma)^2}$$

$$+ 2\alpha^{\theta^2} L_w^2 c_2\epsilon^2 + \frac{4L_w^2}{(1-\gamma)^2}\alpha^{\theta^2} U_g^2.$$

$\square$

# G Small Gain Theorem and Small Gain Analysis

## G.1 Small Gain Theorem

Now we introduce the small gain theorem.

**Lemma G.1.** *Suppose $x$ and $y$ satisfy the following two inequalities:*

$$x \leq a_1 y + a_2,$$
$$y \leq b_1 x + b_2 + b_3 \sqrt{y},$$

*where all coefficients are non-negative. Then, $y$ can be upper bounded by the following inequality:*

$$y \leq \frac{2b_2 + b_3^2 + 2a_2 b_1}{1 - 2a_1 b_1}.$$

*Proof.* Proof of this lemma can be found in Olshevsky & Gharesifard (2022). □

## G.2 Small Gain Analysis under i.i.d. Sampling

Now recall the result from Actor analysis is

$$\left(\frac{\alpha_\theta}{2} - \alpha_\theta^2 H_V\right) \Delta_V \leq \frac{\mathbb{E}\left[V_{\theta_1}^* - V_{\theta_{T+1}}^*\right]}{T} + c_1 \left(\alpha_\theta^2 H_V + \frac{\alpha_\theta}{2}\right) \Delta_Q$$
$$+ c_2 \left(\alpha_\theta^2 H_V + \frac{\alpha_\theta}{2}\right) \epsilon^2 + \frac{2H_V}{(1-\gamma)^2} \alpha^{\theta^2} U_g^2.$$

and the one from Critic analysis is

$$\left(2\alpha^w - \frac{1}{\sqrt{2}} \alpha^\theta L_w \left(\frac{1}{\lambda_{\min}} - 2\alpha^w\right) - \sqrt{2}\alpha^\theta L_w \sqrt{c_1 \left(\frac{1}{\lambda_{\min}} - 2\alpha^w\right) - 2\alpha^{\theta^2} L_w^2 c_1}\right) \cdot \Delta_Q$$

$$\leq \frac{\mathbb{E}\left[||w_1 - \hat{w}_{\theta_1}^*||^2\right]}{T} + \left(\frac{1}{\sqrt{2}} \alpha^\theta L_w + 2\alpha^{\theta^2} L_w^2\right) \Delta_V$$

$$+ \left(\sqrt{2}\alpha^\theta L_w \sqrt{c_1 C_{w,1} + c_2 \epsilon^2 \left(\frac{1}{\lambda_{\min}} - 2\alpha^w\right)} + \frac{\alpha^{\theta^2} H_w U_g^2}{\sqrt{2}(1-\gamma)^2} \sqrt{\frac{1}{\lambda_{\min}} - 2\alpha^w}\right) \sqrt{\Delta_Q}$$

$$+ C_{w,1} + \frac{1}{\sqrt{2}} \alpha^\theta L_w C_{w,1} + \sqrt{2}\alpha^\theta L_w \sqrt{c_2 C_{w,1} \epsilon^2} + \frac{\sqrt{C_{w,1}} \alpha^{\theta^2} H_w U_g^2}{\sqrt{2}(1-\gamma)^2}$$

$$+ 2\alpha^{\theta^2} L_w^2 c_2 \epsilon^2 + \frac{4L_w^2}{(1-\gamma)^2} \alpha^{\theta^2} U_g^2.$$

What we really care about is the relationship between $T, \epsilon$ and $m$. So from now on, we will use $O(\cdot)$ and $\tilde{O}(\cdot)$ ($\tilde{O}(\cdot)$ hides the potential logarithm factor of $m$) notations and only consider these variables. First, observe the following dependency on $T, m$ and $\epsilon$:

$$L_Q = O(1),$$
$$H_Q = \tilde{O}(\frac{1}{\sqrt{m}}),$$
$$L_\delta = O(1),$$
$$U_\delta = O(\epsilon) + O(1),$$
$$L_f = \tilde{O}\left(\frac{1}{\sqrt{m}}(\epsilon + 1)\right) + O(1),$$
$$U_f = O(\epsilon) + O(1),$$
$$U_g = O(\epsilon) + O(1),$$
$$L_{\bar{f}} = O\left((\log T + 1)(\epsilon + 1)\right),$$
$$L_{F_\theta} = O\left((\log T + 1)(\epsilon + 1)\right),$$
$$L_{F_w} = \tilde{O}\left(\left(\frac{1}{\sqrt{m}} + 1\right)(\epsilon + 1)\right).$$

If we choose $\alpha^w = \alpha^\theta = \frac{1}{\sqrt{T}}$ and given that all other coefficients are independent with $\epsilon$, $T$ and $m$, we conclude

$$C_{w,1} = \tilde{O}\left(\frac{1}{\sqrt{T}} \cdot \frac{1}{\sqrt{m}}\right) + O\left(\frac{1}{\sqrt{T}}\epsilon\right) + O\left(\frac{1}{\sqrt{T}}\epsilon^2\right) + O\left(\frac{1}{T}\right).$$

We can set

$$x = \Delta_V,$$
$$y = \Delta_Q,$$
$$a_1 = \frac{c_1\left(\alpha_\theta{}^2 H_V + \frac{\alpha_\theta}{2}\right)}{\frac{\alpha_\theta}{2} - \alpha_\theta{}^2 H_V} = O(1),$$
$$a_2 = \frac{\frac{\mathbb{E}\left[V_{\theta_1}^* - V_{\theta_{T+1}}^*\right]}{T} + c_2\left(\alpha_\theta{}^2 H_V + \frac{\alpha_\theta}{2}\right)\epsilon^2 + \frac{2H_V}{(1-\gamma)^2}\alpha^{\theta^2}U_g^2}{\frac{\alpha_\theta}{2} - \alpha_\theta{}^2 H_V} = O\left(\frac{1}{\sqrt{T}}\right) + O\left(\epsilon^2\right),$$
$$b_1 = \frac{\frac{1}{\sqrt{2}}\alpha^\theta L_w + 2\alpha^{\theta^2} L_w^2}{2\alpha^w - \frac{1}{\sqrt{2}}\alpha^\theta L_w\left(\frac{1}{\lambda_{\min}} - 2\alpha^w\right) - \sqrt{2}\alpha^\theta L_w\sqrt{c_1\left(\frac{1}{\lambda_{\min}} - 2\alpha^w\right)} - 2\alpha^{\theta^2} L_w^2 c_1} = O(1),$$
$$b_2 = \frac{\frac{\mathbb{E}\left[||w_1 - \hat{w}_{\theta_1}^*||^2\right]}{T} + C_{w,1} + \frac{1}{\sqrt{2}}\alpha^\theta L_w C_{w,1} + \sqrt{2}\alpha^\theta L_w\sqrt{c_2 C_{w,1}\epsilon^2}}{2\alpha^w - \frac{1}{\sqrt{2}}\alpha^\theta L_w\left(\frac{1}{\lambda_{\min}} - 2\alpha^w\right) - \sqrt{2}\alpha^\theta L_w\sqrt{c_1\left(\frac{1}{\lambda_{\min}} - 2\alpha^w\right)} - 2\alpha^{\theta^2} L_w^2 c_1}$$
$$+ \frac{\frac{\sqrt{C_{w,1}}\alpha^{\theta^2} H_w U_g^2}{\sqrt{2}(1-\gamma)^2} + 2\alpha^{\theta^2} L_w^2 c_2\epsilon^2 + \frac{4L_w^2}{(1-\gamma)^2}\alpha^{\theta^2}U_g^2}{2\alpha^w - \frac{1}{\sqrt{2}}\alpha^\theta L_w\left(\frac{1}{\lambda_{\min}} - 2\alpha^w\right) - \sqrt{2}\alpha^\theta L_w\sqrt{c_1\left(\frac{1}{\lambda_{\min}} - 2\alpha^w\right)} - 2\alpha^{\theta^2} L_w^2 c_1}$$
$$= O\left(\frac{1}{\sqrt{T}}\right) + O(\epsilon) + \tilde{O}\left(\frac{1}{\sqrt{m}}\right),$$
$$b_3 = \frac{\sqrt{2}\alpha^\theta L_w\sqrt{c_1 C_{w,1} + c_2\epsilon^2\left(\frac{1}{\lambda_{\min}} - 2\alpha^w\right)} + \frac{\alpha^{\theta^2} H_w U_g^2}{\sqrt{2}(1-\gamma)^2}\sqrt{\frac{1}{\lambda_{\min}} - 2\alpha^w}}{2\alpha^w - \frac{1}{\sqrt{2}}\alpha^\theta L_w\left(\frac{1}{\lambda_{\min}} - 2\alpha^w\right) - \sqrt{2}\alpha^\theta L_w\sqrt{c_1\left(\frac{1}{\lambda_{\min}} - 2\alpha^w\right)} - 2\alpha^{\theta^2} L_w^2 c_1}$$
$$= O\left(\frac{1}{\sqrt{T}}\right) + O(\epsilon) + \tilde{O}\left(\frac{1}{\sqrt{m}}\right).$$

Now we can apply Small Gain Theorem, where we conclude

$$y \le O\left(\frac{1}{\sqrt{T}}\right) + O(\epsilon) + \tilde{O}\left(\frac{1}{\sqrt{m}}\right).$$

and

$$x \le a_1 y + a_2 = O\left(\frac{1}{\sqrt{T}}\right) + O(\epsilon) + \tilde{O}\left(\frac{1}{\sqrt{m}}\right).$$

### G.3 Small Gain Analysis in the Markov Sampling Case

Now recall the result from Actor analysis is

$$\left(\frac{\alpha_\theta}{2} - \alpha_\theta{}^2 H_V\right)\Delta_V \le \frac{\mathbb{E}\left[V_{\theta_1}^* - V_{\theta_{T+1}}^*\right]}{T} + c_1\left(\alpha_\theta{}^2 H_V + \frac{\alpha_\theta}{2}\right)\Delta_Q$$
$$+ c_2\left(\alpha_\theta{}^2 H_V + \frac{\alpha_\theta}{2}\right)\epsilon^2 + \frac{2H_V}{(1-\gamma)^2}\alpha^{\theta^2}U_g^2,$$

and the one from Critic analysis is

$$\left(2\alpha^w - \frac{1}{\sqrt{2}}\alpha^\theta L_w\left(\frac{1}{\lambda_{\min}} - 2\alpha^w\right) - \sqrt{2}\alpha^\theta L_w\sqrt{c_1\left(\frac{1}{\lambda_{\min}} - 2\alpha^w\right)} - 2\alpha^{\theta^2}L_w^2 c_1\right) \cdot \Delta_Q$$

$$\leq \frac{\mathbb{E}\left[||w_1 - \hat{w}_{\theta_1}^*||^2\right]}{T} + \left(\frac{1}{\sqrt{2}}\alpha^\theta L_w + 2\alpha^{\theta^2}L_w^2\right)\Delta_V$$

$$+ \left(\sqrt{2}\alpha^\theta L_w\sqrt{c_1 C_{w,1} + c_2\epsilon^2\left(\frac{1}{\lambda_{\min}} - 2\alpha^w\right)} + \frac{\alpha^{\theta^2}H_w U_g^2}{\sqrt{2}(1-\gamma)^2}\sqrt{\frac{1}{\lambda_{\min}} - 2\alpha^w}\right)\sqrt{\Delta_Q}$$

$$+ C_{w,1} + C_{w,2} + \frac{1}{\sqrt{2}}\alpha^\theta L_w C_{w,1} + \sqrt{2}\alpha^\theta L_w\sqrt{c_2 C_{w,1}\epsilon^2} + \frac{\sqrt{C_{w,1}}\alpha^{\theta^2}H_w U_g^2}{\sqrt{2}(1-\gamma)^2}$$

$$+ 2\alpha^w U_f\left(\frac{\sqrt{2}\alpha^\theta L_w U_g}{1-\gamma} + \frac{\alpha^{\theta^2}H_w U_g^2}{\sqrt{2}(1-\gamma)^2}\right) + 2\alpha^{\theta^2}L_w^2 c_2\epsilon^2 + \frac{4L_w^2}{(1-\gamma)^2}\alpha^{\theta^2}U_g^2.$$

If we choose $\alpha^w = \alpha^\theta = \frac{1}{\sqrt{T}}$ and given that all other coefficients are independent with $\epsilon$, $T$ and $m$, we conclude

$$C_{w,1} = \tilde{O}\left(\frac{1}{\sqrt{T}} \cdot \frac{1}{\sqrt{m}}\right) + O\left(\frac{1}{\sqrt{T}}\epsilon\right) + O\left(\frac{1}{\sqrt{T}}\epsilon^2\right) + O\left(\frac{1}{T}\right),$$

and

$$C_{w,2} = O\left((\epsilon+1)\frac{(\log T)^2}{T}\right) + \tilde{O}\left((\epsilon+1)\left(\frac{1}{\sqrt{m}}+1\right)\frac{\log T}{T}\right).$$

We can set

$$x = \Delta_V,$$

$$y = \Delta_Q,$$

$$a_1 = \frac{c_1\left(\alpha_\theta^2 H_V + \frac{\alpha_\theta}{2}\right)}{\frac{\alpha_\theta}{2} - \alpha_\theta^2 H_V} = O(1),$$

$$a_2 = \frac{\frac{\mathbb{E}\left[V_{\theta_1}^* - V_{\theta_{T+1}}^*\right]}{T} + c_2\left(\alpha_\theta^2 H_V + \frac{\alpha_\theta}{2}\right)\epsilon^2 + \frac{2H_V}{(1-\gamma)^2}\alpha^{\theta^2}U_g^2}{\frac{\alpha_\theta}{2} - \alpha_\theta^2 H_V} = O\left(\frac{1}{\sqrt{T}}\right) + O\left(\epsilon^2\right),$$

$$b_1 = \frac{\frac{1}{\sqrt{2}}\alpha^\theta L_w + 2\alpha^{\theta^2}L_w^2}{2\alpha^w - \frac{1}{\sqrt{2}}\alpha^\theta L_w\left(\frac{1}{\lambda_{\min}} - 2\alpha^w\right) - \sqrt{2}\alpha^\theta L_w\sqrt{c_1\left(\frac{1}{\lambda_{\min}} - 2\alpha^w\right)} - 2\alpha^{\theta^2}L_w^2 c_1} = O(1),$$

$$b_2 = \frac{\frac{\mathbb{E}\left[||w_1 - \hat{w}_{\theta_1}^*||^2\right]}{T} + C_{w,1} + C_{w,2} + \frac{1}{\sqrt{2}}\alpha^\theta L_w C_{w,1} + \sqrt{2}\alpha^\theta L_w\sqrt{c_2 C_{w,1}\epsilon^2} + \frac{\sqrt{C_{w,1}}\alpha^{\theta^2}H_w U_g^2}{\sqrt{2}(1-\gamma)^2}}{2\alpha^w - \frac{1}{\sqrt{2}}\alpha^\theta L_w\left(\frac{1}{\lambda_{\min}} - 2\alpha^w\right) - \sqrt{2}\alpha^\theta L_w\sqrt{c_1\left(\frac{1}{\lambda_{\min}} - 2\alpha^w\right)} - 2\alpha^{\theta^2}L_w^2 c_1}$$

$$+ \frac{2\alpha^w U_f\left(\frac{\sqrt{2}\alpha^\theta L_w U_g}{1-\gamma} + \frac{\alpha^{\theta^2}H_w U_g^2}{\sqrt{2}(1-\gamma)^2}\right) + 2\alpha^{\theta^2}L_w^2 c_2\epsilon^2 + \frac{4L_w^2}{(1-\gamma)^2}\alpha^{\theta^2}U_g^2}{2\alpha^w - \frac{1}{\sqrt{2}}\alpha^\theta L_w\left(\frac{1}{\lambda_{\min}} - 2\alpha^w\right) - \sqrt{2}\alpha^\theta L_w\sqrt{c_1\left(\frac{1}{\lambda_{\min}} - 2\alpha^w\right)} - 2\alpha^{\theta^2}L_w^2 c_1}$$

$$= O\left(\frac{\tau_{\mathrm{mix}}^2}{\sqrt{T}}\right) + O(\epsilon) + \tilde{O}\left(\frac{1}{\sqrt{m}}\right),$$

$$b_3 = \frac{\sqrt{2}\alpha^\theta L_w\sqrt{c_1 C_{w,1} + c_2\epsilon^2\left(\frac{1}{\lambda_{\min}} - 2\alpha^w\right)} + \frac{\alpha^{\theta^2}H_w U_g^2}{\sqrt{2}(1-\gamma)^2}\sqrt{\frac{1}{\lambda_{\min}} - 2\alpha^w}}{2\alpha^w - \frac{1}{\sqrt{2}}\alpha^\theta L_w\left(\frac{1}{\lambda_{\min}} - 2\alpha^w\right) - \sqrt{2}\alpha^\theta L_w\sqrt{c_1\left(\frac{1}{\lambda_{\min}} - 2\alpha^w\right)} - 2\alpha^{\theta^2}L_w^2 c_1}$$

$$= O\left(\frac{1}{\sqrt{T}}\right) + O(\epsilon) + \tilde{O}\left(\frac{1}{\sqrt{m}}\right).$$

Now we can apply Small Gain Theorem, where we conclude

$$y = O\left(\frac{(\log T)^2}{\sqrt{T}}\right) + O(\epsilon) + \tilde{O}\left(\frac{1}{\sqrt{m}}\right),$$

and

$$x \leq a_1 y + a_2 = O\left(\frac{(\log T)^2}{\sqrt{T}}\right) + O(\epsilon) + \tilde{O}\left(\frac{1}{\sqrt{m}}\right).$$

