# OpenReview forum: "Convergence of Actor-Critic with Multi-Layer Neural Networks"
_NeurIPS.cc/2023/Conference — NeurIPS 2023 poster_

### Official Review · Reviewer_qbM5 · 2023-06-13

**Soundness:** 2 fair
**Presentation:** 2 fair
**Contribution:** 2 fair
**Rating:** 4
**Confidence:** 3

**Summary:**

The paper proposes an analysis of the the actor critic setting with function approximator and it proves the convergence using deep neural networks with an arbitrary number of hidden layers.

**Strengths:**

The claims are ambitious.

**Weaknesses:**

The analysis of the paper sidesteps many key elements from the literature that contradicts the possibility to have a critic that provably converges when using non-linear function approximators, let alone when combined with an actor. When learning with the Bellman iterations, a compounding of errors can occur due to the non-linearity of the function approximator with respect to the parameters, which can lead to divergence even with the continuity and Lipschitz assumptions as described in the paper.

The discussion from Section 4.2 is also not convincing.

Additional comments:
- line 77: the reward function goes into $R$, but what is R? Did the authors mean the real numbers $\mathbb R$?
- Many discussion points lack a precise formalization, e.g. line 248: "Such an analysis is inherently more technically challenging, since when the actor can wait for the critic to go through sufficiently many iterations, one could argue that the resulting Q-values are approximately accurate and the process resembles gradient descent."

**Questions:**

Why do the examples of off-policy divergence not apply in your analysis (see for instance Sutton and Barto intro to RL book in Section 11.2 "Examples of Off-policy Divergence").

**Limitations:**

Limitations are not really discussed and it might be that some of the claims (see questions above) are not correct.

---

> ### Author Rebuttal · Authors · 2023-08-03
>
> We welcome the opportunity to contextualize our results with respect to the earlier literature showing examples of divergence with non-linear function approximations. However, we do want to note to the reviewer that they misunderstand the state of this research area. Please see our responses below.
>
> > The analysis of the paper sidesteps many key elements from the literature that contradicts the possibility to have a critic that provably converges when using non-linear function approximators...
>
> There is no such literature.
>
> What does exist are examples that show that nonlinear approximation *can* result in divergence. The earliest such example that we are aware of comes from the classic 1997 paper of Tsitsiklis & Van Roy. But this does not mean that RL with nonlinear approximations *always* diverges. This paper is just the latest of a long line of literature which develops progressively more realistic conditions under which various RL methods willl converge with nonlinear approximations.
>
> > When learning with the Bellman iterations, a compounding of errors can occur due to the non-linearity of the function approximator with respect to the parameters, which can lead to divergence even with the continuity and Lipschitz assumptions as described in the paper.
>
> In the first part of this sentence, you have correctly described the main difficulty that any analysis of nonlinear approximation needs to overcome.
>
> But as to the second part of your sentence, note that we have assumed more than just continuity and Lipschitzness. Among other assumptions, it is crucial for our results that the neural network is *randomly initialized*. We also make a quantitative version of an assumption that the underlying neural network maps different inputs to different outputs, which we support with simulations (see our global response).
>
> > Why do the examples of off-policy divergence not apply in your analysis (see for instance Sutton and Barto intro to RL book in Section 11.2 "Examples of Off-policy Divergence").
>
> Note that the version of actor-critic we consider in this paper is on-policy: after each actor update, a sample is generated using the new actor policy, and the critic takes a step using this new sample. So examples of divergence for off-policy TD are not relevant here.
>
> It may be useful to quote from Section 11.5 of Sutton and Barto:``Semi-gradient methods may diverge under off-policy training, as we have seen earlier in this chapter, *and under contrived cases of nonlinear function approximation (Tsitsiklis and Van Roy, 1997)*'' (emphasis ours). The whole point of this paper is to come up with conditions that rule out such "contrived cases" of nonlinear approximation.
>
> > The discussion from Section 4.2 is also not convincing
>
> If you are still not convinced that our results do not contradict existing work, please state precisely which specific theorem from which paper or textbook is violated by our main results.
>
> It is perhaps worth spelling out a couple of critical ingredients in our work that enable us to obtain the result we did. First is that the critic runs not TD, but projected TD; the projection is on a set of fixed diameter, and this is important in terms of stabilizing the updates. Second, observe that nowhere in our paper do we have any result to the effect that either the critic or the actor parameters converge; instead, the *quality of the approximation* given by those parameters is bounded by
> $$ O \left( \frac{1}{\sqrt{T}} + \epsilon^2 + \frac{1}{\sqrt{m}}\right),$$ where $T$ is the number of steps, $\epsilon$ is the best possible approximation quality of the neural architecture, and $m$ is the neural network width. Here "quality of approximation" means both how well the critic is able to approximate the true value function of the actor's policy, and how well the actor is able to find a point where the gradient of its policy is zero.
>
> It is entirely consistent with our results to have the parameters wander within the domain without convergence, as long as all the quality of the approximation given by the neural network remains good. For example, there may be a region where the gradient of the actor's policy is very close to zero, and nothing in our results prevents the actor's parameters from wandering within that region.
>
>
> *With all this in mind, we urge you to re-examine the contribution of our work.* Please do not hesitate to use OpenReview to contact us during the author-reviewer interaction period.
>
> > line 77: the reward function goes into R, but what is R? Did the authors mean the real numbers ?
>
> Thanks for pointing out this typo! Yes here it should be the real numbers. We will change it to \mathcal{R}.
>
> > Many discussion points lack a precise formalization, e.g. line 248: "Such an analysis is inherently more technically challenging, since when the actor can wait for the critic to go through sufficiently many iterations, one could argue that the resulting Q-values are approximately accurate and the process resembles gradient descent."
>
> The whole point of that section is to explain informally the challenges faced by our analysis.

---

> > ### Comment · Reviewer_qbM5 · 2023-08-14
> > **Not convinced at this point**
> >
> > In the context of "on-policy" learning, finite state space, finite action space, what is the objective of making use of neural networks if it's not to make use of its generalization capabilities? If it does use the generalization capabilities, then there is a possibility that the algorithm does not converge.
> >
> > Additional remarks:
> > Reviewer akFw mentions that (s)he has "little background on convergence of actor-critic methods or in analyses of deep networks" (while putting a 5 in confidence).
> > Reviewer ADfc has a confidence of 1.
> > Reviewer FwfS explicitly mentions that (s)he didn't work on this topic and didn't check the proofs.

---

> > > ### Author Response · Authors · 2023-08-14
> > >
> > > > In the context of "on-policy" learning, finite state space, finite action space, what is the objective of making use of neural networks if it's not to make use of its generalization capabilities? If it does use the generalization capabilities, then there is a possibility that the algorithm does not converge.
> > >
> > > It sounds like if you are referring to some kind of theorem here (presumably one that says good generalization implies divergence). If so, **please state precisely what theorem you are referring to.**
> > >
> > > We do not know which result is being referred to here: if you tell us, we'll be able to comment on how it does or does not apply to the settign considered in this paper. Even referring to "generalization" in the context of RL requires more detail since there are a number of competing formalizations of this concept in the literature (as opposed to supervised learning, where there is almost universal agreement on one set of definitions for generalization).

---

> > > > ### Author Response · Authors · 2023-08-21
> > > >
> > > > Since the reviewer did not respond to our request for clarification (made 7 days ago) and this is the last day of the author-reviewer interaction period, we will attempt to answer the concerns raised to the best of our ability. Without knowing exactly what the reviewer meant, we will make the best possible guesses we can.
> > > >
> > > > As for the "good generalization implies divergence" claim that the reviewer seems to implicitly be making: as far as we know, this is false even for supervised learning. For example, SGD with $\sim 1/t$ step-size on a strongly convex objective converges with probability one and yet has generalization error inversely proportional to dataset size, see Theorem 3.10 of the arxiv (not published) version of the paper "Train Faster, Generalize Better: Stability of Stochastic Gradient Descent," by Hardt, Recht, and Singer, 2016 for the generalization bound, and ``On the Almost Sure Convergence of Stochastic Gradient Descent in Non-Convex Problems, '' Mertikopoulos, Hallak, Kavis, Cevher, NeurIPS 2020 for the convergence result (this paper is for non-convex objective but gives convergence to the optimal point when specialized to strongly convex functions -- happy to provide more details here if needed). In the context of RL, we're not even sure how to interpret the claim about generalization/divergence in question.
> > > >
> > > > We will address three points present in the reviewer's question:
> > > >
> > > > (1) Why do practitioners use neural networks?
> > > >
> > > > (2) Why do we use neural networks?
> > > >
> > > > (3) Is generalization relevant to our paper?
> > > >
> > > > The answer to **(1)** is that neural networks are a practical solution to the feature selection problem. When tabular methods cannot be applied due to large state space, we need to set a policy based on features, but we do not a-priori know what the right features are. Neural networks are often successful in practice because they are able to learn well-performing features during training.
> > > >
> > > > As to **(2)**, we study neural networks because we want to develop a theory that helps practitioners. This paper is a step towards that goal.
> > > >
> > > > As to **(3)** the answer is no, we cannot see how a discussion of generalization would be relevant for this paper. Indeed, we're broadly aware of two senses generalization is used in RL. One is used in the off-line case, when you train a policy on someone else's dataset and need to guarantee you have not been over-fitting. But our algorithm is on-line.
> > > >
> > > > The second sense that generalization is sometimes used in RL is in the sense of generalization across environments (e.g., training a policy on lunar lander  that will still work once you change various environment parameters such as the force of gravity). But we focus only on convergence results when you train a policy on a single MDP.

---

> > > > > ### Comment · Reviewer_qbM5 · 2023-08-21
> > > > > **Thanks for the reply+additional clarification on previous comment**
> > > > >
> > > > > Thanks for the tentative of clarifications.
> > > > >
> > > > > For a given actor (policy) and a given batch of tuples obtained in the environment, it might be that in expectation over the whole state-action space you don't get closer to converging for the critic by applying the Bellman iterations given that an update can have unexpected effects on the whole state-action space (in particular outside of the tuples that are in the batch). Yet, your claim is that even if on top of that there is an inter-play between actor and critic, you can ensure convergence with neural networks. This claim, if true in some specific contexts, would be interesting.
> > > > >
> > > > > The main elements to check the validity of the only theorem of the paper are in the appendix (30 pages long). I actually tried looking at that because I don't think the claim would hold in the stated hypotheses but it is however difficult to navigate the appendix because nothing in the paper provides hints about the most important aspects. In particular:
> > > > > 1. it is to me a bit unclear how the different lemmas/theorems fit together to reach the main theorem, and
> > > > > 2. (slightly more minor) we don't know what is original (your own results) and what is not.
> > > > >
> > > > > Additional comment: In the main paper, the statement "the number of states is *typically* many magnitudes larger than the number of parameters in the critic" is not very clear given that there are still quite a few RL domains where the state space are a few discrete features or can be discretized in a reasonable number of discrete states (NB: the number of parameters in NNs can also be at least in the millions). This sentence is followed by "Thus Q(w) *will* map $w$ to a much higher dimensional space." Besides the logical problem of using the word typically followed by a general statement, why is this of any importance? Would the results not hold otherwise?

---

> > > > > > ### Author Response · Authors · 2023-08-21
> > > > > >
> > > > > > Our point-by-point replies are given below.
> > > > > >
> > > > > > > For a given actor (policy) and a given batch of tuples obtained in the environment, it might be that in expectation over the whole state-action space you don't get closer to converging for the critic by applying the Bellman iterations given that an update can have unexpected effects on the whole state-action space (in particular outside of the tuples that are in the batch).
> > > > > >
> > > > > > This is not quite right. To see this, consider first the case of TD(0) with *linear* approximation. In that case, it is also true that an update can have unexpected effects on the whole state-action space, in particular outside of the tuples that are in the batch. And yet you **can** then argue that you get closer to your final limit at each step.
> > > > > >
> > > > > > Indeed, the key is to pick the right metric to measure what it means to be "close": a good metric for this is given in Eq. (12) in our paper (and note that this insight is not ours, but was shown in [Liu & Olshevsky, ICML 2021], cited in our paper). Specifically, in the TD(0) case the quantity that, in expectation, gets closer to the limit is
> > > > > >
> > > > > > $$ V(\omega_t) = (1-\gamma) ||V(\omega_t) - V(\omega^*)||^2 + \gamma ||V(\omega_t) - V(\omega^*)||^2$$
> > > > > >
> > > > > >  where $\omega^*$ is the TD(0) fixed point (which exists in the linear case) and the first norm should be a Dir norm while the second norm above should be a D-norm (defined in our paper; openreview will not let us put subscripts). More precisely, we have that
> > > > > >
> > > > > > $$ E [ V(\omega_{t+1})  | \omega_{t} ] < V(\omega_t),$$
> > > > > >
> > > > > > provided once again we are in the setting of TD(0) -- not AC -- and the approximation is linear (and the step-size is small enough).
> > > > > >
> > > > > > **tldr:** This discussion shows that the connection between "updates have unexpected effects outside the batch" and "you don't get closer to converging [in expectation]" is not there because in the linear case the first of these statements hold but not the second. Of course, to make things work for AC and with neural networks on top of this requires significant modifications, but hopefully this explains one of the core ideas.
> > > > > >
> > > > > > > Yet, your claim is that even if on top of that there is an inter-play between actor and critic, you can ensure convergence with neural networks.
> > > > > >
> > > > > > Almost -- we do not show that the parameters converge. Instead, we show that the performance of the underlying neural networks is good, in a precise sense discussed both in the paper and in our initial rebuttal to you. Briefly, the goal is for the gradient of the actor's policy to be zero; instead, our main result says that the running average of the gradients is upper bounded by something that scales linearly in $\epsilon$ (approximation capability of critic) and inversely with $T$ (number of iteration) and $m$ (the neural network width).
> > > > > >
> > > > > > > In the main paper, the statement "the number of states is typically many magnitudes larger than the number of parameters in the critic" is not very clear given that there are still quite a few RL domains where the state space are a few discrete features or can be discretized in a reasonable number of discrete states (NB: the number of parameters in NNs can also be at least in the millions).
> > > > > >
> > > > > > If the number of states is "few" or "reasonable" as stated in your reply, then why not just use a tabular method?
> > > > > >
> > > > > > Indeed, recall that our ultimate goal is to develop a theory that aids practitioners who use neural networks. But this is only done when a tabular method *cannot* be used, i.e., when the number of states is prohibitively large.
> > > > > >
> > > > > > In particular, if the state-space was only in the millions, a tabular method would be feasible. For the kind of scenarios that necessitate the use of neural networks, consider any situation where the state is an image (e.g., whenever RL is used for Atari games the state-space is typically taken to be the last 4 images of the game state). Then the number of possible states is astronomical. Indeed, the number of possible even tiny 60x40 images is $256^{60 \times 40}$.
> > > > > >
> > > > > > > Besides the logical problem of using the word typically followed by a general statement, why is this of any importance? Would the results not hold otherwise?
> > > > > >
> > > > > > First: we don't quite see the logical problem. "Typically people use umbrellas when it rains" -- there is nothing wrong with this statement. Certain parts of our paper are very formal, whereas others are informal, and this assertion comes from an informal part where we discuss the intuition behind why Assumption 2.7 holds in practice (an assertion which is supported by simulations -- see our global rebuttal).
> > > > > >
> > > > > > Second: as to your question, the results require Assumption 2.7, and indeed they would not hold without it. So the discussion motivating Assumption 2.7 *on an intuitive level* is important to include.

---

> > > > > > > ### Author Response · Authors · 2023-08-21
> > > > > > >
> > > > > > > > I actually tried looking at that because I don't think the claim would hold in the stated hypotheses
> > > > > > >
> > > > > > > Let us try to persuade you otherwise.
> > > > > > >
> > > > > > > First, we all know that practitioners use neural networks all the time and get good performance out of them. And yet textbooks are full of examples of divergence, even in the case of TD(0). It therefore stands to reason that we *just need the right theorem* -- a theorem that will identify the correct set of conditions that are satisfied in practice that would rule out the "contrived examples" (using the language of Sutton & Barto) of divergence.
> > > > > > >
> > > > > > > We believe we have provided an important piece of this: the three key ingredients that ensure good performance are random initialization, not letting the neural network drift too far from initialization, and Assumption 2.7, which we support with simulations.
> > > > > > >
> > > > > > > Second: there already are results in the previous literature that prove similar stuff for neural networks with one hidden layer (under some more restrictive conditions). So if you are skeptical about correctness, your skepticism should be based on some property that neural networks with multiple hidden layers have that they do not have with only one hidden layer. While there are many potential such properties, we do not believe any of them are relevant to getting convergence for AC; rather it is only that it is more difficult to reason about deeper neural networks.
> > > > > > >
> > > > > > > > but it is however difficult to navigate the appendix because nothing in the paper provides hints about the most important aspects. In particular: it is to me a bit unclear how the different lemmas/theorems fit together to reach the main theorem, and
> > > > > > >
> > > > > > >
> > > > > > > Please see https://openreview.net/forum?id=QlfGOVD5PO&noteId=eRJPP3ZLh7 which links to this page but to our reply to a different reviewer, where we provided a sketch of our proof. But indeed, the proof is not only long but contains many estimates which have to be put together in just the right way.
> > > > > > >
> > > > > > >
> > > > > > > > (slightly more minor) we don't know what is original (your own results) and what is not.
> > > > > > >
> > > > > > > This paper is based on a fusion of three ideas from the earlier literature:
> > > > > > >
> > > > > > > (1) The use "gradient splitting" to argue that a certain Lyapunov function decreases at each step in linear TD. This is from [Liu & Olshevsky, ICML 2021].
> > > > > > >
> > > > > > > (2) The Hessian of a neural network in a bounded region of a random initial condition has norm $O(1/m)$, where $m$ is the width of the layers. This fact is from [Liu et al, NeurIPS 2020] and was used earlier to analyze neural TD in [Tian et al, ICLR 2022]. The key idea of that work is to treat neural TD as inexact gradient descent.
> > > > > > >
> > > > > > > (3) The introduction of an auxiliary chain to handle the Markovian transitions. This is inspired by  Zou et al. (2019); Wu et al. (2020); Chen & Zhao (2022).
> > > > > > >
> > > > > > > Our contribution was to put together these three techniques and carefully handle additional error terms that arise from using neural approximations for both actor and critic. This was quite challenging because, speaking informally, the actor and critic errors influence each other; if the critic is more wrong, the actor can take steps that will not bring it to a region of low gradients, and the constant evolution of actor parameters will then make sure the critic parameters do not converge.  This kind of "feedback" is why analysis of AC is much more challenging than analysis of TD.
> > > > > > >
> > > > > > > As should be evident by glancing at Table 1 and then reading the discussion immediately following the statement of Theorem 3.1 in our paper, what comes out of successfully executing on this plan is **much** stronger compared to previous work analyzing neural AC. We hope that this discussion helped the reviewer clarify both the high level structure of our paper, as well as the ideas from the previous literature we have used.

---

> > > > > > > > ### Comment · Reviewer_qbM5 · 2023-08-22
> > > > > > > > **Thanks for the clarifications**
> > > > > > > >
> > > > > > > > Thanks for the clarifications. This information clears some of my concerns.
> > > > > > > >
> > > > > > > > Overall I can't rule out that this paper makes an interesting contribution but given the presentation and the highly technical nature of the paper I also can't fully certify that the contribution is meaningful (also because it is not my main area of expertise). The paper itself is quite high-level (e.g. pages 3 and 4 restate well-known "basic" elements of RL and function approximators) while the appendix is highly technical without any clear plan to make it approachable (even though I appreciate that the rebuttal improves a bit on that respect). I will increase my score to 4 and keep my confidence at 3.

---

### Official Review · Reviewer_oPsP · 2023-07-04

**Soundness:** 3 good
**Presentation:** 2 fair
**Contribution:** 2 fair
**Rating:** 4
**Confidence:** 4

**Summary:**

The paper studies non-asymptotic convergence rate of actor-critic algorithm. The critic network is parameterized by multi-layer neural network whose activation function is assumed to be smooth, excluding ReLU operator. The actor is assumed to be general smooth non-linear function.

 The convergence rate for mean squared error of gradient of value function and $\mathcal{N}$ norm of Q--unction is proved to be $\tilde{O}(1/T^{1/2})$ with additional error term $\epsilon^2+O(1/m^{1/2})$.

**Strengths:**

The algorithm uses single-step size and constant projection radius, which is closer to practice than two-time scale step size and projection radius being dependent on the neural network width $m$. Moreover, the depth of the neural network can be chosen arbitrary. Overall, the authors managed to derive the convergence rate for Neural AC combining the works of Tian et al., and Olshevsky et al..

**Weaknesses:**

The main weakness is that the work seems to simply combine the works of Tian et al. and Olshevsky et al. It is difficult to clarify what are the challenges and contributions of combining Tian and Olshevsky et al..

Moreover, the limitation of this work is that it considers smooth activation excluding the ReLU activation function.

In key ideas ( Section 4 ), is the small gain theorem different from that of Olshevsky et al.? If it is different, there should be comment on what's different, and if not, please add citation for it. Moreover, I believe the title of section 4 is not appropriate since both ideas are from Liu et al. and Olshevsky et al.. It would be better to explain the difficulties in combining the existing works.

- Miscelleneous

There is grammar error in line 28, "was consider->was considered"

In line 99, please add citation for the policy gradient theorem.

In line 304, pleas add reference for the textbook.

In line 597, please write as a complete sentence.


**Questions:**

Regarding Assumption 2.6, is such assumption on the critic approximation standard in the literature of finite time analysis of actor critic? If so, please provide some comments and citations in the paper.

In line 384, please give more detail why the state action pair is sampled from $(1-\gamma)\phi_{\theta_t}$.

In line 572, what is the motivation to introduce $ w_{\mathrm{mid}} $? I guess it allows the decomposition of the term but what meaning does each term have?


**Limitations:**

The strength that the authors argue that the analysis does not require initialization point being close to the solution, and the projection radius not dependent on the width of neural network, seems to be direct result from Tian et al., rather than being a new discovery. Moreover, the general framework of actor-critic analysis is adopted from Olshevsky et al. It is not clearly explained what the difficulties and challenges of the analysis combininng Tian et al. and Olshevsky et al.. Hence, I am currently leaning towards rejection.


Haoxing Tian, Ioannis Paschalidis, and Alex Olshevsky. On the performance of temporal difference learning with neural networks. In The Eleventh International Conference on Learning Represen- tations, 2023.

Alex Olshevsky and Bahman Gharesifard. A small gain analysis of single timescale actor critic. arXiv preprint arXiv:2203.02591, 2022.

---

> ### Author Rebuttal · Authors · 2023-08-04
>
> > The main weakness...
>
> Our contribution is not a simple combination of two papers. Even superficially, note that our paper deals with actor-critic, yet our Lyapunov function is not the same as Olshevsky et al. (which relied on square norm). Further, Olshevsky et al. was for the i.i.d. sampling case, not the more difficult Markovian sampling case considered here. Tian et al. considered Markov sampling, but Markov sampling in AC is much difficult than that in TD(0) since in AC the policy changes from step to step.
>
> Let us outline more substantively the difficulty overcome by this paper. The use of Markov sampling, and the use of multi-layer neural networks, introduces additional errors into the analysis. Whereas Olshevsky et al. was based on the analogy between actor-critic and bilevel optimization, here we have an update that can be analogized to bilevel optimization **with errors**. Dealing with this is not straightforward. In the context of AC, the main difficulty is showing the errors do not compound, but are instead carried over into the final bound. To see the difficulty, imagine the critic makes an update with errors. Those errors are then used by the actor to move in the direction of a non-optimal policy. The actor then makes a step in the direction of the new non-optimal policy, with an additional error. It is easy to see how this effort could result in the algorithm's divergence.
>
> Moreover, the situation here is fundamentally different than what we have Tian et al. which was inspired by gradient descent tools. In that case, it is simpler to show that errors do not compound since gradient descent does not have the same feedback loop as above.
>
> It is well-understood in the optimization literature that such an error analysis can be highly nontrivial, and there are quite a few papers whose main contribution is an analysis of a standard optimization method with errors. Note that in our case, we show that the source of error that comes from the neural network enters *linearly* into the final objective ($1/\sqrt{m}$, which is a bound on "how nonlinear" the neural network can get, where m is the width), as opposed to the *quadratic* way that epsilon enters into the final bound. This difference shows a nontrivial way in which the source of the errors, which governs how the errors enter into the analysis, matters.
>
> > Moreover, the limitation...
>
> Indeed, we do need the smoothness of the input-to-output map as we will use the result from Liu et al. (2020), which claims the neural network is $O(1/\sqrt{m})$-smooth with respect to its parameters. However, many activation functions are twice differentiable (e.g., sigmoid, tanh, arctan, softplus) and one could always use a smooth approximation to a ReLU activation (e.g., GeLU or ELU).
>
> > In key ideas Section 4...
>
> The small-gain theorem is the same as the one used in Olshevsky et al.
>
> > It would be better to explain the difficulties in combining the existing works.
>
> As discussed above, one of the difficulty lies in Markov sampling in AC. The idea is simple, which is to assume uniform mixing and show the difference to mean-path update is geometrically decreasing based on the distance to stationary distribution is geometrically decreasing. However, this problem, which is easy to address, as shown in Bhandari et al. (2018), is difficult in AC, since the policy changes from step to step. Inspired by Zou et al. (2019); Wu et al. (2020); Chen & Zhao (2022), we introduce an auxiliary chain as the bridge from the current policy to stationary distribution.
>
> Another difficulty lies in the generalization from single-layer neural network to multi-layer neural network. Our goal is not including linearization about initialization but using the Hessian of the neural network in our paper, as the only indicator on whether the neural network in close to a linear approximator. Though it is also addressed in Tian et al., in AC we have two sets of parameters, and they are actually coupled together. Decoupling these updates is a challenging work, and that is the reason why Assumption 2.6 comes in.
>
> > Regarding Assumption 2.6...
>
> Assumption 2.6 is crucial for our analysis because it enables us to decouple the critic parameter $\omega$ and the actor parameter $\theta$. As far as we know, this assumption is not included in any previous papers. This is likely the case for two reasons. First, for those of papers which only consider linear function approximation, the relationship as mentioned in Assumption 2.6 is also linear. Second, for those papers which consider neural network function approximator, they perform analysis in a double-loop setting, where the actor update is not performed until the critic network converges. Clearly, in this case the critic parameter $\omega$ and the actor parameter $\theta$ are *naturally decoupled* and the error term from both network also can be treated separately. Our result, to our best knowledge, is the first one that combines single-loop AC and a neural network (and we even generalize to multi-layer neural network), which requires Assumption 2.6.
>
> > In line 384...
>
> Thanks for your question. We now make it more clear why the state action pair follows $(1-\gamma)\phi$ in Appendix A. Basically, by total probability, $P(S = s) = \sum_{t=0}^{+\infty} P(S = s|T = t) \cdot P(T = t)$. The first term on the right hand side is exactly $P(S_t = s)$ and the second term equals to $(1-\gamma)\gamma^t$. We provide a pdf that contains a detailed proof.
>
> > In line 572 ...what meaning does each term have?
>
> We're not sure there's a clean intuition here, i.e., a meaning for each term. In that line, we need to exploit the fact that the function will become more and more linear when the width of neural network $m$ goes larger and larger, and the application of the mid-value theorem appears in one of the steps of this.

---

> > ### Comment · Reviewer_oPsP · 2023-08-14
> >
> > Dear authors, thank you for your rebuttal. However, my concerns have not been addressed. Even though, different Lyapunov function is used from Olshevsky et al., still the key ideas are from Olshevsky et al., e.g., bound on the critic error ( Lemma B.18 ) and the small-gain theorem. Moreover, the techinque to bound Markovian noise seems to resemble that of Wu et al.. Regrading the bound on the critic, since the policy network is not multi-layer neural network, the result in Tian et al. seems to follow without significant difficulties. Overall, such points limit the novelty of the analysis. Hence, as for now, I maintain my score.

---

> > > ### Author Response · Authors · 2023-08-14
> > >
> > > > Regrading the bound on the critic, since the policy network is not multi-layer neural network, the result in Tian et al. seems to follow without significant difficulties.
> > >
> > > *This is not correct: our policy network can be a multi-layer neural network.* Please see Assumption 2.4 for the only requirements we make on the policy.
> > >
> > > [And note that this assumption is weaker than what is in Olshevsky & Gharesifard and and Wu et al., since it does not assume smoothness (i.e., that the gradient is Lipschitz) of either $\pi$ or $\ln \pi$]
> > >
> > > >  Moreover, the techinque to bound Markovian noise seems to resemble that of Wu et al..
> > >
> > > Note that we acknowledged these influences in our previous reply. More broadly, every paper relies on a combination of earlier techniques which are put together in a new way with some kind of new ingredients. It is not unfair to say that our work is a combination of techniques from Olshevsky & Gharesifard, Tian et al., and Wu et al., coupled together with a careful error analysis to bound additional terms that arise from approximations that were not present in earlier works.  Combining different techniques in this way is nontrivial, required a change in the underlying Lyapunov function, and overcomes important limitations from the earlier literature (our paper is the first to handle arbitrary number of layers, first to do single timescale AC with neural networks, and avoids the common limitation of having a projection radius that scales inversely with $m$).

---

> > > > ### Comment · Reviewer_oPsP · 2023-08-15
> > > >
> > > > Even though the policy network can be modeled as multi-layer neural network, the authors do not show, if policy network is modeled as multi-layer, the effect of depth and width of neural network could have.  Moreover, since the limitations that authors have tackled, handling arbitrary number of layers and projection radius being independent of $m$, has been shown in Tian et al. for TD-learning, the result is not really surprising . However, I agree with the author's point that combining different techniques still can be a contriubtion. As for now, I will increase my score from 3 to 4.

---

> > > > > ### Author Response · Authors · 2023-08-15
> > > > >
> > > > > Thank you for being willing to update your rating!
> > > > >
> > > > > And you are absolute right that the effect of width of the actor is not explored here. Indeed, there is the potential for a follow-up paper which exploits this to obtain stronger results under more restrictive assumptions. Let us denote by $m'$ the width of the actor network (with a prime to distinguish from width of the critic network). Then one could argue that one should be able to establish a certain notion of global optimality for the actor: the ultimate performance of the actor should be upper bounded by the best possible performance of the actor in a ball of constant radius around the initial point, plus an error term which scales inversely with $m'$.
> > > > >
> > > > > But this would require a different analysis on the actor-side and possibly a different small-gain theorem and would ultimately be more restrictive than the current paper (which also allows the actor to be a transformer or some other architecture). It may also require the introduction of a projection for the actor parameters.

---

### Official Review · Reviewer_7mQ7 · 2023-07-06

**Soundness:** 3 good
**Presentation:** 4 excellent
**Contribution:** 4 excellent
**Rating:** 7
**Confidence:** 3

**Summary:**

The authors prove a bound on the approximation error of sample-based actor critic learning on a more general and realistic setting, namely allowing for a NN approximator of any depth. This is a very non-trivial result and an interesting contribution to the literature.

Note: I have not gone through the full proof in the appendix and thus cannot fully comment on the validity of the main contributions.

**Strengths:**

The paper is very well written and the ideas are clearly communicated. I especially appreciate the tables and figures to aid in explaining the approach and how it fits into the literature.

The main theorem is a very interesting theoretical result.

**Weaknesses:**

There are a few parts of the paper where the presentation could be improved a bit, but for the most part I thought the paper was very clearly written.

**Questions:**

Your final bound does not appear to be affected by the depth of the NN approximator, but clearly depth does improve approximation error. Does a tighter bound exist that takes into account depth or would depth mostly be captured in the \epsilon error term?

On line 135, you refer to \sigma_w, but I don't see that variable defined. What is that referring to?

In Assumption 2.6, what is the gradient of w with respect to?

Small notes:
I would recommend having all equation statements be numbered so it's easier for readers to refer to them.

I find the notation on the equation in line 161 a bit confusing, namely that Q refers to both a function that takes in weights and outputs a Q-function and the Q-function itself.



**Limitations:**

I don't think there are significant potential negative societal impacts of this work.

---

> ### Author Rebuttal · Authors · 2023-08-04
>
> > The paper is very well written and the ideas are clearly communicated. I especially appreciate the tables and figures to aid in explaining the approach and how it fits into the literature.
>
> > The main theorem is a very interesting theoretical result.
>
> Thank you for the encouraging review!
>
>
> > Your final bound does not appear to be affected by the depth of the NN approximator, but clearly depth does improve approximation error. Does a tighter bound exist that takes into account depth or would depth mostly be captured in the $\epsilon$ error term?
>
> Indeed, that is correct: the dependence on depth is implicitly captured in the $\epsilon$ term.
>
> > On line 135, you refer to $\sigma_w$, but I don't see that variable defined. What is that referring to?
>
> We'll make this clearer in the final version of the paper; line 135 here is actually supposed to be the definition of $\sigma_w$. That is, we can choose the projection radius $\sigma_w$ arbitrarily. This projection radius will appear as a constant in the various $O(\cdot)$ notations throughout the paper.
>
> > In Assumption 2.6, what is the gradient of w with respect to?
>
> We'll make this clearer in the final version of the paper as well: in Assumption 2.6, we view $\hat{\omega}_{\theta}^*$ as a function of $\theta$, with respect to which we take the gradient.
>
> > I find the notation on the equation in line 161 a bit confusing, namely that Q refers to both a function that takes in weights and outputs a Q-function and the Q-function itself.
>
> Thanks for pointing out our typo there! Indeed it should be $Q_{\theta}^*$ instead of $Q_{\theta}$. Throughout the paper, we use $\theta$ to denote the parameters of the policy, while $Q_{\theta}^*$ denote the true value functions corresponding to policy $\pi(\theta)$.

---

### Official Review · Reviewer_FwfS · 2023-07-07

**Soundness:** 4 excellent
**Presentation:** 3 good
**Contribution:** 4 excellent
**Rating:** 7
**Confidence:** 3

**Summary:**

This is a theoretical paper which studies actor-critic reinforcement learning algorithms in which both the actor and the critic are represented using deep neural networks with more than one hidden layer. The previous research addressed linear representations and neural networks with only one hidden layer. This paper derives convergence rates for both the actor and the critic for neural networks with more than one hidden layer.


**Strengths:**

- Writing is of very high quality, with well-considered notation, and sensible flow.

- The problem statement is clear, and it is supported by required references.

- The problem is important, and the paper furthers our understanding of the behaviour of RL with function approximation using deep neural networks.


**Weaknesses:**

- A few small typos can be found in the paper. E.g., in line 28 "was consider in". A few articles are also missing in various places.

- In the paper, the authors emphasize the need to stay close to the initial conditions of the critic (e.g. in Sec. 2.4). This makes sense from the regularisation point of view in general, but in RL, this may mean that the optimial policy may not be found if the algorithm is forced to say close to the initial conditions. Perhaps the motivation and the consequences of staying close to the initial conditions could be clarified.

- Consider Eq. (9), and assume that $\epsilon$ is small, but higher than 0. Even if $\epsilon$ is very small, the best action may change when a lover Q-value is allowed, i.e., the best action determined by $\theta_t$ may be different from the best action according to $Q(s,a)$. Do the smoothness assumptions made through the paper help to cope with the change of the best action in this case? Note that small $\epsilon$ may not be sufficient to make the result significant since the policy itself may be affected even when $\epsilon$ is tiny.

In line 201, the authors say that $C$, $\beta$, and $\mu_\min$ do not depend on $\theta$. But, I am not sure if this is true for $\mu_\min$ since the stationary distribution $\mu_\theta$ depends on the policy. When we have deterministic actions in the MDP, some states may even have probability of zero in the stationary distribution of the ensuing Markov chain.


**Questions:**

See previous box.

**Limitations:**

One thing to note is that I don't prove convergence rates in my work, and I did not go through the proofs to verify their correctness.

---

> ### Author Rebuttal · Authors · 2023-08-04
>
> Thank you for the detailed review and questions! We'll clarify these points in the final version of the paper. Meanwhile, our point-by-point replies are below.
>
> > In the paper, the authors emphasize the need to stay close to the initial conditions of the critic (e.g. in Sec. 2.4). This makes sense from the regularisation point of view in general, but in RL, this may mean that the optimal policy may not be found if the algorithm is forced to say close to the initial conditions. Perhaps the motivation and the consequences of staying close to the initial conditions could be clarified.
>
> Note that even in TD(0) with linear function approximation, a much simpler problem than the one considered here, one needs a projection to stabilize the algorithm in theory in the presence of Markovian sampling, see e.g., the analysis in Bhandari et.al (2018). The size of the projection radius is one of the improvements of this paper, as in many other NTK-style papers, the radius is even required to be decaying with $1/\sqrt{m}$, e.g., Cayci et.al (2022). So one of the contributions of this work is that it is much better on this point: the algorithm is required to stay a lot less close to the initial condition.
>
> It is natural to wonder, as the reviewer does, whether this projection radius leads to issues in the sense of the critic failing to approximate the true Q-values. In our simulations, we have found that this does happen, but is not as common as perfect approximation by the critic. Please see our global response for two examples where the critic is able to perfectly approximate the true value function under our assumption, and one case where it does not. We are not aware of any theoretical results analyzing the distinctions between these cases.
>
>
> > Consider Eq. (9), and assume that is small, but higher than 0. Even if is very small, the best action may change when a lover Q-value is allowed, i.e., the best action determined by may be different from the best action according to . Do the smoothness assumptions made through the paper help to cope with the change of the best action in this case? Note that small may not be sufficient to make the result significant since the policy itself may be affected even when is tiny.
>
> The smoothness assumptions are part of it, but note that as the reviewer points out, any $\epsilon > 0$, no matter how small, may induce a "discontinuity," in the sense that even at convergence it will cause different actions to be taken as compared to if the true $Q$-values are known. This is a particular obstacle to the analysis and smoothness assumption do not solve it by themselves. There are various bounds throughout our paper that are additive in $\epsilon$ (e.g., Lemma B.4), and this happens exactly for the reason the reviewer outlines -- there is an additive source of error that does not go to zero even as everything converges to the optimal values. Speaking informally,  what one needs to show is that all these additive errors do not get "amplified" by the AC dynamics.
>
>
> > In line 201, the authors say that $C$, $\beta$, and $\mu_{\rm min}$ do not depend on $\theta$. But, I am not sure if this is true for  since the stationary distribution  depends...
>
> Note that here $\mu_{\rm min}$ is a uniform lower-bound for the stationary distribution independent of $\theta$. So for this analysis, we have  assumed that $\mu_{\rm min}$ is bounded away from zero. In other words, we assumed the stationary distribution is bounded away from zero independently of $\theta$. Unfortunately, if this statement is not true, then one cannot prove anything about actor-critic methods. It is easy to see that if $\mu_{\theta}$ corresponding to some set of states is zero, then we may never converge to the optimal policy, even in the tabular case for both actor and critic (because we might never visit those states and so will never see the rewards associated with the actions in them).

---

> > ### Comment · Reviewer_FwfS · 2023-08-20
> > **Thank you for your answers**
> >
> > I am happy with your answers to my questions and I don't have anything to add.

---

### Official Review · Reviewer_ADfc · 2023-07-10

**Soundness:** 3 good
**Presentation:** 3 good
**Contribution:** 3 good
**Rating:** 5
**Confidence:** 1

**Summary:**

This is an RL theory paper with an improved convergence bound for actor-critic methods

**Strengths:**

To be frank, I'm not a theory person at all and I have no idea why this paper is assigned to me. I cannot really judge the theoretical contribution of this work. By assuming all the statements are correct, I personally (from a practitioner's perspective) feel that a convergence theorem that can apply to multi-layer networks looks great.

**Weaknesses:**

Out of my expertise.

**Questions:**

N/A

---

> ### Author Rebuttal · Authors · 2023-08-03
>
> We appreciate the reviewer's frankness! But we would like to to disagree: we think applied RL people are qualified to review RL-theory papers.
>
> Indeed, we do not want to be in a state where only RL-theory people are reviewing each other's papers: that is likely to lead to the community becoming insular and progressively more decoupled from practice. As far as this submission goes, somewhat remarkably it has 8 reviews; it could certainly benefit from feedback by someone who would consider it from a practical lens.
>
> Let us try to get the conversation started by explaining our overall agenda: our goal is to create something that would be useful for applied RL people.
>
> While there will always be gaps between theory and practice,  we would like to have a situation where, if a method diverges in practice, practitioners can always default to theoretically supported step-sizes/architectures/updates that would guarantee convergence. At the very least, the more practical methods could be tried again after the theoretically convergent method has been running for a while with the justification that a "good start" may be required for the more practical methods.
>
> We are currently quite far from this but we believe this paper is a worthwhile step towards this goal: we obtain the first results on neural actor critic that do not assume a single-hidden-layer. Nevertheless, let us point out some simplifications we have made that were necessary for us to get this result that do not match what practitioners do:
>
> * Small fixed step-size of $O(1/\sqrt{T})$. More generalized step-size schedules would be nice.
> * No replay buffer. In practice, replay buffers are pretty much universal in 2023.
> * Smooth activations: so Leaky ReLU, GeLU, ELU are all fine for our result, but ReLU is not.
> * No PPO-style truncation in updates.
> * Fixed projection radius. We would conjecture that while we cannot get rid of it without causing divergence or adding more assumptions, we can set the projection radius to be $\sim \sqrt{m}$.
>
>
> All of these are limitations of the current result. We are working on remedying these to bring the result closer to what occurs in practice and make the vision outlined above a reality. However, resolving any of these issues will likely require new ideas far beyond what is in the current paper (e.g., the single point in ReLU with no differentiability creates major problems that are not known to be surmountable -- at least, not without extra assumptions).
>
> While we do not expect an applied RL person to evaluate all the details of the proofs, it is certainly possible to evaluate both the ultimate value of the project of developing RL methods which come with convergence guarantees, as well as the place of this paper within that agenda.

---

> > ### Comment · Reviewer_ADfc · 2023-08-12
> >
> > To be clear, I did check the overall paper and liked the conclusion provided. That's why I have a score of 5 and will feel happy to see this paper accept.
> >
> > But, as you said, it is out of my capabilities to check the mathematical details. I do think correctness and soundness are critical. That's why I left a particularly low confidence score to reflect this in the hope that someone more professional can take the job.
> >
> > But anyway, good luck.

---

### Official Review · Reviewer_GAW7 · 2023-07-25

**Soundness:** 2 fair
**Presentation:** 2 fair
**Contribution:** 3 good
**Rating:** 6
**Confidence:** 2

**Summary:**

This paper presents a convergence analysis of Actor Critic method with multiply layer networks.

**Strengths:**

The convergence analysis of AC with multi-layer networks is important, as AC methods with neural networks plays the core role of the success of DRL.

**Weaknesses:**

The paper is difficult to follow and the writing can be significantly improved. (More in Questions)

**Questions:**

I would ask the authors to clarify the following questions, which I believe can significantly improve the paper if addressed:
1. What are the differences for AC methods with single hidden layer networks and multiple layer networks, especially for proving the convergence? In the current version, these differences are not well explained. To improve the clarity and strengthen the paper's contribution, the authors should provide a more detailed comparison of these two methods, highlighting the specific challenges that arise when proving the convergence for each architecture. This will enable readers to better understand the significance of the proposed approach in tackling the convergence problem and its relevance in the context of existing research.
2. The proof of convergence presented in the paper is challenging to follow, and its integration within the main context is inadequate. To improve the overall readability and accessibility of the paper, I recommend including a sketch of the proof in the main body, i.e., Section 3. This will allow readers, especially those unfamiliar with previous work on the convergence analysis of AC methods, to grasp the high-level idea behind the proof. Providing a concise outline of the proof in the main paper will enhance the paper's accessibility and make it more appealing to a broader audience.
3. Expanding on the suggestions mentioned in point 1, once the key differences between AC methods with single hidden layer networks and multiple layer networks are clearly stated, it would greatly benefit the readers if the authors could elaborate on the key techniques utilized to address these differences and overcome the associated challenges (Section 4 in the current version is far from satisfactory). By doing so, the authors can provide valuable insights into the novelty and contributions of the proposed approach. Understanding the techniques employed to tackle specific obstacles will enable the readers to evaluate the significance of the paper more effectively and appreciate its potential impact on the field.


-------------------
After rebuttal, as the authors address most of my concerns, I would increase my score to 6. I would still suggest the authors to carefully revise the paper to make it more readable if accepted.

**Limitations:**

No. The limitation is not discussed. As there are many assumptions, I suggest that the authors should discuss whether these assumptions can be relaxed, as well as the cases where these assumptions cannot hold.

---

> ### Author Rebuttal · Authors · 2023-08-04
>
> Thank you for bringing these points up! Our responses are below.
>
> > What are the differences for AC methods with single hidden layer networks and multiple layer networks, especially for proving the convergence? In the current version, these differences are not well explained. To improve the clarity and strengthen the paper's contribution, the authors should provide a more detailed comparison of these two methods, highlighting the specific challenges that arise when proving the convergence for each architecture. This will enable readers to better understand the significance of the proposed approach in tackling the convergence problem and its relevance in the context of existing research.
>
> It is accurate to say that previous work followed variations on the following strategy in the single-layer case:
>
> **(1)** Linearize around the initial point.
>
> **(2)** Argue that the linearization has singular values bounded away from zero.
>
> **(3)** As a consequence of (2), the loss decreases quickly in a neighborhood of the initial point.
>
> **(4)** Because of (3), the function never moves far away from the initial point (as loss decreases to zero, so does the size of the gradient).
>
> We say above "some variation" since every paper did something slightly different. For example, sometimes instead of (1), previous works added a small projection radius (size $O(1/\sqrt{m})$, where $m$ is the network width).
>
> To answer the reviewer's question: we believe the argument generally fails for higher depth in point (2), which is much harder to make for a multi-layer network. But for us this is somewhat beside the point, because **we do not want to make this argument work for the larger depth case** as it will also imply the neural network stays close to initialization -- which we do not want!
>
> Indeed, a natural counter-argument regarding the value of following (1)-(4) above is that the approximation scheme is essentially linear (since the closeness of the nonlinear map and  the first-order expansion around the initial point is what allows the whole proof strategy to go through).
>
> > To improve the overall readability and accessibility of the paper, I recommend including a sketch of the proof in the main body...
>
> > ...once the key differences between AC methods with single hidden layer networks and multiple layer networks are clearly stated, it would greatly benefit the readers if the authors could elaborate on the key techniques utilized to address these differences and overcome the associated challenges...
>
> We are happy to add this to the paper. Here is what this kind of sketch/discussion will look like. First, a sketch at the highest level of generality:
>
> **(1)** The nonlinearity of the network can be bounded in terms of the width $m$: larger width networks are "less nonlinear."
>
> Formally, here we rely on the previous results of Liu et al (2020) that the norm of the Hessian of a neural network in a constant ball around a random initial point is $O(1/\sqrt{m})$ with high probability. A linear function has zero gradient, so this bound may be informally viewed as capping "how nonlinear" the network is.
>
> **(2)** Analysis of linear AC is based on an iteration which says that the actor's reward $V(\theta_t)$ decreases at each step by something proportional to $||\nabla V(\theta_t)||^2$, with some error terms coming from the critic approximation. We can now add an additional error term to this recursion that comes from the nonlinearity. That error term has size $O(1/\sqrt{m})$ from item (1).
>
> **(3)** Finally, we need to argue that this error term is carried linearly into the final bound on how big $||\nabla V(\theta_t)||$ is at a random iterate.
>
> The difficulty here is that both the actor and the critic are using each other's updates, so adding an error term is far from trivial -- one must somehow argue the error-prone iterations are not mutually destabilizing.
>
> Now having made this sketch, we next point out the actual equations this translates to. A quick look at our paper will show that the analysis is quite technical and requires a lot of careful estimates -- but let us forget about this and just point at the two key equations. The **first** is
> \begin{align}
>     \left(\frac{\alpha_{\theta}}{2} - {\alpha_{\theta}}^2 H_V \right) \Delta_V
>     \le & \frac{E \left[ V_{\theta_1}^* - V_{\theta_{T+1}}^* \right]}{T} + c_1 \left( {\alpha_{\theta}}^2 H_V + \frac{\alpha_{\theta}}{2} \right) \Delta_Q + c_2 \left( {\alpha_{\theta}}^2 H_V + \frac{\alpha_{\theta}}{2} \right) \epsilon^2 + \frac{2 H_V}{(1-\gamma)^2} {\alpha^\theta}^2 U_g^2.
> \end{align} Precise definitions can be found in the paper itself, but informally:
>
> * $\Delta_V$ is the average gradient squared of the actor over $T$ iterations.
> * $\Delta_Q$ is the average approximation error of the critic over $T$ iterations.
> * $H_V$ is a measure of the nonlinearity of the neural approximation.
>
> What his equation says is: *if* the critic is accurate on average, and the network is not too nonlinear, then the actor has been close to a critical point.
>
> The **second** key equation is too long for OpenReview and pushes us above the character limit but it is Lemma D.1 in the paper. This makes the reverse argument: it argues that if the network is not too nonlinear and the actor gradients have, on average, been close to zero, then the critic average approximation error is small. Both of these key equations are conditional in character (they assert that if one thing is small, then another thing is small), but can be made unconditional using the "small gain" argument described in Section 4.2.
>
> > As there are many assumptions, I suggest that the authors should discuss whether these assumptions can be relaxed, as well as the cases where these assumptions cannot hold.
>
> With the exception of Assumption 2.7, our assumptions are standard and have been used in previous works. For a discussion of Assumption 2.7, please see our global response.

---

### Official Review · Reviewer_X8JD · 2023-07-26

**Soundness:** 3 good
**Presentation:** 4 excellent
**Contribution:** 3 good
**Rating:** 5
**Confidence:** 4

**Summary:**

The paper is a well written and clear result demonstrating the convergence of the actor critic algorithm. To my knowledge, this is the first example of global convergence of the actor critic algorithm using neural network parametrization. It is a good extension of the actor critic sample complexity analysis given in works such as Xu et.al (2020) from a linear function approximation to a neural network approximation for the value function.

**Strengths:**

Extends well established analysis of single timescale actor critic for a linear function approximation to one with a neural network approximation for the value function.

Provides better convergence bounds than current analyses of actor critic using neural network approximations, while not having the restriction in the depth of the networks that existing results have.

The paper is well written, concise and easy to follow.


**Weaknesses:**

Since a finite state space is being assumed here, the comparison to existing results  such as Wang et. al (2019) and Cayci et. al. (2022) does not seem to be valid. Both these works assume an infinite state space. Since the constant c1 is a multiple of the cardinality of the state space, an infinite state space does not seem to work for the analysis given here.

The upper bound on the norm of the Hessian of the neural network in Liu et.al (2020) is stated as a probabilistic bound. This bound is stated as deterministic in the lemma B.1.

Assumption 2.7 while being obvious for a linear function approximation, has not been assumed in the works cited where a neural network approximation has been used such as Cai. et. al (2019) and Xu and Gu (2020). Thus the validity of the assumption has not been established for the setup being analyzed.

**Questions:**

Can the existing result be extended to an infinite state space? That is not immediately clear from the analysis done here.

As a consequence of the finite state space, what is the advantage of assuming a neural network approximation here and not a tabular form of the value function?


**Limitations:**

Since this is a theoretical work which analyses existing algorithms negative societal impact is limited.

---

> ### Author Rebuttal · Authors · 2023-08-04
>
> > Since a finite state space is being assumed here, the comparison to existing results such as Wang et. al (2019) and Cayci et. al. (2022) does not seem to be valid. Both these works assume an infinite state space. Since the constant c1 is a multiple of the cardinality of the state space, an infinite state space does not seem to work for the analysis given here.
>
> Thank you for bringing this up! The answer is **yes, our analysis works for infinite state spaces with minor modifications**.
>
> We did not view this as an important difference between our work and the previous literature at submission time, which is why the proof as written only works for finite state spaces -- as the reviewer correctly points out. But this is quite easy to fix. Indeed, let us consider Lemma B.4 where the bounds for $c_1$ (and actually also $c_2$ ) brought up by the reviewer can be made independent of the number of states. We sketch this analysis below.
>
> We will use the notation in the paper. Our starting point is
>
> $$ (1-\gamma) D_Q = E_{(s,a) \sim (1-\gamma) \phi_{\theta_t}} [ \ln \pi(a|s, \theta_t) (Q(s,a,w_t)-Q_{\theta_t}^*(s,a)) ].$$
>
> Using Assumption 2.4, we have
>
> $$(1-\gamma) ||D_Q|| \le L_{\pi}' E_{(s,a) \sim (1-\gamma) \phi_{\theta_t}} [ |Q(s,a,w_t)-Q_{\theta_t}^*(s,a)| ].$$
>
> Next, using $(\mathbb{E}[X])^2 \le \mathbb{E}[X^2]$, we conclude that
>
> $$(1-\gamma)^2 ||D_Q||^2 \le {L_{\pi}'}^2 E_{(s,a) \sim (1-\gamma) \phi_{\theta_t}} [ |Q(s,a,w_t)-Q_{\theta_t}^*(s,a)|^2 ].$$
>
> But also, $$|Q(s,a,w_t)-Q_{\theta_t}^*(s,a)|^2 \le 2 |Q(s,a,w_t) - \hat{Q}_{\theta_t}^*(s,a)|^2 + 2 \epsilon^2,$$
>
> where we used Assumption 2.5. Hence,
>
> $$(1-\gamma)^2 ||D_Q||^2 \le 2 {L_{\pi}'}^2 (E_{(s,a) \sim (1-\gamma) \phi_{\theta_t}} [ |Q(s,a,w_t) - \hat{Q}_{\theta_t}^*(s,a)|^2] +\epsilon^2).$$
>
> Combining this with Lemma B.3 and the fact that $(1-\gamma)\phi_{\theta_t}(s,a) \le 1$, we obtain
>
> $$(1-\gamma)^2 ||D_Q||^2 \le \frac{2 {L_{\pi}'}^2}{\lambda_{\rm min}'} N_{\theta} ( Q(w) - \hat{Q}^*_{\theta} ) + 2 {L_{\pi}'}^2 \epsilon^2. $$
>
> We can thus set $c_1 = \frac{2 {L_{\pi}'}^2}{(1-\gamma)\lambda_{\rm min}'}, c_2 = \frac{2 {L_{\pi}'}^2}{1-\gamma},$ and this concludes the argument.
>
> Making the result work for infinite state spaces will require making a number of other modifications (e.g., various sums become integrals), but one can simply follow the argument we have laid out.
>
> > The upper bound on the norm of the Hessian of the neural network in Liu et.al (2020) is stated as a probabilistic bound. This bound is stated as deterministic in the lemma B.1.
>
> Correct: we assume Assumption 2.3 holds; in Liu et.al (2020), this fact was shown to hold with high probability over the random initialization (see Lemma F.4 in Liu et.al (2020)). In the final version of our paper, we will emphasize this difference to avoid confusion. Note that this was done solely to improve presentation: otherwise, one would need to add "with high probability over the initialization" to our main result.
>
> > Assumption 2.7 while being obvious for a linear function approximation, has not been assumed in the works cited where a neural network approximation has been used such as Cai. et. al (2019) and Xu and Gu (2020). Thus the validity of the assumption has not been established for the setup being analyzed.
>
> Our response on this point is three-fold. *First*, and most importantly, **please see our global response which contains simulations supporting this assumption**. It is, of course, not possible to test an assumption that begins with "for every $w$...", but what we can do is sample a lot of different $w$'s and see whether the resulting distribution has support bounded away from zero (and it does).
>
> *Second*: actually, our assumption is *almost identical* to a similar assumption in Xu and Gu (2020), namely Assumption 5.3 in their work. Let us spell this out.
>
> Indeed, it is trivial from the definitions that their assumptions implies ours when the function approximation is linear. In the general case of nonlinear approximation, we point out that in Definition 5.1 of Xu and Gu (2020), a linearization around the initial point is performed. If we do the same, i.e., if we approximate
>
> $$Q(w) \approx Q(w_0) + \nabla Q(w_0) (w-w_0),$$
>
> using the first term of the Taylor expansion, then our Assumption 2.7 becomes
>
> $$(w-\hat{w})^T {\nabla Q(w_0)}^T \nabla Q(w_0) (w-\hat{w}) \ge \lambda' (w-\hat{w})^T (w-\hat{w}).$$
>
> Now it is clear that our Assumption 2.7 follows if the smallest eigenvalue of ${\nabla Q(w_0)}^T \nabla Q(w_0)$ is bounded away from $0$, which is implied by Assumption 5.3 of Xu and Gu (2020).
>
> *Third*: to see that some assumption along these lines is necessary consider the case of linear approximation. In that case, existing analyses assume that the features are non-redundant, i.e., the feature matrix $\Phi$ (stacking up the feature vectors $\phi(s,a)^T$) has smallest singular value which is bounded away from zero.
>
> Now how can such an assumption be generalized to the nonlinear case? One way to guarantee that a matrix $\Phi$ has smallest singular value bounded away from zero is
>
>  $$ || \Phi w_1 - \Phi w_2 || \geq \sigma || w_1 - w_2 ||,$$
>
> for some $\sigma > 0$ and all $w_1, w_2$. But now this admits a straightforward generalization to the non-linear case:
>
> $$ || V(w_1) - V(w_2) || \geq \sigma  ||w_1 - w_2||,$$ which is slightly stronger than what we assume.
>
> **In summary,** we have made four main points in this rebuttal:
>
> * The various constants we bound can easily be made independent of $n$.
>
> * We have added experimental evidence supporting Assumption 2.7.
>
> * That assumption is almost identical to what was assumed in earlier work. Specifically, if we linearize in the same way as Xu and Gu (2020) do, then Assumption 2.7 is not as strong as what is assumed in that paper.
>
> * Assumption 2.7 is a natural way to generalize the standard non-redundancy of features assumption to the nonlinear case.

---

> > ### Comment · Reviewer_X8JD · 2023-08-17
> >
> > Thank you for your response to the review.
> >
> > With regards to the assumption 2.7, your global response as well as the response specifically to my review is convincing. I am raising my score for that.
> >
> > With regards to extension of the work to an infinite state space, the example of Lemma B.4 does indeed extend to an infinite state where the summation is replaced by an expectation with respect to a probability measure. However, consider the case of Lemma B.1. The constant $L_{Q}$ and $H_{Q}$ are defined as summations over $(s,a)$. I think the previous process of replacing the summation would not be applicable here.  Additionally note in lemma B.12 for the term I2 the optimal Q function is written in terms of the inverse of the probability matrix subtracted from the identity matrix. For a finite state space the bound on the inverse matrix term can be used from lemma B.11 which is taken from Olshevsky & Gharesifard (2022). However for an infinite state space does the bound in Olshevsky & Gharesifard (2022) still apply as the matrix is now infinite  dimensional? It does not look to be the case as in Lemma 5.6 of Olshevsky & Gharesifard (2022) only the infimum norm of the inverse matrix is shown to be bounded. This does imply that the l2 norm (I assume  ||.|| is referring to the l2 norm) is bounded for the finite state case, however if the state space is infinite, this does not appear to be the case. This term might indeed be bounded even for an infinite state space, however, it seems that some other technique than the one in Olshevsky & Gharesifard (2022) might be needed.
> >
> > For the probability statement, simply writing with high probability seems a little too informal. Typically, if the convergence result is probabilistic, it is written for example as in the  statement in theorem 8 of  Sun et al. [2019].
> >
> > Sun, Wen, Nan Jiang, Akshay Krishnamurthy, Alekh Agarwal, and John Langford. "Model-based rl in contextual decision processes: Pac bounds and exponential improvements over model-free approaches." In Conference on learning theory, pp. 2898-2933. PMLR, 2019.

---

> > > ### Author Response · Authors · 2023-08-17
> > >
> > > Thank you for raising your score! You raise some very good technical points and we provide point-by-point answers below. More generally, the proof does indeed contain several points where there is a sum of state-action pairs, and each of those points will require a few modifications to work with infinite state-spaces.
> > >
> > > >  Additionally note in lemma B.12 for the term I2 the optimal Q function is written in terms of the inverse of the probability matrix subtracted from the identity matrix. For a finite state space the bound on the inverse matrix term can be used from lemma B.11 which is taken from Olshevsky & Gharesifard (2022). However for an infinite state space does the bound in Olshevsky & Gharesifard (2022) still apply as the matrix is now infinite dimensional? It does not look to be the case as in Lemma 5.6 of Olshevsky & Gharesifard (2022) only the infimum norm of the inverse matrix is shown to be bounded. This does imply that the l2 norm (I assume ||.|| is referring to the l2 norm) is bounded for the finite state case, however if the state space is infinite, this does not appear to be the case. This term might indeed be bounded even for an infinite state space, however, it seems that some other technique than the one in Olshevsky & Gharesifard (2022) might be needed.
> > >
> > > Fair enough -- but please note that the point of Lemma B.12 is to establish the Lipschitzness of the value function. If one considers continuous states and actions instead, one could simply use versions of Lipschitzness which have already been established in the literature. In particular, see Theorem 3 ``Policy Gradient in Markov Decision Processes,'' Pirotta, Restelli, Bascetta, Machine Learning 100:255-283, 2015 which provides what is needed here.
> > >
> > > However, we acknowledge the reviewer's point here in that this requires some modifications: in particular, the above paper makes more assumptions than ours does, both to deal with the continuous states/actions, as well as in terms of requiring slightly stronger smoothness conditions in parametrizing the policy.
> > >
> > > > However, consider the case of Lemma B.1. The constant and are defined as summations over $(s,a)$ . I think the previous process of replacing the summation would not be applicable here.
> > >
> > > Indeed. To get around the sums over state-action pairs in Lemma B.1, one can proceed as follows:
> > >
> > > (1) Change the sums in Lemma B1 in definitions of $L_Q, H_Q$ to a $\sup$
> > >
> > > (2) Lemma B.1 still holds with this change. In particular, the key statement is that $H_Q=O(1/m)$ holds because this was demonstrated in Liu et al. for any bounded input -- so we can certainly take the supremum over a compact set of inputs.
> > >
> > > (3) Now let us see how the rest of the proof works, focusing on the key issue, the scaling of $H_Q$ as $O(1/m)$. The $H_Q$ makes its first appearance in the block of equations in lines 571-572 of the paper. This block is arguably one of the key equation of the paper since it puts together various lemmas to analyze what happens to the Lyapunov function for the critic on a per-iteration basis. It is of the form
> > > $$ \mbox{ inner product between expected update and direction to optimal solution} \leq \mbox{ something negative } + \mbox{ error terms } $$
> > >
> > > Now the key point is that the LHS here has the form:
> > >
> > > $$ a^T D (\gamma P - I) b$$
> > >
> > > where $a,b$ are vectors and $D$ is a diagonal matrix whose entries are nonnegative and add up to one, $P$ is a stochastic matrix, and $I$ is, as always, the identity matrix. The expressions on lines 571-572 are lengthy, but they actually all have the same form as the above.
> > >
> > > It is immediate that we can bound this as
> > >
> > > $$ a^T D (\gamma P - I) b \leq ||a|| \cdot ||b||,$$
> > >
> > > **where the norms on the RHS are infinity norms** -- so we can use $H_Q$ to bound the size of *each entry* of the vectors $a$ and $b$. *The key thing that makes it work is that $P$ is a stochastic matrix and $D$ is a diagonal matrix with entries summing to one.*  It should now be clear that this argument will work verbatim if the number of state-action pairs is e.g., countable -- it makes no difference whether the sum if a scountable or finite sum. From there one can generalize to infinite state-space by replacing sums by integrals.
> > >
> > >
> > > If we had anticipated that the difference between finite-state-finite-action and infinite would have been brought up as important, we would have been careful to write these arguments in a way that made all this easy to see. As things stand, we can assert that we don't see any obstacles to making this work for infinite state spaces, but we appreciate that this may not be apparent to someone reviewing the paper.

---

> > > > ### Author Response · Authors · 2023-08-17
> > > >
> > > > >For the probability statement, simply writing with high probability seems a little too informal. Typically, if the convergence result is probabilistic, it is written for example as in the statement in theorem 8 of Sun et al. [2019].
> > > >
> > > > Here the situation is slightly different. We assume a certain statement about the initialization, and previous literature gives a theorem to the effect this statement holds with high probability over random initializations. We then go on to prove that $E[\mbox{error}] \leq \mbox{error bound}$ under this assumption.
> > > >
> > > > If one objects to this, what we could do is define event $A$ to be the event that the aforementioned ``certain statement'' holds. We then have the pair of assertions that, first,  $E[\mbox{error}| A] \leq \mbox{error bound}$, and second that event $A$ holds with high probability.

---

### Official Review · Reviewer_akFw · 2023-07-27

**Soundness:** 4 excellent
**Presentation:** 4 excellent
**Contribution:** 4 excellent
**Rating:** 7
**Confidence:** 5

**Summary:**

This paper establishes the convergence of single-timescale actor-critic with neural networks representing the value and policy with > 1 layer, strengthening over prior results in the linear setting and two-scale approaches.


**Strengths:**

I will preface my review by saying that I have little background on convergence of actor-critic methods or in analyses of deep networks -- and thus not much to say about the significance of the technical advances.

I found that despite this lack of background, this paper was an absolute pleasure to read. The paper is very well-written, and the authors do a great job of motivating the problem and the technical approach. Each assumption is well-motivated, and the two tools -- nonlinear gradient splitting and the nonlinear small gain theorem -- are also described in a way that is easy to understand. I wish that more theory papers were written like this!

While I briefly looked through the proofs in the appendix, I unfortunately do not have the expertise to gauge correctness.


**Weaknesses:**

While the mechanism of the proof was very well explained in the paper, I would have liked to see some more discussion about the significance of the result and it's implications for future work. Why is this result interesting? What does it enable? Perhaps it would be useful to spend a little more time discussing the applicability of the proposed tools and theory beyond their application to AC -- what other places may these technical tools be useful?

**Questions:**

1. I would have loved to see a little more discussion on future directions. What are the next steps to relax? Or is it to more tightly characterize the convergence?

2. How does this play with value learning when the TD objective does not correspond to a gradient descent (e.g. in off-policy learning)?

---

> ### Author Rebuttal · Authors · 2023-08-04
>
> > I found that despite this lack of background, this paper was an absolute pleasure to read. The paper is very well-written, and the authors do a great job of motivating the problem and the technical approach. Each assumption is well-motivated, and the two tools -- nonlinear gradient splitting and the nonlinear small gain theorem -- are also described in a way that is easy to understand. I wish that more theory papers were written like this!
>
> Thank you very much for your encouraging comments! We have done our best to make the core result of our work as accessible as possible.
>
> > I would have loved to see a little more discussion on future directions. What are the next steps to relax? Or is it to more tightly characterize the convergence?
>
> Thanks for your question! We'll discuss the next steps at a level which is more speculative than what is in our submission. Note that the discussion here overlaps with our responses to some of the other reviewers.
>
> *Our ultimate goal is to create something that would actually be useful for applied RL*. While there will always be gaps between theory and practice,  we would like to have a situation where, if a particular method diverges in practice, one could always default to theoretically supported step-sizes/architectures/updates that would guarantee convergence.
>
> We are currently quite far from this but we believe this paper is a worthwhile step forward. Our next step is to relax some of the ingredients that went into obtaining this result that do not match what practitioners do. These ingredients include:
>
> * Small fixed step-size of $O(1/\sqrt{T})$. More generalized step-size schedules would be nice.
> * No replay buffer. In practice, replay buffers are pretty much universal in 2023.
> * Smooth activations: so Leaky ReLU, GeLU, ELU are all fine for our result, but ReLU is not.
> * No PPO-style truncation in updates.
> * Fixed projection radius. We would conjecture that while we cannot get rid of it without causing divergence or adding more assumptions, we can set the projection radius to be $\sim \sqrt{m}$.
>
> >  How does this play with value learning when the TD objective does not correspond to a gradient descent (e.g. in off-policy learning)?
>
> That is a good question. We have been thinking about this for a while without making much progress; below, we summarize our current state of thinking.
>
> In the off-policy case, we have an additional multiplicative factor that comes from the difference between behavior and policy  distribution. Ignoring the differences between gradient and gradient splittings, off-policy TD is then analogous to the update:
>
> $$x_{t+1} = x_t - \alpha_t D \nabla f(x_t),$$
>
> where $D$ is diagonal matrix with entries which are strictly positive. Analyzing this as written above not a problem -- but unfortunately if you take the above equation and replace gradient with gradient splitting, then we have no idea how to analyze it.
>
> This is a good juncture to emphasize that it is *not* true that we can take any arguments from convex optimization and make them work for TD by arguing that one is simply replacing gradients with gradient-splittings (a number of other reviewers were wondering about this point). Indeed, there are properties that gradients have that splittings do not, and they can make a big difference. It may still be possible to use the toolkit we use in this paper to analyze off-policy TD; however, doing so will not be straightforward and will likely require some new ideas.

---

### Author Rebuttal · Authors · 2023-08-09

Dear Reviewers,

We would like to use the "global" review to provide two simulations asked for by some of the reviewers.

*First*, one reviewer (implicitly) asks for evidence supporting our Assumption 2.7. Our attached pdf actually
shows evidence supporting the stronger assumption

$$ || Q(w_1) - Q(w_2) || \geq \lambda ||w_1 - w_2||,$$

for all neural networks $Q$ generated during the actor-critic process, and all $w_1, w_2$ in the set onto which we are projecting;
crucially, $\lambda$ here is a parameter that should be bounded away from zero. Since it is impossible to test this for all $w_1,w_2$, we use  randomly chosen points. We have performed many simulations for Cartpole, MountainCar, and Acrobot (three finite-action MDPs in OpenAI Gym) and everything we have seen is consistent with being able to take $\lambda=e^{0.75}$ in the above equation. For an example of the kind of simulations we are doing, please see the three example runs in the attached PDF, where the quantity $||Q(w_1) - Q(w_2)||/||w_1 - w_2||$ is labeled as ``ratio.''

Let us take the opportunity to explain that Assumption 2.7 is a straightforward generalization of the non-redundant features assumption in linear actor-critic. Indeed, in linear AC one stacks up the feature vectors $\phi(s,a)^T$ into a matrix $\Phi$ (so that the value function can be approximated as $V \approx \Phi \theta$) and assumes that the smallest singular value of $\Phi$ is bounded away from zero. This is equivalent to non-redundancy of features (i.e., you should not be able to write one component of the vectors $\phi(s,a)^T$ as a linear combination of the other components).

Another way to state this condition is that

$$|| \Phi x - \Phi y || \geq \lambda ||x-y||$$

for some $\lambda$ and all $x,y$. It should now be immediate how our assumption above is a straightforward generalization of this to the nonlinear case. The discussion around this assumption in our manuscript contains several heuristic (i.e., non-rigorous) justifications for why this assumption holds, and we supplement it here with some simulations.

*Second*, one of the reviewers brought up the possibility that initializing a random network randomly and projecting onto a ball of constant radius around that set can lead to a loss of representation abilities. That is indeed a concern (though we point out that previous works required projection on a ball of radius $O(1/\sqrt{m})$ around the initial point, where $m$ is the width, so our paper should be viewed as a step forwad to mitigate this). We show in the attached pdf (right-hand column labeled ``error'') that this does not happen in Cartpole and Acrobot, where the critic seems to be able to perfectly approximate the value function of the actor; but we also show this may happen in MountainCar where the simulations are consistent with the critic error having a floor away from zero.

---

### Decision · Program_Chairs · 2023-09-21

**Decision:**

Accept (poster)

**Comment:**

This paper studies the convergence rates of an actor-critic algorithm for reinforcement learning that use deep neural networks with multiple hidden layers for both the actor and the critic. It fills a gap in existing literature, primarily focusing on linear or single-layer neural network representations.  Overall, the work is well-written and represents a significant theoretical contribution to understanding actor-critic algorithms with complex neural network architectures. We request the authors to include the suggested changes/clarifications in the final version of the paper.